# Proportional Fairness in Federated Learning

**Guojun Zhang, Saber Malekmohammadi, Xi Chen**
*{guojun.zhang, saber.malekmohammadi, xi.chen4}@huawei.com*
*Huawei Noah's Ark Lab*

**Yaoliang Yu**
*yaoliang.yu@uwaterloo.ca*
*University of Waterloo*

**Reviewed on OpenReview:** *https: // openreview. net/ forum? id= ryUHgEdWCQ*

## Abstract

With the increasingly broad deployment of federated learning (FL) systems in the real world, it is critical but challenging to ensure fairness in FL, i.e. reasonably satisfactory performances for each of the numerous diverse clients. In this work, we introduce and study a new fairness notion in FL, called *proportional fairness* (PF), which is based on the relative change of each client's performance. From its connection with the bargaining games, we propose *PropFair*, a novel and easy-to-implement algorithm for finding proportionally fair solutions in FL, and study its convergence properties. Through extensive experiments on vision and language datasets, we demonstrate that PropFair can approximately find PF solutions, and it achieves a good balance between the average performances of all clients and of the worst 10% clients. Our code is available at `https://github.com/huawei-noah/Federated-Learning/tree/main/FairFL`.

## 1 Introduction

Federated learning (FL, McMahan et al. 2017) has attracted an intensive amount of attention in recent years, due to its great potential in real world applications such as IoT devices (Imteaj et al., 2021), healthcare (Xu et al., 2021) and finance (Long et al., 2020). In FL, different clients collaboratively learn a global model that presumably benefits all, without sharing the local data.

However, clients differ. Due to the heterogeneity of client objectives and resources, the benefit each client receives may vary. How can we make sure each client is treated *fairly* in FL?

To answer this question, we first need to define what we mean by fairness. Similar to fairness in other fields (Jain et al., 1984; Sen, 1986; Rawls, 1999; Barocas et al., 2017), in FL, there is no unified definition of fairness. In social choice theory, two of the most popular definitions are *utilitarianism* and *egalitarianism*. The goal of utilitarian fairness is to maximize the utility of the total society; while egalitarian fairness requires the worst-off people to receive enough benefits. Coincidentally, they correspond to two of the fair FL algorithms: Federated Averaging (FedAvg, McMahan et al. 2017) and Agnostic Federated Learning (AFL, Mohri et al. 2019). In FedAvg (AFL), we minimize the averaged (worst-case) loss function, respectively. Utilitarian and egalitarian might be in conflict with each other: one could improve the worst-case clients, but better-off clients would be degraded to a large extent.

To achieve some balance between utilitarian and egalitarian fairness, other notions of fairness have been studied. Inspired by $\alpha$-*fairness* from telecommunication (Mo & Walrand, 2000), Li et al. (2020c) proposed $q$-Fair Federated Learning ($q$-FFL). By replacing the client weights with the softmax function of the client losses, Li et al. (2020a) proposed Tilted Empirical Risk Minimization (TERM). However, it remains vague what type of balance these algorithms are trying to yield.

In this work, we bring another fairness notion into the zoo of fair FL, called *proportional fairness* (PF, Kelly 1997). It also balances between utilitarian and egalitarian fairness, but is more intuitive. As a illustrative example, suppose we only have two clients and if we can improve the performance of one client *relatively* by 2% while decreasing another one by 1%, then the solution is more proportionally fair. In practice, this view of relative change is quotidian. In stock market, people care more about how much they gain/lose compared to the cost; in telecommunication, people worry about the data transmission speed compared to the bandwidth. In a word, PF studies the *relative change* of each client, rather than the *absolute change*.

Under convexity, PF is equivalent to the *Nash bargaining solution* (NBS, Nash 1950), a well-known concept from cooperative game theory. Based on the notion of PF and its related NBS, we propose a new FL algorithm called *PropFair*. Our contributions are the following:

- With the *utility* perspective and Nash bargaining solutions, we propose a surrogate loss for achieving proportionally fair FL. This provides new insights to fair FL and is distinct from existing literature which uses the *loss* perspective for fairness (see Section 2).

- *Theoretical guarantee:* we prove the convergence of PropFair to a stationary point of our objective, under mild assumptions. This proof can generalize to any other FL algorithm in the unified framework we propose.

- *Empirical viability:* we test our algorithm on several popular vision and language datasets, and modern neural architectures. Our results show that PropFair not only approximately obtains proportionally fair FL solutions, but also attains more favorable balance between the averaged and worst-case performances.

- Compared to previous works (Mohri et al., 2019; Li et al., 2020c; 2021), we provide a comprehensive benchmark for popular fair FL algorithms with systematic hyperparameter tuning. This could facilitate future fairness research in FL.

Note that we mainly focus on fairness in *federated learning*. Perhaps more widely known and orthogonal to fair FL, fairness has also been studied in general machine learning (Appendix F.3.3), such as demographic parity (Dwork et al., 2012), equalized odds (Hardt et al., 2016) and calibration (Gebel, 2009). These definitions require knowledge of sensitive attributes and true labels. Although it is possible to adapt these fairness definitions into FL, by e.g., treating each sensitive attribute as a client, the adaptation may not always be straightforward due to the unique challenge of privacy in FL. Such adaptation can be interesting future work and we do not consider it here.

**Notations.** We use $\boldsymbol{\theta}$ to denote the model parameters, and $\ell(\boldsymbol{\theta}, (\boldsymbol{x}, y))$ to represent the prediction loss of $\boldsymbol{\theta}$ on the sample $(\boldsymbol{x}, y)$. $\ell_S$ denotes the average prediction loss on batch $S$. For each client $i$, the data distribution is $\mathcal{D}_i$ and the expected loss of $\boldsymbol{\theta}$ on $\mathcal{D}_i$ is $f_i$. We denote $\boldsymbol{f} = (f_1, \ldots, f_n)$ with $n$ the number of clients, and use $p_i$ as the linear weight of client $i$. Usually, we choose $p_i = n_i/N$ with $n_i$ the number of samples of client $i$, and $N$ the total number of samples across all clients. We use $\Phi : \mathbb{R}^n \to \mathbb{R}$ to denote the scalarization of $\boldsymbol{f}$ and $\varphi : \mathbb{R} \to \mathbb{R}$ for some scalar function that operates on each $f_i$. We denote $\boldsymbol{\lambda} \in \mathbb{R}^n$ as the dual parameter, and $\mathsf{A}_\varphi$ as Kolmogorov's generalized mean. The utilities of each client $i$ is $u_i \in \mathbb{R}$ whose exact definition depends on the context, and $\boldsymbol{u} = (u_1, \ldots, u_n)$ denotes the vector all client utilities. A more complete notation table can be found in Appendix A.

## 2 A Unified Framework of Fair FL Algorithms

Suppose we have $n$ clients, and a model parameterized by $\boldsymbol{\theta}$. Because of data heterogeneity, for each client $i$ the data distribution $\mathcal{D}_i$ is different. The corresponding loss function becomes:

$$f_i(\boldsymbol{\theta}) := \mathbb{E}_{(\boldsymbol{x}, y) \sim \mathcal{D}_i}[\ell(\boldsymbol{\theta}, (\boldsymbol{x}, y))], \tag{2.1}$$

where $\ell$ is the prediction loss (such as cross entropy) of model $\boldsymbol{\theta}$ for each sample. The goal of FL is essentially to learn a model $\boldsymbol{\theta}$ that every element in the vector $\boldsymbol{f} = (f_1, \ldots, f_n)$ is small, a.k.a. multi-objective optimization (MOO, Jahn et al. 2009). Hu et al. (2022) took this approach and used Multiple Gradient Descent Algorithm (MGDA) to find Pareto stationary points.

Another popular approach to MOO is scalarization of $\boldsymbol{f}$ (Chapter 5, Jahn et al., 2009), by changing the vector optimization to some scalar optimization: $\min_{\boldsymbol{\theta}}(\Phi \circ \boldsymbol{f})(\boldsymbol{\theta})$ with $\Phi : \mathbb{R}^n \to \mathbb{R}$. In this work, we mainly focus on $\Phi$ being a (additively) separable function:

$$\min_{\boldsymbol{\theta}} \sum_i p_i (\varphi \circ f_i)(\boldsymbol{\theta}), \quad \varphi : \mathbb{R} \to \mathbb{R}. \tag{2.2}$$

The linear weights $p_i$'s are usually pre-defined and satisfy $p_i \geq 0, \sum_i p_i = 1$. In FL, a usual choice of $p_i$ is $p_i = n_i/N$ with $n_i$ the number of samples for client $i$ and $N$ the total number of samples. Here $\varphi$ is a monotonically increasing function, since if any $f_i$ increases, the total loss should also increase.

In order to properly locate proportional fairness in the fairness literature, we first review existing fairness definitions that have been applied to FL. In the following subsections, we show that different choices of scalar function $\varphi$ lead to different fair FL algorithms with their respective fairness principles.

## 2.1 Utilitarianism

The simplest choice of $\varphi$ would be the identity function, $\varphi(f_i) = f_i$:

$$\min_{\boldsymbol{\theta}} F(\boldsymbol{\theta}) := \sum_i p_i f_i(\boldsymbol{\theta}), \text{ with } p_i \geq 0 \text{ pre-defined, and } \sum_i p_i = 1. \tag{2.3}$$

This corresponds to the first FL algorithm, Federated Averaging (FedAvg, McMahan et al. 2017). Combined with eq. 2.1, the objective eq. 2.3 is equivalent to centralized training with all the client samples in one place.

From a fairness perspective, eq. 2.3 can be called *utilitarianism*, which can be traced back to at least Bentham (1780). From a utilitarian perspective, a solution $\boldsymbol{\theta}$ is fair if it maximizes an average of the client utilities. (Here we treat client $i$'s utility $u_i$ as $-f_i$. In general, $u_i \in \mathbb{R}$ is some value client $i$ wishes to maximize.)

## 2.2 Egalitarianism (Maximin Criterion)

In contrast to FedAvg, Agnostic Federated Learning (AFL, Mohri et al. 2019) does not assume a pre-defined weight for each client, but aims to minimize the worst convex combination:

$$\min_{\boldsymbol{\theta}} \max_{\boldsymbol{p}} \sum_i p_i f_i(\boldsymbol{\theta}), \text{ with } \mathbf{1}^\top \boldsymbol{p} = 1, \text{ and } \boldsymbol{p} \geq \mathbf{0}. \tag{2.4}$$

Note that $\boldsymbol{p} \in \mathbb{R}^n$ is a vector on the probability simplex. An equivalent formulation is:

$$\min_{\boldsymbol{\theta}} \max_i f_i(\boldsymbol{\theta}). \tag{2.5}$$

In other words, we minimize the worst-case client loss. In social choice, this corresponds to the egalitarian rule (or more specifically, the maximin criterion, see Rawls 1974). In MOO, this corresponds to $\Phi(\boldsymbol{f}) = \max_i f_i$ (above eq. 2.2). There is one important caveat of AFL worth mentioning: the generalization. In practice, each client loss $f_i$ is in fact the expected loss on the empirical distribution $\widehat{\mathcal{D}}_i$, i.e.,

$$\widehat{f}_i(\boldsymbol{\theta}) = \mathbb{E}_{(\boldsymbol{x},y) \sim \widehat{\mathcal{D}}_i}[\ell(\boldsymbol{\theta}, (\boldsymbol{x}, y))]. \tag{2.6}$$

In FL, some clients may have few samples and the empirical estimate $\widehat{f}_i$ may not faithfully reflect the underlying distribution. If such a client happens to be the worst-case client, then AFL would suffer from defective generalization. We provide a concrete example in Appendix C, and this phenomenon has also been observed in our experiments.

## 2.3 $\alpha$-Fairness

Last but not least, we may slightly modify the function $\varphi$ in FedAvg to be $\varphi(f_i) = f_i^{q+1}/(q+1)$:

$$\min_{\boldsymbol{\theta}} \frac{1}{q+1} \sum_i p_i f_i^{q+1}(\boldsymbol{\theta}), \text{ with } p_i \geq 0 \text{ pre-defined, and } \sum_i p_i = 1. \tag{2.7}$$

This is called $q$-Fair Federated Learning ($q$-FFL, Li et al. 2020c), and $q \geq 0$ is required. If $q = 0$, then we retrieve FedAvg; if $q \to \infty$, then the client who has the largest loss $f_i$ will be emphasized more, which corresponds to AFL. In general, $q$-FFL interpolates between the two. From a fairness perspective, $q$-FFL can relate to $\alpha$-fairness (Mo & Walrand, 2000), a popular concept from the field of communication. Suppose each client has utility $u_i \in \mathbb{R}$ and $\boldsymbol{u} = (u_1, \ldots, u_n) \in \mathcal{U} \subseteq \mathbb{R}^n$, with $\mathcal{U}$ the feasible set of client utilities, then $\alpha$-fairness associates with the following problem:

$$\max_{\boldsymbol{u} \in \mathcal{U}} \sum_i p_i \phi_\alpha(u_i), \text{ with pre-defined } p_i \geq 0 \text{ and } \phi_\alpha(u_i) = \begin{cases} \log u_i & \text{if } \alpha = 1, \\ u_i^{1-\alpha}/(1-\alpha) & \text{if } \alpha > 0 \text{ and } \alpha \neq 1. \end{cases} \tag{2.8}$$

$q$-FFL modifies the $\alpha$-fairness with two changes: (1) take $\alpha = -q$, and allow $\alpha \leq 0$; (2) replace $u_i$ with the loss $f_i$. Therefore, $q$-FFL is an analogy of $\alpha$-fairness. However, the objective eq. 2.7 misses the important case with $\alpha = 1$, also known as *proportional fairness* (PF, Kelly et al. 1998), which we will study in § 3. Note that the formulation eq. 2.7 is not fit for studying PF, since if we take $q \to -1$ (corresponding to $\alpha = 1$), then we obtain $\sum_i p_i \log f_i$, which need not be convex even when each $f_i$ is (see also § 3.1.1).

## 2.4 Dual View of Fair FL Algorithms

In this subsection, we show that many existing fair FL algorithms can be treated in a surprisingly unified way. In fact, eq. 2.2 is equivalent to minimizing the Kolmogorov's generalized mean (Kolmogorov, 1930):

$$\mathsf{A}_\varphi(\boldsymbol{f}(\boldsymbol{\theta})) := \varphi^{-1}\left(\sum_{i=1}^n p_i \varphi(f_i(\boldsymbol{\theta}))\right). \tag{2.9}$$

Examples include $\varphi(f_i) = f_i$ (FedAvg), $\varphi(f_i) = f_i^{q+1}$ ($q$-FFL, $q \geq 0$) and $\varphi(f_i) = \exp(\alpha f_i)$ ($\alpha \geq 0$). The last choice is known as Tilted Empirical Risk Minimization (TERM, Li et al. 2020a).

We can now supply a dual view of the aforementioned FL algorithms that is perhaps more revealing. Concretely, let $\varphi$ be (strictly) increasing, convex and thrice differentiable. Then, the generalized mean function $\mathsf{A}_\varphi$ is convex iff $-\varphi'/\varphi''$ is convex (Theorem 1, Ben-Tal & Teboulle, 1986). Applying the convex conjugate of $\mathsf{A}_\varphi$ we obtain the equivalent problem:

$$\min_{\boldsymbol{\theta}} \mathsf{A}_\varphi(\boldsymbol{f}(\boldsymbol{\theta})) \equiv \min_{\boldsymbol{\theta}} \max_{\boldsymbol{\lambda} \geq 0} \sum_i \lambda_i f_i(\boldsymbol{\theta}) - \mathsf{A}_\varphi^*(\boldsymbol{\lambda}), \ \mathsf{A}_\varphi^*(\boldsymbol{\lambda}) := \sup_{\boldsymbol{f}} \boldsymbol{\lambda}^\top \boldsymbol{f} - \mathsf{A}_\varphi(\boldsymbol{f}), \tag{2.10}$$

where $\mathsf{A}_\varphi^*(\boldsymbol{\lambda})$ is the *convex conjugate* of $\mathsf{A}_\varphi$. Note that $\boldsymbol{f} \geq \boldsymbol{0}$ and thus we require $\boldsymbol{\lambda} \geq \boldsymbol{0}$. Under strong duality, we may find the optimal dual variable $\boldsymbol{\lambda}^*$, with which our fair FL algorithms are essentially FedAvg with the fine-tuned weighting vector $\boldsymbol{\lambda}^*$.

**Constraints of $\boldsymbol{\lambda}$.** Solving the convex conjugate $\mathsf{A}_\varphi^*$ often gives additional constraints on $\boldsymbol{\lambda}$. For example, for FedAvg we can find that $\mathsf{A}_\varphi^*(\boldsymbol{\lambda}) = 0$ if $\lambda_i = p_i$ for all $i \in [n]$ and $\mathsf{A}_\varphi^*(\boldsymbol{\lambda}) = \infty$ otherwise. For $\varphi(f_i) = f_i^{q+1}$, we obtain the conjugate function corresponding to $q$-FFL:

$$\mathsf{A}_\varphi^*(\boldsymbol{\lambda}) = 0, \text{ if } \boldsymbol{\lambda} \geq \boldsymbol{0} \text{ and } \sum_i p_i^{-1/q} \lambda_i^{(q+1)/q} \leq 1, \text{ and } \infty \text{ otherwise.} \tag{2.11}$$

Bringing eq. 2.11 into eq. 2.10 and using Hölder's inequality we obtain the maximizer $\lambda_i \propto p_i f_i^q$. Similarly, we can derive the convex conjugate of TERM (Li et al., 2020a) as:

$$\mathsf{A}_\varphi^*(\boldsymbol{\lambda}) = \sum_i \frac{\lambda_i}{\alpha} \log \frac{\lambda_i}{p_i} \text{ if } \boldsymbol{\lambda} \geq \boldsymbol{0}, \boldsymbol{\lambda}^\top \boldsymbol{1} = 1, \text{ and } \infty \text{ otherwise.} \tag{2.12}$$

The maximizer is achieved at $\lambda_i \propto p_i e^{\alpha f_i}$. In other words, TERM gives a higher weight to clients with worse losses. Detailed derivations of the convex conjugates can be found in Appendix E.

In Table 1, we summarize all the algorithms we have discussed, including their motivating principles, objectives as well as the constraints of $\boldsymbol{\lambda}$ induced by $\mathsf{A}_\varphi^*$. Although the fair FL algorithms are motivated from different principles, most of them achieve a balance between utilitarianism and egalitarianism, thus allowing us to compare them on the same ground (§ 5).

Table 1: Different fairness concepts and their corresponding FL algorithms. $f_i$ is the loss function for the $i^{\text{th}}$ client. The requirement of $\boldsymbol{\lambda}$ can be found in § 2.4 and § 3.2. We defer the description and the dual view of PropFair to § 3.

| FL algorithm | Principle | Objective | Constraints of $\boldsymbol{\lambda}$ |
|---|---|---|---|
| FedAvg | Utilitarian | $\sum_i p_i f_i$ | $\lambda_i = p_i$ |
| AFL | Egalitarian | $\max_i f_i$ | $\boldsymbol{\lambda} \geq \boldsymbol{0}, \boldsymbol{1}^\top \boldsymbol{\lambda} \leq 1$ |
| $q$-FFL | $\alpha$-fairness | $\sum_i p_i f_i^{q+1}$ | $\lambda_i \propto p_i f_i^q, \boldsymbol{\lambda} \geq \boldsymbol{0}$ |
| TERM | n/a | $\sum_i p_i e^{\alpha f_i}$ | $\lambda_i \propto p_i e^{\alpha f_i}, \boldsymbol{\lambda} \geq \boldsymbol{0}, \boldsymbol{1}^\top \boldsymbol{\lambda} = 1$ |
| **PropFair** | **Proportional** | $-\sum_i p_i \log(M - f_i)$ | $\lambda_i \propto \frac{p_i}{M - f_i}, \prod_i (\lambda_i/p_i)^{p_i} = 1$ |

## 3 Adapting Proportional Fairness to FL

Now we study how to add the missing piece mentioned in Section 2.3 to FL: proportional fairness. From a utility perspective, eq. 2.8 with $\alpha = 1$ reduces to:

$$\max_{\boldsymbol{u} \in \mathcal{U}} \sum_i p_i \log u_i, \text{ with } p_i \geq 0 \text{ pre-defined, and } \sum_i p_i = 1. \tag{3.1}$$

Note that we now specify the domain of $\boldsymbol{u}$ to be $\mathcal{U} \subseteq \mathbb{R}_{++}^n$. The objective in eq. 3.1 is sometimes known as the *Nash product* (up to logarithmic transformation), and the maximizer $\boldsymbol{u}^*$ is also called the *Nash bargaining solution* (NBS, Nash 1950). Axiomatic characterizations of the Nash bargaining solution are well-known, for instance by the following four axioms: Pareto optimality, symmetry, scale equivariance and monotonicity (e.g., Maschler et al., 2020, Theorem 16.35). Moreover, Figure 1 gives an illustration of the NBS. Among all the solutions that maximize the total utility, the Nash bargaining solution achieves equal utility for the two players, and the largest worst-case utility.

The first-order optimality condition (Bertsekas, 1997) of eq. 3.1 can be written as:

$$\langle \boldsymbol{u} - \boldsymbol{u}^*, \nabla \sum_{i=1}^n p_i \log u_i^* \rangle \leq 0, \text{ for any } \boldsymbol{u} \in \mathcal{U}, \tag{3.2}$$

resulting in the following definition of proportional fairness (Kelly et al., 1998):

$$\boldsymbol{u}^* \in \mathcal{U} \text{ is proportionally fair if } \sum_i p_i \frac{u_i - u_i^*}{u_i^*} \leq 0, \text{ for any } \boldsymbol{u} \in \mathcal{U}. \tag{3.3}$$

Intuitively, $(u_i - u_i^*)/u_i^*$ is the relative utility gain for player $i$ given its utility switched from $u_i^*$ to $u_i$. PF simply states that at the solution $\boldsymbol{u}^*$, the average relative utility cannot be improved. For instance, for two players with $p_1 = p_2 = 1/2$ we have:

$$\frac{u_1 - u_1^*}{u_1^*} \leq -\frac{u_2 - u_2^*}{u_2^*}, \tag{3.4}$$

which says that if by deviating from the optimal solution $(u_1^*, u_2^*)$, player 2 could gain $p$ percentage more in terms of utility, then player 1 will have to lose a percentage at least as large as $p$.

The Nash bargaining solution is equivalent to the PF solution according to the following proposition:

**Proposition 3.1** (**equivalence**, e.g. Kelly 1997; Boche & Schubert 2009). *For any convex set $\mathcal{U} \in \mathbb{R}_{++}^n$, a point $u \in \mathcal{U}$ is the Nash bargaining solution iff it is proportionally fair. If $\mathcal{U}$ is non-convex, then a PF solution, when exists, is a Nash bargaining solution.*

A PF solution, whenever exists, is a Nash bargaining solution over $\mathcal{U}$. While the converse also holds if $\mathcal{U}$ is convex, for nonconvex $\mathcal{U}$, PF solutions may not exist. In contrast, NBS always exists if $\mathcal{U}$ is compact, and

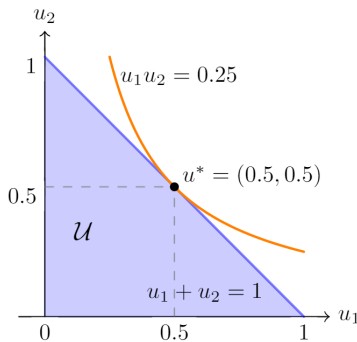

Figure 1: Figure inspired by Nash (1950). $\mathcal{U}$: the feasible set of utilities. Blue line: maximizers of the total utility, on which the Nash bargaining solution $\boldsymbol{u}^*$ stands out as the fairest.

thus we solve eq. 3.1 as a necessary condition of PF. From Jensen's inequality, we can show that:

$$\sum_i p_i \log u_i \leq \log \sum_i p_i u_i. \tag{3.5}$$

In other words, solving the NBS yields a lower bound of the averaged utility. On the other hand, if any of the utilities is close to zero, then the left hand side of eq. 3.5 would decrease to $-\infty$. Therefore, the NBS does not yield extremely undesirable performance for any client. In a nutshell, the NBS achieves a balance between maximizing the average and the worst-case utilities.

### 3.1 The PropFair algorithm for federated learning

In order to realize proportional fairness in FL, we need to solve eq. 3.1. With parametrization of $u_i$, the utility set $\mathcal{U}$ becomes the set of all possible choices of $(u_1(\boldsymbol{\theta}), \ldots, u_n(\boldsymbol{\theta}))$, and our goal is to find a global model $\boldsymbol{\theta}$ to solve eq. 3.1:

$$\max_{\boldsymbol{\theta}} \sum_i p_i \log u_i(\boldsymbol{\theta}). \tag{3.6}$$

#### 3.1.1 What is the right choice of utilities?

One immediate question is: *how do we define these utilities in FL?* Ideally, the utility should be the test accuracy, which is unfortunately not amenable to optimize. Instead, we could use the training loss $f_i$. There are a few alternatives:

- Replace $u_i$ with $f_i$ as done in $q$-FFL, and minimize the aggregate loss, $\sum_i p_i \log f_i$;
- Replace $u_i$ with $f_i$ as done in $q$-FFL, and maximize the aggregate utility, $\sum_i p_i \log f_i$;
- Choose $u_i = M - f_i$, and maximize $\sum_i p_i \log(M - f_i)$, with $M$ some hyperparameter to be determined.

The first approach will encourage the client losses to be even more disparate. For instance, suppose $p_1 = p_2 = \frac{1}{2}$, and then $(f_1, f_2) = (\frac{1}{3}, \frac{2}{3})$ has smaller product than $(f_1, f_2) = (\frac{1}{2}, \frac{1}{2})$. The second approach is not a choice either as it is at odds with minimizing client losses. Therefore, we are left with the third option. By contrast, for any $M \geq 1$ and $p_1 = p_2 = 1/2$, one can show that $(f_1, f_2) = (\frac{1}{2}, \frac{1}{2})$ always gives a better solution than $(f_1, f_2) = (\frac{1}{3}, \frac{2}{3})$. The resulting objective becomes:

$$\min_{\boldsymbol{\theta}} \pi(\boldsymbol{\theta}) := -\sum_i p_i \log(M - f_i(\boldsymbol{\theta})). \tag{3.7}$$

#### 3.1.2 Huberization

However, the objective eq. 3.7 also raises issues: what if $M - f_i$ is small and blows up the gradient, or even worse, what if $M - f_i$ is negative and the logarithm does not make sense at all? Inspired by Huber's

---

**Algorithm 1:** PropFair

---

**1** **Input:** global epoch $T$, client number $n$, loss function $f_i$ for client $i$, number of samples $n_i$ for client $i$, initial global model $\boldsymbol{\theta}_0$, local step number $K_i$, baseline $M$, threshold $\epsilon$, $p_i = n_i/N$, batch size $m$, learning rate $\eta$

**2** **for** $t$ *in* $0, 1 \ldots T - 1$ **do**

**3** $\quad$ randomly select $\mathcal{C}_t \subseteq [n]$

**4** $\quad$ $\boldsymbol{\theta}_{t,0}^{(i)} = \boldsymbol{\theta}_t$ for $i \in \mathcal{C}_t$, $N = \sum_{i \in \mathcal{C}_t} n_i$

**5** $\quad$ **for** $i$ *in* $\mathcal{C}_t$ **do** // in parallel

**6** $\quad\quad$ $j = 1$, draw $K_i$ mini-batches of samples from client $i$

**7** $\quad\quad$ **for** $S_i$ *in the* $K_i$ *batches* **do**

**8** $\quad\quad\quad$ $\ell_{S_i}(\boldsymbol{\theta}) = \frac{1}{|S_i|} \sum_{(\boldsymbol{x},y) \in S_i} \ell(\boldsymbol{\theta}, (\boldsymbol{x}, y))$

**9** $\quad\quad\quad$ $f_i^{\log}(\boldsymbol{\theta}) = -\log_{[\epsilon]}(M - \ell_{S_i}(\boldsymbol{\theta}))$

**10** $\quad\quad\quad$ $\boldsymbol{\theta}_{t,j}^{(i)} \leftarrow \boldsymbol{\theta}_{t,j-1}^{(i)} - \eta \nabla f_i^{\log}(\boldsymbol{\theta}_{t,j-1}^{(i)})$, $j \leftarrow j + 1$

**11** $\quad$ $\boldsymbol{\theta}_{t+1} = \sum_{i \in S_t} p_i \boldsymbol{\theta}_{t,K_i}^{(i)}$

**12** **Output:** global model $\boldsymbol{\theta}_T$

---

approach of robust estimation (Huber, 1964), we propose a "huberized" version of eq. 3.7:

$$\min_{\boldsymbol{\theta}} -\sum_i p_i \log_{[\epsilon]}(M - f_i(\boldsymbol{\theta})), \text{ with } \log_{[\epsilon]}(M - t) := \begin{cases} \log(M - t), & \text{if } t \le M - \epsilon, \\ \log \epsilon - \frac{1}{\epsilon}(t - M + \epsilon), & \text{if } t > M - \epsilon. \end{cases} \tag{3.8}$$

Essentially, $\log_{[\epsilon]}(M - t)$ is a robust $\mathcal{C}^1$ extension of $\log(M - t)$ from $[0, M - \epsilon]$ to $\mathbb{R}_+$: its linear part ensures that at $t = M - \epsilon$, both the value and the derivative are continuous. If any $f_i$ is close or greater than $M$, then eq. 3.8 switches from logarithm to its linear version. Based on eq. 3.8 we propose Algorithm 1 called *PropFair*. It modifies FedAvg (McMahan et al., 2017) with a simple drop-in replacement, by replacing the loss of each batch $\ell_S^i$ with $\log_{[\epsilon]}(M - \ell_S^i(\boldsymbol{\theta}))$. This allows easy adaptation of PropFair with minimal change into any of the current FL platforms, such as Flower (Beutel et al., 2020) and Tensorflow Federated.[1] Also note that in Algorithm 1 we average over the batch before the composition with $\log_{[\epsilon]}$. This order cannot be switched since otherwise the local variance will be $m$ times larger (see eq. B.52).

**Remark.** When $M \to \infty$ and $f_i(\boldsymbol{\theta})$ is small compared to $M$, the loss function for client $i$ becomes:

$$f_i^{\log}(\boldsymbol{\theta}) = -\log(M - f_i(\boldsymbol{\theta})) \approx -\log M + \frac{f_i(\boldsymbol{\theta})}{M}.$$

Thus, FedAvg can be regarded as a first-order approximation of PropFair. We utilize this approximation in our implementation. Another way to obtain FedAvg is to take $\epsilon = M$ and thus $\log_{[\epsilon]}(M - t)$ always uses the linear branch. In contrast, if $\epsilon \to 0$, then eq. 3.8 becomes more similar to eq. 3.7.

### 3.2 Dual view of PropFair

With the dual view from Section 2.4, we can also treat PropFair as minimizing a weighted combination of loss functions (plus constants), similar to other fair FL algorithms. Note that if $\varphi(f_i) = -\log(M - f_i)$ in eq. 2.9, then we have PropFair (see Table 1):

**Proposition 3.2 (dual view of PropFair).** *The generalized mean eq. 2.9 for PropFair can be written as:*

$$\mathsf{A}_\varphi(\boldsymbol{f}) = \max_{\boldsymbol{\lambda} \ge \boldsymbol{0}, \prod_i (\lambda_i/p_i)^{p_i} \ge 1} \boldsymbol{\lambda}^\top \boldsymbol{f} - M(\boldsymbol{\lambda}^\top \boldsymbol{1} - 1), \tag{3.9}$$

*Solving the inner maximization of eq. 2.10 gives $\prod_{i=1}^n \left(\frac{\lambda_i}{p_i}\right)^{p_i} = 1$ and $\lambda_i \propto \frac{p_i}{M - f_i}$.*

Similar to TERM/$q$-FFL, PropFair puts a larger weight on worse-off clients with a larger loss.

---

[1]https://www.tensorflow.org/federated

# 4 The optimization side of PropFair

In this section, we discuss the convexity of our PropFair objective and show the convergence guarantee of Algorithm 1. This gives formal fairness guarantee for our algorithm, and potentially for the convergence of many others in the scalarization class, eq. 2.2. For simplicity we only study the case when $f_i \leq M - \epsilon$ for all $i$.

## 4.1 Convexity of the PropFair objective

Convexity is an important and desirable property in optimization (Boyd & Vandenberghe, 2004). With convexity, every stationary point is optimal (Bertsekas, 1997). From the composition rule, if $f_i$ is convex for each client $i$, then $M - f_i$ is concave, and thus $\sum_i p_i \log(M - f_i(\boldsymbol{\theta}))$ is concave as well (e.g., Boyd & Vandenberghe, 2004). In other words, for convex losses, our optimization problem eq. 3.7 is still convex as we are maximizing over concave functions. Moreover, our PropFair objective is convex *even* when $f_i$'s are not. For example, this could happen if $f_i(\boldsymbol{\theta}) = M - \exp(\boldsymbol{\theta}^\top \mathbf{A}_i \boldsymbol{\theta})$ and each $\mathbf{A}_i$ is a positive definite matrix. In fact, it suffices to require each $M - f_i$ to be log-concave (Boyd & Vandenberghe, 2004).

## 4.2 Adaptive learning rate and curvature

Denote $\varphi(t) = -\log(M - t)$. We can compute the 1$^{\text{st}}$- and 2$^{\text{nd}}$-order derivatives of $\varphi \circ f_i$:

$$\nabla(\varphi \circ f_i) = \frac{\nabla f_i}{M - f_i}, \ \nabla^2(\varphi \circ f_i) = \frac{(M - f_i)\nabla^2 f_i + (\nabla f_i)(\nabla f_i)^\top}{(M - f_i)^2}. \tag{4.1}$$

This equation tells us that at each local gradient step, the gradient $\nabla(\varphi \circ f_i)$ has the same direction as $\nabla f_i$, and the only difference is the step size. Compared to FedAvg, PropFair automatically has an *adaptive learning rate* for each client. When the local client loss function $f_i$ is small, the learning rate is smaller; when $f_i$ is large, the learning rate is larger. This agrees with our intuition that to achieve fairness, a worse-off client should be allowed to take a more aggressive step, while a better-off client moves more slowly to "wait for" other clients.

In the Hessian $\nabla^2(\varphi \circ f_i)$, an additional positive semi-definite (p.s.d.) term $(\nabla f_i)(\nabla f_i)^\top$ is added. Thus, $\nabla^2(\varphi \circ f_i)$ can be p.s.d. even if the original Hessian $\nabla^2 f_i$ is not. Moreover, the denominator $(M - f_i)^2$ has a similar effect of coordinating the curvatures of various clients as in the gradients.

## 4.3 Convergence results

Let us now formally prove the convergence of PropFair by bounding its progress, using standard assumptions (Li et al., 2019; Reddi et al., 2020) such as Lipschitz smoothness and bounded variance. Every norm discussed in this subsection is Euclidean (including the proofs in Appendix B).

In fact, PropFair can be treated as an easy variant of FedAvg, with the local objective $f_i$ replaced with $f_i^{\log}$. Therefore, we just need to prove the convergence of FedAvg and the convergence of PropFair would follow similarly. In general, similar results also hold for objectives in the form of eq. 2.2.

Let us state the assumptions first. Since in practice we use stochastic gradient descent (SGD) for training, we consider the effect of mini-batches. We also assume that the (local) variance of mini-batches and the (global) variance among clients are bounded.

**Assumption 4.1** (**Lipschitz smoothness and bounded variances**). *Each function $f_i$ is L-Lipschitz smooth, i.e., for any $\boldsymbol{\theta}, \boldsymbol{\theta}' \in \mathbb{R}^d$ and any $i \in [n]$, we have $\|\nabla f_i(\boldsymbol{\theta}) - \nabla f_i(\boldsymbol{\theta}')\| \leq L\|\boldsymbol{\theta} - \boldsymbol{\theta}'\|$; For any $i, j \in [n]$, $f_i - f_j$ is $\sigma$-Lipschitz continuous and $\mathbb{E}_{(\boldsymbol{x},y) \sim \mathcal{D}_i} \|\nabla \ell(\boldsymbol{\theta}, (\boldsymbol{x}, y)) - \nabla f_i(\boldsymbol{\theta})\|^2 \leq \sigma_i^2 \ \forall \boldsymbol{\theta} \in \mathbb{R}^d$.*

Following the notations of Reddi et al. (2020), we use $\sigma^2$ and $\sigma_i^2$ to denote the global and local variances for client $i$. This assumption allows us to obtain the convergence result for FedAvg (see Algorithm 2). For easy reference, we include FedAvg (McMahan et al., 2017) in Algorithm 2, whose goal is to optimize the overall performance. At each round, each client takes local SGD steps to minimize the loss function based on the client data. Afterwards, the server computes a weighted average of the parameters of these participating clients, and shares this average among them. Note that for client $i$, the number of local steps is $K_i$ with

---

**Algorithm 2:** FedAvg

---

**1** **Input:** global epoch $T$, client number $n$, loss function $f_i$, number of samples $n_i$ for client $i$, initial global model $\boldsymbol{\theta}_0$, local step number $K_i$ for client $i$, batch size $m$, learning rate $\eta$, $p_i = n_i/N$

**2** **for** $t$ *in* $0, 1 \ldots T-1$ **do**

**3** $\quad$ randomly select $\mathcal{C}_t \subseteq [n]$

**4** $\quad$ $\boldsymbol{\theta}_t^{(i)} = \boldsymbol{\theta}_t$ for $i \in \mathcal{C}_t$, $N = \sum_{i \in \mathcal{C}_t} n_i$

**5** $\quad$ **for** $i$ *in* $\mathcal{C}_t$ **do** // in parallel

**6** $\quad\quad$ starting from $\boldsymbol{\theta}_t^{(i)}$, take $K_i$ local SGD steps on $f_i$ to find $\boldsymbol{\theta}_{t+1}^{(i)}$

**7** $\quad$ $\boldsymbol{\theta}_{t+1} = \sum_{i \in \mathcal{C}_t} p_i \boldsymbol{\theta}_{t+1}^{(i)}$

**8** **Output:** global model $\boldsymbol{\theta}_T$

---

learning rate $\eta$. In line 3 of Algorithm 2, if $\mathcal{C}_t = [n]$ then we call it *full participation*, otherwise it is called *partial participation*. We prove the following convergence result of FedAvg. Note that we defined $F$ in eq. 2.3, and $m$ is the batch size.

**Theorem 4.2** (**FedAvg**). *Given Assumption 4.1, assume that the local learning rate satisfies $\eta K_i \leq \frac{1}{6L}$ for any $i \in [n]$ and*

$$\eta \leq \frac{1}{L} \sqrt{\frac{1}{24(e-2)(\sum_i p_i^2)(\sum_i K_i^4)}}. \tag{4.2}$$

*Running Algorithm 2 for $T$ global epochs we have:*

$$\min_{0 \leq t \leq T-1} \mathbb{E} \|\nabla F(\boldsymbol{\theta}_t)\|^2 \leq \frac{12}{(11\mu - 9)\eta} \left( \frac{F_0 - F^*}{T} + \Psi_\sigma \right),$$

*with $\mu = \sum_i p_i K_i$ for full participation and $\mu = \min_i K_i$ for partial participation, $F_0 = F(\boldsymbol{\theta}_0)$, $F^* = \min_{\boldsymbol{\theta}} F(\boldsymbol{\theta})$ the optimal value, and*

$$\Psi_\sigma = \eta \|\boldsymbol{p}\|^2 \left[ \sum_{i=1}^n K_i^2 \left( \frac{L\eta\sigma_i^2}{2m} + \sigma^2 \right) + (e-2)\eta^2 L^2 \sum_{i=1}^n K_i^3 \left( \frac{\sigma_i^2}{m} + 6K_i\sigma^2 \right) \right], \quad \boldsymbol{p} = (p_1, \ldots, p_n).$$

Our result is quite general: we allow for both full and partial participation; multiple and heterogeneous local steps; and non-uniform aggregation with weight $p_i$. The variance term $\Psi_\sigma$ decreases with smaller local steps $K_i$, which agrees with our intuition that each $K_i$ should be as small as possible given the communication constraint. Moreover, to minimize $\|\boldsymbol{p}\|^2$ we should take $p_i = 1/n$ for each client $i$, which means if the samples are more evenly distributed across clients, the error is smaller. In presence of convexity, we can see that FedAvg converges to a neighborhood of the optimal solution, and the size of the neighborhood is controlled by the heterogeneity of clients and the variance of mini-batches. When we have the global variance term $\sigma = 0$, our result reduces to the standard result of stochastic gradient descent (e.g., Ghadimi & Lan 2013), since we have

$$\min_{0 \leq t \leq T-1} \mathbb{E} \|\nabla F(\boldsymbol{\theta}_t)\|^2 \leq \frac{12}{(11\mu - 9)\eta} \frac{F_0 - F^*}{T} + O(\eta),$$

and by taking $\eta = O(1/\sqrt{T})$, we obtain $\min_{0 \leq t \leq T-1} \mathbb{E} \|\nabla F(\boldsymbol{\theta}_t)\|^2 = O(1/\sqrt{T})$.

We note that we are not the first to prove the convergence of FedAvg. For instance, Li et al. (2019) assumes that each function $f_i$ is strongly convex and each client takes the same number of local steps; Karimireddy et al. (2020) assumes the same number of local steps, gradient bounded similarity and uniform weights $p_i = 1/n$. These assumptions may not reflect the practical use of FedAvg. For example, usually each client has a different number of samples and they may take different numbers of local updates. Moreover, for neural networks, (global) strong convexity is usually not present. Compared to these results, we consider

different local client steps, heterogeneous weights and partial participation in the non-convex case, which is more realistic.

Based on Theorem 4.2, we can similarly prove the convergence of other FL algorithms which minimize eq. 2.2, if there are some additional assumptions. For the PropFair algorithm, as an example, we need to additionally assume the Lipschitzness and bounded variances for the client losses:

**Assumption 4.3** (**boundedness, Lipschitz continuity and bounded variances for client losses**). *For any $i \in [n]$, $\boldsymbol{\theta} \in \mathbb{R}^d$ and any batch $S_i \sim \mathcal{D}_i^m$ of $m$ i.i.d. samples, we have:*

$$0 \leq \ell_{S_i}(\boldsymbol{\theta}) := \frac{1}{|S_i|} \sum\nolimits_{(\boldsymbol{x},y)\in S_i} \ell(\boldsymbol{\theta},(\boldsymbol{x},y)) \leq \frac{M}{2},$$

*and for any $\boldsymbol{\theta}, \boldsymbol{\theta}' \in \mathbb{R}^d$, $\|f_i(\boldsymbol{\theta}) - f_i(\boldsymbol{\theta}')\| \leq L_0\|\boldsymbol{\theta} - \boldsymbol{\theta}'\|$ holds. We also assume that for any $i,j \in [n]$ and $\boldsymbol{\theta} \in \mathbb{R}^d$, $\|f_i(\boldsymbol{\theta}) - f_j(\boldsymbol{\theta})\|^2 \leq \sigma_0^2$ and $\mathbb{E}_{(\boldsymbol{x},y)\sim\mathcal{D}_i}\|\ell(\boldsymbol{\theta},(\boldsymbol{x},y)) - f_i(\boldsymbol{\theta})\|^2 \leq \sigma_{0,i}^2$ hold.*

we can obtain the convergence guarantee of PropFair to a neighborhood of some stationary point:

**Theorem 4.4** (**PropFair**). *Denote $\widetilde{L} = \frac{4}{M^2}(\frac{3}{2}ML + L_0^2)$ and $p_i = \frac{n_i}{N}$. Given Assumptions 4.1 and 4.3, assume that the local learning rate satisfies:*

$$\eta \leq \min\left\{\min_{i\in[n]} \frac{1}{6\widetilde{L}K_i}, \frac{1}{8\widetilde{L}}\sqrt{\frac{1}{(e-2)(\sum_i p_i^2)(\sum_i K_i^4)}}\right\}. \tag{4.3}$$

*By running Algorithm 1 for $T$ global epochs we have:*

$$\min_{0\leq t\leq T-1} \mathbb{E}\|\nabla\pi(\boldsymbol{\theta}_t)\|^2 \leq \frac{12}{(11\mu-9)\eta}\left(\frac{\pi_0-\pi^*}{T} + \widetilde{\Psi}_\sigma\right),$$

*with $\mu = \sum_i p_i K_i$ for full participation and $\mu = \min_i K_i$ for partial participation, $\pi_0 = \pi(\boldsymbol{\theta}_0)$, $\pi^* = \min_{\boldsymbol{\theta}} \pi(\boldsymbol{\theta})$ the optimal value, and*

$$\widetilde{\Psi}_\sigma = \eta\|\boldsymbol{p}\|^2\left[\sum_{i=1}^n K_i^2\left(\frac{\widetilde{\sigma}_i^2}{m} + 2\widetilde{\sigma}^2\right) + 16(e-2)\eta^2\tilde{L}^2\sum_{i=1}^n K_i^4\left(\frac{\widetilde{\sigma}_i^2}{m} + \widetilde{\sigma}^2\right)\right]$$

*where $\widetilde{\sigma}_i^2 = \frac{8}{M^4}(9M^2\sigma_i^2 + 4L_0^2\sigma_{0,i}^2)$ and $\widetilde{\sigma} = \frac{4}{M}\left(\frac{3}{2}\sigma + \frac{L_0}{M}\sigma_0\right)$.*

Our Theorem 4.4 inherits similar advantages from Theorem 4.2. One major difference is that when $\tilde{\sigma} = 0$, one cannot retrieve the same rate of SGD. This is expected since each batch $\varphi \circ \ell_{S_i}$ is no longer an unbiased estimator $\varphi \circ f_i$ due to the composition. Nevertheless, due to data heterogeneity in FL, the global variance $\tilde{\sigma}$ is often large, in which case the local variance term $\tilde{\sigma}_i^2/m$ in $\widetilde{\Psi}_\sigma$ can be comparable to $\tilde{\sigma}^2$ by controlling the batch size $m$.

## 5 Experiments

In this section, we verify properties of PropFair by answering the following questions: (1) can PropFair achieve proportional fairness as in eq. 3.3? (2) what balance does PropFair achieve between the average and worst-case performances? We report them separately in Section 5.2 and Section 5.3.

### 5.1 Experimental setup

We first give details on our datasets, models and hyperparameters, which are in accordance with existing works. See Appendix D for additional experimental setup. A comprehensive survey of benchmarking FL algorithms can be found in e.g. Caldas et al. (2018); He et al. (2020).

**Datasets.** We follow standard benchmark datasets as in the existing literature, including CIFAR-{10, 100} (Krizhevsky et al., 2009), TinyImageNet (Le & Yang, 2015) and Shakespeare (McMahan et al., 2017).

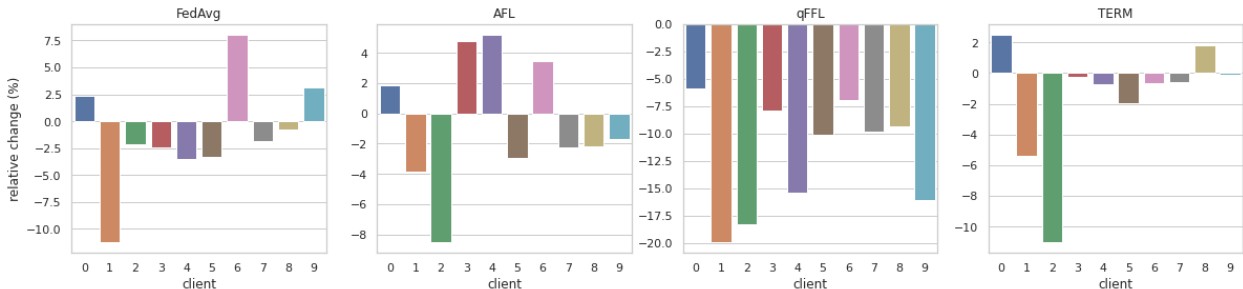

Figure 2: The relative improvement/deterioration $(u_i - u_i^*)/u_i^*$ of the test accuracy $u_i$ of each client $i$ of other baseline algorithms compared to PropFair. The dataset is CIFAR-10. The (weighted) average of the relative changes for FedAvg, AFL, $q$-FFL and TERM are respectively: $-2.21\%$, $-1.32\%$, $-12.95\%$, $-1.79\%$. We choose the hyperparameters based on Table 4 in Appendix D.

For vision datasets (CIFAR-{10, 100}/TinyImageNet), the task is image classification, and following Wang et al. (2019b) we use Dirichlet allocation to split the dataset into different clients. For the language dataset (Shakespeare), the task is next-character prediction. We use the default realistic partition based on different users. We first partition the dataset into different clients, and further split each client dataset into its own training and test sets. This reflects the real scenario, where each client evaluates the performance by itself.

**Models, optimizer and loss function.** For vision datasets we use ResNet-18 (He et al., 2016) with Group Normalization (Wu & He, 2018). As discussed by Hsieh et al. (2020), Group Normalization (with `num_groups=2`) works better than batch normalization, especially in the federated settings. For the Shakespeare dataset, we use LSTM (Hochreiter & Schmidhuber, 1997). We find the best learning rates through grid search (see Appendix D).

**Other hyperparameters.** We implement full participation and one local epoch throughout (with many local steps for each client). Due to data heterogeneity, the number of local steps $K_i$ for each client $i$ varies. For CIFAR-{10, 100} we partition the data into 10 clients; for TinyImageNet/Shakespeare we choose 20 clients.

**Evaluation metrics.** We validate proportional fairness eq. 3.3 of our PropFair algorithm, where we treat each $u_i$ as the test accuracy of client $i$. To show that PropFair achieves a proper balance between utilitarian and egalitarian fairness, we use the average and the worst 10% test accuracies. These are standard fairness metrics used in the literature (e.g. Li et al., 2020a;c). In Appendix D we also present other standard metrics such as standard deviation and worst 20%.

## 5.2 Verification of proportional fairness

In this subsection, we show that PropFair can, to some extent, achieve proportional fairness as defined in eq. 3.3. We treat $u_i$ as the test accuracy of client $i$, and compute

$$\sum_i p_i \frac{u_i - u_i^*}{u_i^*}, \tag{5.1}$$

with $p_i = n_i/N$ and $\boldsymbol{u}^* := (u_1^*, \ldots, u_n^*)$ the test accuracies obtained by the PropFair model. Although we cannot verify eq. 5.1 for every $\boldsymbol{u}$, we can at least validate the negativity for some competitive $\boldsymbol{u}$'s, of, e.g., models learned by other fair FL algorithms.

### 5.2.1 CIFAR-10

We first compute eq. 5.1 where $\boldsymbol{u}^*$ is the test accuracies obtained by PropFair and $\boldsymbol{u}$ is the test accuracies found by one of the other fair FL algorithms, including FedAvg, AFL, $q$-FFL and TERM. Figure 2 shows the relative changes of each client, $(u_i - u_i^*)/u_i^*$, from which we can see that compared to the solution found by PropFair, for another fair FL solution, most clients are degraded by a large relative amount, and only a few clients are improved by a small amount.

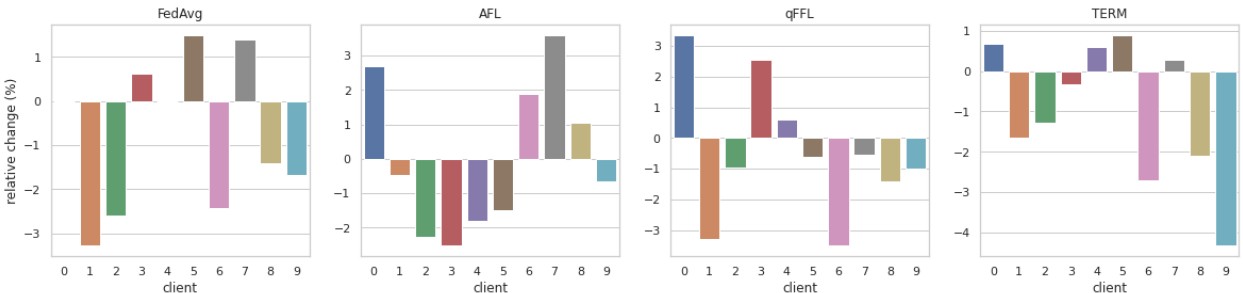

Figure 3: The relative improvement/deterioration $(u_i - u_i^*)/u_i^*$ of the test accuracy of each client $i$, pretrained with PropFair and fine-tuned with another baseline. The dataset is CIFAR-100. The average of the relative changes for FedAvg, AFL, $q$-FFL and TERM are respectively: $-0.86\%, +0.05\%, -0.67\%, -1.01\%$.

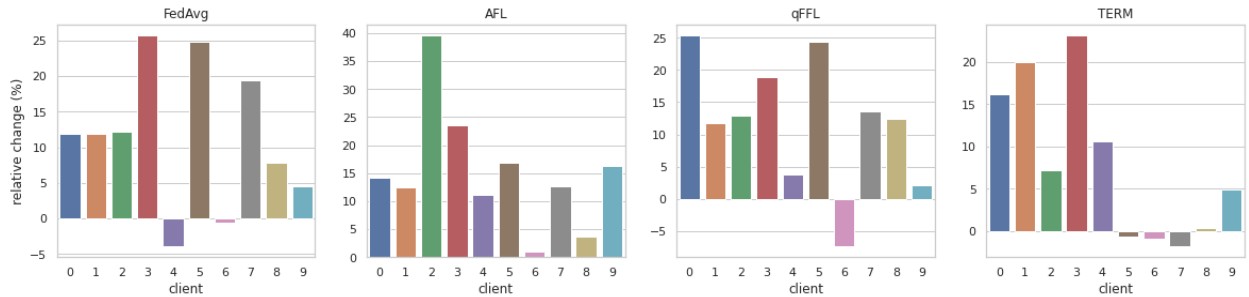

Figure 4: The relative improvement/deterioration $(u_i - u_i^*)/u_i^*$ of test accuracy of each client $i$, pretrained with another baseline and fine-tuned with PropFair. The dataset is CIFAR-100. The average of the relative changes over FedAvg, AFL, $q$-FFL and TERM are respectively: $10.88\%, 13.93\%, 11.58\%, 8.67\%$.

### 5.2.2 CIFAR-100

In fact, we may compute eq. 5.1 with a stronger $\boldsymbol{u}$. For CIFAR-100, we still treat $\boldsymbol{u}^*$ as the test accuracies obtained by PropFair. The difference is that we use $\boldsymbol{u}^*$ as the initialization, and fine-tune with other fair FL algorithms, to find $\boldsymbol{u}$. If other fair FL algorithms cannot improve the proportional fairness of $\boldsymbol{u}^*$, then eq. 5.1 should be negative. As we see in Figure 3, this result indeed holds approximately (except the slight improvement for AFL).

By contrast, none of the baseline fair FL algorithms can achieve the same level of proportional fairness as our PropFair. In Figure 4, we see that if we start from a model pretrained with a baseline fair FL algorithm, and fine-tune with our Propfair, most client performances are improved, sometimes by a large margin.

### 5.3 Comparison between PropFair and existing fair FL algorithms on other metrics

In Figure 5, we compare PropFair with existing fair FL algorithms using the average and the worst 10% test accuracies across clients, including FedAvg (McMahan et al., 2017), $q$-FFL (Li et al., 2020c), AFL (Mohri et al., 2019) and TERM (Li et al., 2020a).

**Average performance.** From Figure 5 we can see that PropFair does not always yield the best average performance, e.g., compared to $q$-FFL on TinyImageNet. This is expected, since maximizing the Nash product does not necessarily give the best average performance. Nevertheless, PropFair remains competitive. Somewhat surprisingly, FedAvg does not always achieve the best average performance, which might be due to optimization issues (Pathak & Wainwright, 2020).

**Worst 10% performance.** We also compare the worst 10% performance of various fair FL algorithms. We observe that PropFair achieves the state-of-the-art in terms of the worst 10% performance, across various

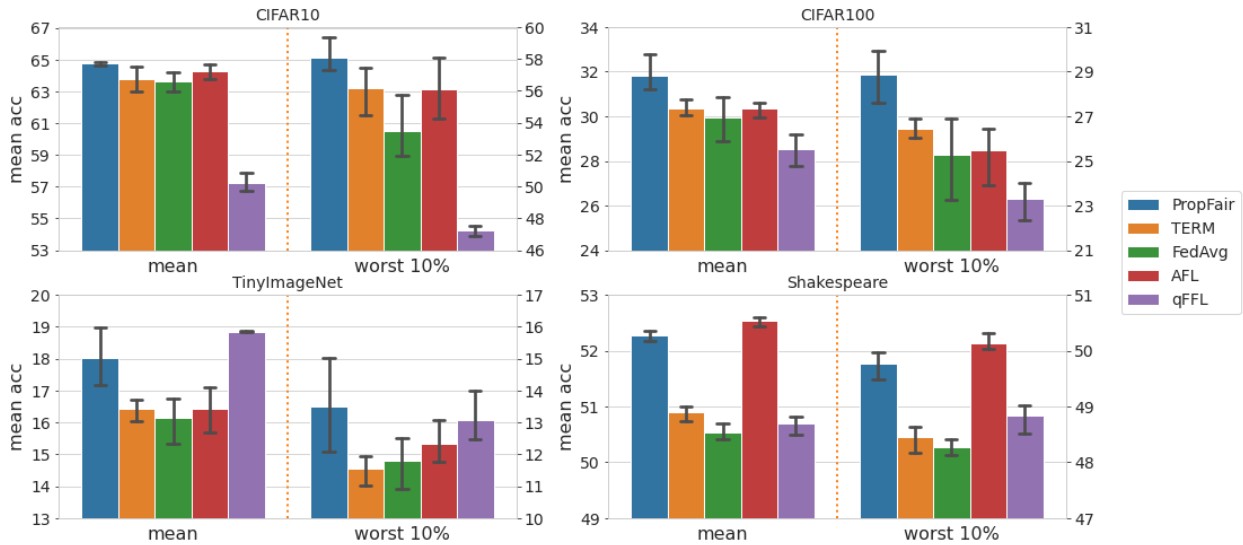

Figure 5: Mean and worst 10% test accuracies for different algorithms. The accuracies are in percentage. (**top left**): CIFAR-10; (**top right**): CIFAR-100; (**bottom left**): TinyImageNet; (**bottom right**): Shakespeare. All subfigures share the same legends and axis labels.

vision and language datasets. This is within our expectation, since from eq. 3.5 and eq. 3.7 we can see that low utility in *any* of the clients would result in a small Nash product.

Specifically, although AFL directly maximizes the worst-case loss function, it does not always achieve the best worst-case performances (see Table 6 in Appendix D), especially for vision datasets. This might be due to the generalization issue of AFL (see Appendix C).

# 6  Related Work in Fair FL

We review recent related work for fair federated learning. In additional to AFL (Mohri et al., 2019), $q$-FFL (Li et al., 2020c) and TERM (Li et al., 2020a), there have been other approaches for fairness in FL. For example, FedMGDA+ (Hu et al., 2022) defines fairness as achieving the Pareto frontier and they proposed to use the MGDA algorithm. As another example, GIFAIR-FL (Yue et al., 2022) encourages the similarity of the client losses by adding a regularization term of the pairwise $\ell_1$ distances. Last but not least, Ditto (Li et al., 2021) proposed a personalization approach to obtain fairness and robustness. A comprehensive recent survey of fairness in FL can be found in Shi et al. (2021), and we have included additional papers of fairness (in FL and in general) in Appendix F.

# 7  Conclusions

Based on the necessity of considering relative changes, we introduce the concept of Proportional Fairness (PF) into the field of federated learning (FL), which is deeply rooted in cooperative game theory. By showing the connection between PF and the Nash bargaining solution, we propose PropFair that maximizes the product of client utilities, where the total relative utility cannot be improved. This guarantees PropFair to have good worst-case performance without sacrificing the total utility much. We verify proportional fairness and the balance between utilitarian and egalitarian fairness in our extensive experiments. As we have shown, many fair FL algorithms, including PropFair, can be unified using Kolmogorov's generalized mean, the deeper understanding of which may lead to future design of fair FL algorithms.

## Broader Impact Statement

With the wide deployment of federated learning, how to ensure fairness in FL algorithms has become a major concern. In this work, we study proportional fairness in FL to make FL systems fairer and thus more trustworthy. This could have important positive social impacts as well. We are not aware of potential negative societal impacts yet but we welcome discussions on them.

### Acknowledgments

We thank the reviewers and the action editor for constructive comments that largely improved our draft. GZ would like to thank Changjian Shui for his constructive feedback on an earlier draft, and Mahdi Beitollahi for pointing out typos. YY is supported by NSERC and WHJIL.

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

| Notation | Meaning |
|---|---|
| $\boldsymbol{\theta}$ | model parameters |
| $n$ | the number of clients |
| $m$ | batch size |
| $\boldsymbol{x} \in \mathbb{R}^d$ | raw input |
| $y \in [C]$ | output label |
| $n_i$ | the number of samples from client $i$ |
| $N = \sum_i n_i$ | the total number of samples from all clients |
| $S_i$ | a batch of samples from client $i$ |
| $\mathcal{D}_i$ | data distribution of client $i$ |
| $\ell(\boldsymbol{\theta}, (\boldsymbol{x}, y))$ | prediction loss (e.g. cross entropy) of model $\boldsymbol{\theta}$ on sample $(\boldsymbol{x}, y)$ |
| $\ell_{S_i}$ | average loss over batch $S_i$ |
| $f_i$ | expected loss over distribution $\mathcal{D}_i$ |
| $\boldsymbol{f} = (f_1, \ldots, f_n)$ | vector of client losses |
| $u_i$ | utility of client $i$ |
| $\boldsymbol{u} = (u_1, \ldots, u_n)$ | vector of client utilities |
| $K_i$ | the number of local steps of client $i$ |
| $p_i$ | pre-defined weight of each client $i$, usually $p_i = n_i/N$ |
| $\boldsymbol{p} = (p_1, \ldots, p_n)$ | vector of client weights |
| $\Phi : \mathbb{R}^n \to \mathbb{R}$ | scalarization function of $\boldsymbol{f}$ |
| $\varphi : \mathbb{R} \to \mathbb{R}$ | a scalar function that acts on each client loss $f_i$ |
| $\mathsf{A}_\varphi$ | Kolmogorov's generalized mean with scalar function $\varphi$ |
| $\boldsymbol{\lambda} = (\lambda_1, \ldots, \lambda_n)$ | dual parameter |
| $\eta$ | local learning rate |
| $\sigma_i^2$ | local variance on distribution $i$ |
| $\sigma^2$ | global variance among clients |
| $F = \sum_i p_i f_i$ | objective of FedAvg |
| $\log_{[\epsilon]}(M - t)$ | huberization of $\log(M - t)$, see eq. 3.8 |
| $\pi = -\sum_i p_i \log_{[\epsilon]}(M - f_i)$ | objective of PropFair |
| $L$ | Lipschitz constant of all $\nabla f_i$'s |
| $e$ | natural logarithm |
| $\mathbb{R}_{++}^n$ | (strictly) positive orthant of $\mathbb{R}^n$ |

Table 2: Notation table.

## A  Notations

We include a notation table for easy navigation. The reader can refer to Table 2 for quick access to the notations.

## B Proofs

**Proposition 3.1** (**equivalence**, e.g. Kelly 1997; Boche & Schubert 2009). *For any convex set $\mathcal{U} \in \mathbb{R}^n_{++}$, a point $u \in \mathcal{U}$ is the Nash bargaining solution iff it is proportionally fair. If $\mathcal{U}$ is non-convex, then a PF solution, when exists, is a Nash bargaining solution.*

*Proof.* The Nash bargaining solution $\boldsymbol{u}^*$ is equivalent to the maximum of the following:

$$\max_{\boldsymbol{u} \in \mathcal{U}} \sum_{i=1}^{n} p_i \log u_i. \tag{B.1}$$

Since $\mathcal{U}$ is convex and $\sum_{i=1}^{n} p_i \log u_i$ is concave in $\boldsymbol{u}$, the necessary and sufficient optimality condition (e.g,. Bertsekas, 1997) is:

$$\langle \boldsymbol{u} - \boldsymbol{u}^*, \nabla \sum_{i=1}^{n} p_i \log u_i^* \rangle \leq 0, \text{ for any } \boldsymbol{u} \in \mathcal{U}, \tag{B.2}$$

or equivalently, eq. 3.3. If $\mathcal{U}$ is non-convex, then the optimality condition eq. B.2 also holds for the convex hull of $\mathcal{U}$. Therefore, $\boldsymbol{u}^*$ is a maximizer of $\sum_{i=1}^{n} p_i \log u_i$ in the convex hull of $\mathcal{U}$ and thus $\mathcal{U}$. □

**Theorem 4.2** (**FedAvg**). *Given Assumption 4.1, assume that the local learning rate satisfies $\eta K_i \leq \frac{1}{6L}$ for any $i \in [n]$ and*

$$\eta \leq \frac{1}{L} \sqrt{\frac{1}{24(e-2)(\sum_i p_i^2)(\sum_i K_i^4)}}. \tag{4.2}$$

*Running Algorithm 2 for $T$ global epochs we have:*

$$\min_{0 \leq t \leq T-1} \mathbb{E}\|\nabla F(\boldsymbol{\theta}_t)\|^2 \leq \frac{12}{(11\mu - 9)\eta} \left( \frac{F_0 - F^*}{T} + \Psi_\sigma \right),$$

*with $\mu = \sum_i p_i K_i$ for full participation and $\mu = \min_i K_i$ for partial participation, $F_0 = F(\boldsymbol{\theta}_0)$, $F^* = \min_{\boldsymbol{\theta}} F(\boldsymbol{\theta})$ the optimal value, and*

$$\Psi_\sigma = \eta\|\boldsymbol{p}\|^2 \left[ \sum_{i=1}^{n} K_i^2 \left( \frac{L\eta\sigma_i^2}{2m} + \sigma^2 \right) + (e-2)\eta^2 L^2 \sum_{i=1}^{n} K_i^3 \left( \frac{\sigma_i^2}{m} + 6K_i\sigma^2 \right) \right], \boldsymbol{p} = (p_1, \ldots, p_n).$$

*Proof.* We first assume full participation in the following theorem. The partial participation version is an easy extension and we discuss it in the end. We use $\boldsymbol{\theta}_{t,j}^{(i)}$ to denote the model parameters of client $i$ at global epoch $t$ and local step $j$. Due to the synchronization step, we have $\boldsymbol{\theta}_{t,0}^{(i)} = \boldsymbol{\theta}_t$, the global model at step $t$, and

$$\boldsymbol{\theta}_{t+1} = \sum_{i=1}^{n} p_i \boldsymbol{\theta}_{t,K_i}^{(i)}, \ p_i = \frac{n_i}{N}, \tag{B.3}$$

where $K_i$ is the local number of steps of client $i$. We also have:

$$\boldsymbol{\theta}_{t,j}^{(i)} = \boldsymbol{\theta}_{t,j-1}^{(i)} - \eta \boldsymbol{g}_{t,j}^{(i)}, \text{ for all } j \in [K_i]. \tag{B.4}$$

where $\boldsymbol{g}_{t,j}^{(i)} = \nabla \ell_{S_i^j}(\boldsymbol{\theta}_{t,j-1}^{(i)})$ is an unbiased estimator of $\nabla f_i(\boldsymbol{\theta}_{t,j-1}^{(i)})$ for $j \in [K_i]$, with $S_i^j$ the $j^{\text{th}}$ batch from client $i$. Combining eq. B.3 and eq. B.4 we have:

$$\boldsymbol{\theta}_{t+1} = \boldsymbol{\theta}_t - \eta \sum_{i=1}^{n} p_i \sum_{j=1}^{K_i} \boldsymbol{g}_{t,j}^{(i)}. \tag{B.5}$$

**Part I** Since each $f_i$ is $L$-Lipschitz smooth, so is their average $F = \sum_i p_i f_i$, from which we obtain that:

$$F(\boldsymbol{\theta}_{t+1}) \leq F(\boldsymbol{\theta}_t) + \langle \nabla F(\boldsymbol{\theta}_t), \boldsymbol{\theta}_{t+1} - \boldsymbol{\theta}_t \rangle + \frac{L}{2}\|\boldsymbol{\theta}_{t+1} - \boldsymbol{\theta}_t\|^2. \tag{B.6}$$

Plugging in eq. B.5 yields:

$$F(\boldsymbol{\theta}_{t+1}) \leq F(\boldsymbol{\theta}_t) - \eta \left\langle \nabla F(\boldsymbol{\theta}_t), \sum_{i=1}^{n} p_i \sum_{j=1}^{K_i} \boldsymbol{g}_{t,j}^{(i)} \right\rangle + \frac{L\eta^2}{2} \left\| \sum_{i=1}^{n} p_i \sum_{j=1}^{K_i} \boldsymbol{g}_{t,j}^{(i)} \right\|^2. \tag{B.7}$$

From the identity $\boldsymbol{g}_{t,j}^{(i)} = \boldsymbol{g}_{t,j}^{(i)} - \nabla F(\boldsymbol{\theta}_t) + \nabla F(\boldsymbol{\theta}_t)$, we write eq. B.7 as:

$$F(\boldsymbol{\theta}_{t+1}) \leq F(\boldsymbol{\theta}_t) - \eta \sum_{i=1}^{n} p_i K_i \|\nabla F(\boldsymbol{\theta}_t)\|^2 - \eta \left\langle \nabla F(\boldsymbol{\theta}_t), \sum_{i=1}^{n} p_i \sum_{j=1}^{K_i} (\boldsymbol{g}_{t,j}^{(i)} - \nabla F(\boldsymbol{\theta}_t)) \right\rangle +$$

$$+ \frac{L\eta^2}{2} \left\| \sum_{i=1}^{n} p_i \sum_{j=1}^{K_i} (\boldsymbol{g}_{t,j}^{(i)} - \nabla F(\boldsymbol{\theta}_t)) + \sum_{i=1}^{n} p_i K_i \nabla F(\boldsymbol{\theta}_t) \right\|^2. \tag{B.8}$$

By further expanding the last term we have:

$$F(\boldsymbol{\theta}_{t+1}) \leq F(\boldsymbol{\theta}_t) - \eta \sum_{i=1}^{n} p_i K_i \|\nabla F(\boldsymbol{\theta}_t)\|^2 - \eta \left\langle \nabla F(\boldsymbol{\theta}_t), \sum_{i=1}^{n} p_i \sum_{j=1}^{K_i} (\boldsymbol{g}_{t,j}^{(i)} - \nabla F(\boldsymbol{\theta}_t)) \right\rangle +$$

$$+ \frac{L\eta^2}{2} \left\| \sum_{i=1}^{n} p_i \sum_{j=1}^{K_i} (\boldsymbol{g}_{t,j}^{(i)} - \nabla F(\boldsymbol{\theta}_t)) \right\|^2 + \frac{L\eta^2}{2} \left( \sum_{i=1}^{n} p_i K_i \right)^2 \|\nabla F(\boldsymbol{\theta}_t)\|^2 +$$

$$+ L\eta^2 \left\langle \sum_{i=1}^{n} p_i K_i \nabla F(\boldsymbol{\theta}_t), \sum_{i=1}^{n} p_i \sum_{j=1}^{K_i} (\boldsymbol{g}_{t,j}^{(i)} - \nabla F(\boldsymbol{\theta}_t)) \right\rangle. \tag{B.9}$$

For simplicity we use $\mu$ as a shorthand for $\sum_{i=1}^{n} p_i K_i$. Grouping similar terms together gives:

$$F(\boldsymbol{\theta}_{t+1}) \leq F(\boldsymbol{\theta}_t) - \eta\mu \left( 1 - \frac{L\eta}{2}\mu \right) \|\nabla F(\boldsymbol{\theta}_t)\|^2$$

$$- \eta(1 - L\eta\mu) \left\langle \nabla F(\boldsymbol{\theta}_t), \sum_{i=1}^{n} p_i \sum_{j=1}^{K_i} (\boldsymbol{g}_{t,j}^{(i)} - \nabla F(\boldsymbol{\theta}_t)) \right\rangle + \frac{L\eta^2}{2} \left\| \sum_{i=1}^{n} p_i \sum_{j=1}^{K_i} (\boldsymbol{g}_{t,j}^{(i)} - \nabla F(\boldsymbol{\theta}_t)) \right\|^2. \tag{B.10}$$

Taking the expectation on both sides and with Cauchy–Schwarz inequality, we have:

$$\mathbb{E} F(\boldsymbol{\theta}_{t+1}) \leq \mathbb{E} F(\boldsymbol{\theta}_t) - \eta\mu \left( 1 - \frac{L\eta}{2}\mu \right) \mathbb{E}\|\nabla F(\boldsymbol{\theta}_t)\|^2 +$$

$$+ \eta(1 - L\eta\mu)\mathbb{E}\left[ \|\nabla F(\boldsymbol{\theta}_t)\| \cdot \left\| \sum_{i=1}^{n} p_i \sum_{j=1}^{K_i} (\nabla f_i(\boldsymbol{\theta}_{t,j-1}^{(i)}) - \nabla F(\boldsymbol{\theta}_t)) \right\| \right] +$$

$$+ \frac{L\eta^2}{2}\mathbb{E} \left\| \sum_{i=1}^{n} p_i \sum_{j=1}^{K_i} (\boldsymbol{g}_{t,j}^{(i)} - \nabla F(\boldsymbol{\theta}_t)) \right\|^2$$

$$\leq \mathbb{E} F(\boldsymbol{\theta}_t) + \left( -\eta\mu \left( 1 - \frac{L\eta}{2}\mu \right) + \frac{1}{2}\eta(1 - L\eta\mu) \right) \mathbb{E}\|\nabla F(\boldsymbol{\theta}_t)\|^2 +$$

$$+ \frac{L\eta^2}{2}\mathbb{E} \left\| \sum_{i=1}^{n} p_i \sum_{j=1}^{K_i} (\boldsymbol{g}_{t,j}^{(i)} - \nabla F(\boldsymbol{\theta}_t)) \right\|^2 + \frac{1}{2}\eta(1 - L\eta\mu)\mathbb{E} \left\| \sum_{i=1}^{n} p_i \sum_{j=1}^{K_i} (\nabla f_i(\boldsymbol{\theta}_{t,j-1}^{(i)}) - \nabla F(\boldsymbol{\theta}_t)) \right\|^2, \tag{B.11}$$

where we used $\mathbb{E}\boldsymbol{g}_{t,j}^{(i)} = \nabla f_i(\boldsymbol{\theta}_{t,j-1}^{(i)})$ and the inequality $ab \le \frac{1}{2}(a^2 + b^2)$ in the second line. Let us now study the two coefficients separately. From the assumption, $6\eta L K_i \le 1$ for any $i \in [n]$, and thus $6L\mu\eta \le 1$. Hence we have:

$$
\begin{aligned}
-\eta\mu\left(1 - \frac{L\eta}{2}\mu\right) + \frac{1}{2}\eta(1 - L\eta\mu) &\le -\eta\left(\mu - \frac{1}{2} - \frac{L\eta}{2}\mu^2 + \frac{1}{2}L\eta\mu\right) \\
&\le -\eta\left(\frac{11\mu - 6}{12} + \frac{L\eta\mu}{2}\right) \\
&\le -\eta\frac{11\mu - 6}{12}.
\end{aligned}
\tag{B.12}
$$

Therefore, eq. B.11 becomes:

$$
\mathbb{E}F(\boldsymbol{\theta}_{t+1}) \le \mathbb{E}F(\boldsymbol{\theta}_t) - \eta\frac{11\mu - 6}{12}\mathbb{E}\|\nabla F(\boldsymbol{\theta}_t)\|^2 + \frac{L\eta^2}{2}\mathbb{E}\left\|\sum_{i=1}^{n} p_i \sum_{j=1}^{K_i}(\boldsymbol{g}_{t,j}^{(i)} - \nabla F(\boldsymbol{\theta}_t))\right\|^2 +
$$

$$
+ \frac{1}{2}\eta(1 - L\eta\mu)\mathbb{E}\left\|\sum_{i=1}^{n} p_i \sum_{j=1}^{K_i}(\nabla f_i(\boldsymbol{\theta}_{t,j-1}^{(i)}) - \nabla F(\boldsymbol{\theta}_t))\right\|^2.
\tag{B.13}
$$

**Part II** With the following identity:

$$
\boldsymbol{g}_{t,j}^{(i)} - \nabla F(\boldsymbol{\theta}_t) = \boldsymbol{g}_{t,j}^{(i)} - \nabla f_i(\boldsymbol{\theta}_{t,j-1}^{(i)}) + \nabla f_i(\boldsymbol{\theta}_{t,j-1}^{(i)}) - \nabla F(\boldsymbol{\theta}_t),
\tag{B.14}
$$

the second last term of eq. B.11 can be simplified as:

$$
\mathbb{E}\left\|\sum_{i=1}^{n} p_i \sum_{j=1}^{K_i}(\boldsymbol{g}_{t,j}^{(i)} - \nabla f_i(\boldsymbol{\theta}_{t,j-1}^{(i)}))\right\|^2 + \mathbb{E}\left\|\sum_{i=1}^{n} p_i \sum_{j=1}^{K_i}(\nabla f_i(\boldsymbol{\theta}_{t,j-1}^{(i)}) - \nabla F(\boldsymbol{\theta}_t))\right\|^2,
\tag{B.15}
$$

where we note that $\boldsymbol{g}_{t,j}^{(i)}$ is an unbiased estimator of $\nabla f_i(\boldsymbol{\theta}_{t,j-1}^{(i)})$. We first bound the first term of eq. B.15:

$$
\begin{aligned}
\mathbb{E}\left\|\sum_{i=1}^{n} p_i \sum_{j=1}^{K_i}(\boldsymbol{g}_{t,j}^{(i)} - \nabla f_i(\boldsymbol{\theta}_{t,j-1}^{(i)}))\right\|^2 &\le \mathbb{E}\left(\sum_{i=1}^{n} p_i \sum_{j=1}^{K_i}\left\|\boldsymbol{g}_{t,j}^{(i)} - \nabla f_i(\boldsymbol{\theta}_{t,j-1}^{(i)})\right\|\right)^2 \\
&\le \mathbb{E}\|\boldsymbol{p}\|^2 \sum_{i=1}^{n}\left(\sum_{j=1}^{K_i}\left\|\boldsymbol{g}_{t,j}^{(i)} - \nabla f_i(\boldsymbol{\theta}_{t,j-1}^{(i)})\right\|\right)^2 \\
&\le \mathbb{E}\|\boldsymbol{p}\|^2 \sum_{i=1}^{n} K_i \sum_{j=1}^{K_i}\left\|\boldsymbol{g}_{t,j}^{(i)} - \nabla f_i(\boldsymbol{\theta}_{t,j-1}^{(i)})\right\|^2 \\
&= \|\boldsymbol{p}\|^2 \sum_{i=1}^{n} K_i \sum_{j=1}^{K_i}\mathbb{E}\left\|\boldsymbol{g}_{t,j}^{(i)} - \nabla f_i(\boldsymbol{\theta}_{t,j-1}^{(i)})\right\|^2 \\
&\le \|\boldsymbol{p}\|^2 \sum_{i=1}^{n} K_i^2\frac{\sigma_i^2}{m},
\end{aligned}
\tag{B.16}
$$

where in the first line, we used triangle inequality; in the second and third lines, we used the Cauchy–Schwarz inequality; in the fourth line we used the linearity of expectation; in the final line we note that given the last part of Assumption 4.1, we have:

$$
\mathbb{E}_{S_i \sim \mathcal{D}_i^m}\left\|\frac{1}{|S_i|}\sum_{(\boldsymbol{x},y)\in S_i}\nabla\ell(\boldsymbol{\theta},(\boldsymbol{x},y)) - \nabla f_i(\boldsymbol{\theta})\right\|^2 = \frac{1}{|S_i|^2}\sum_{(\boldsymbol{x},y)\in S_i}\mathbb{E}_{(\boldsymbol{x},y)\sim\mathcal{D}_i}\|\nabla\ell(\boldsymbol{\theta},(\boldsymbol{x},y)) - \nabla f_i(\boldsymbol{\theta})\|^2 \le \frac{\sigma_i^2}{m},
\tag{B.17}
$$

where we used the property that each $(\boldsymbol{x}, y)$ is an i.i.d. sample from $\mathcal{D}_i$, and that the estimation is unbiased (by the definition of $f_i$). Similarly we bound the second term of eq. B.15:

$$\mathbb{E}\left\|\sum_{i=1}^{n} p_i \sum_{j=1}^{K_i} (\nabla f_i(\boldsymbol{\theta}_{t,j-1}^{(i)}) - \nabla F(\boldsymbol{\theta}_t))\right\|^2 \leq \|\boldsymbol{p}\|^2 \sum_{i=1}^{n} K_i \sum_{j=1}^{K_i} \mathbb{E}\left\|\nabla f_i(\boldsymbol{\theta}_{t,j-1}^{(i)}) - \nabla F(\boldsymbol{\theta}_t)\right\|^2. \quad \text{(B.18)}$$

With the following identity:

$$\nabla f_i(\boldsymbol{\theta}_{t,j-1}^{(i)}) - \nabla F(\boldsymbol{\theta}_t) = \nabla f_i(\boldsymbol{\theta}_{t,j-1}^{(i)}) - \nabla f_i(\boldsymbol{\theta}_t) + \nabla f_i(\boldsymbol{\theta}_t) - \nabla F(\boldsymbol{\theta}_t), \quad \text{(B.19)}$$

and taking the squared norm on both sides, we have:

$$\|\nabla f_i(\boldsymbol{\theta}_{t,j-1}^{(i)}) - \nabla F(\boldsymbol{\theta}_t)\|^2 \leq 2\|\nabla f_i(\boldsymbol{\theta}_{t,j-1}^{(i)}) - \nabla f_i(\boldsymbol{\theta}_t)\|^2 + 2\|\nabla f_i(\boldsymbol{\theta}_t) - \nabla F(\boldsymbol{\theta}_t)\|^2$$
$$\leq 2L^2\|\boldsymbol{\theta}_{t,j-1}^{(i)} - \boldsymbol{\theta}_t\|^2 + 2\sigma^2, \quad \text{(B.20)}$$

where we note that:

$$\|\nabla f_i(\boldsymbol{\theta}_t) - \nabla F(\boldsymbol{\theta}_t)\| = \left\|\nabla f_i(\boldsymbol{\theta}_t) - \sum_{j=1}^{n} p_j \nabla f_j(\boldsymbol{\theta}_t)\right\|$$
$$= \left\|\sum_{j=1}^{n} p_j (\nabla f_i(\boldsymbol{\theta}_t) - \nabla f_j(\boldsymbol{\theta}_t))\right\|$$
$$\leq \sum_{j=1}^{n} p_j \|\nabla f_i(\boldsymbol{\theta}_t) - \nabla f_j(\boldsymbol{\theta}_t)\|$$
$$\leq \sigma, \quad \text{(B.21)}$$

where in the third line we used the triangle inequality and in the last line we used Assumption 4.1. Plugging eq. B.20 into eq. B.18 yields:

$$\mathbb{E}\left\|\sum_{i=1}^{n} p_i \sum_{j=1}^{K_i} (\nabla f_i(\boldsymbol{\theta}_{t,j-1}^{(i)}) - \nabla F(\boldsymbol{\theta}_t))\right\|^2 \leq 2\|\boldsymbol{p}\|^2 \sum_{i=1}^{n} K_i^2 \sigma^2 + 2L^2\|\boldsymbol{p}\|^2 \sum_{i=1}^{n} K_i \sum_{j=1}^{K_i} \mathbb{E}\|\boldsymbol{\theta}_{t,j-1}^{(i)} - \boldsymbol{\theta}_t\|^2. \quad \text{(B.22)}$$

Bringing eq. B.16 and eq. B.22 into eq. B.15 we write:

$$\mathbb{E}\left\|\sum_{i=1}^{n} p_i \sum_{j=1}^{K_i} (\boldsymbol{g}_{t,j}^{(i)} - \nabla F(\boldsymbol{\theta}_t))\right\|^2 \leq \|\boldsymbol{p}\|^2 \sum_{i=1}^{n} K_i^2 \left(\frac{\sigma_i^2}{m} + 2\sigma^2\right) + 2L^2\|\boldsymbol{p}\|^2 \sum_{i=1}^{n} K_i \sum_{j=1}^{K_i} \mathbb{E}\|\boldsymbol{\theta}_{t,j-1}^{(i)} - \boldsymbol{\theta}_t\|^2$$
$$\leq \|\boldsymbol{p}\|^2 \sum_{i=1}^{n} K_i^2 \left(\frac{\sigma_i^2}{m} + 2\sigma^2\right) + 2L^2\|\boldsymbol{p}\|^2 \sum_{i=1}^{n} K_i \sum_{j=0}^{K_i-1} \mathbb{E}\|\boldsymbol{\theta}_{t,j}^{(i)} - \boldsymbol{\theta}_t\|^2. \quad \text{(B.23)}$$

**Part III** Now let us give an upper bound for $\mathbb{E}\|\boldsymbol{\theta}_{t,j}^{(i)} - \boldsymbol{\theta}_t\|^2$. From eq. B.23, we only need to focus on $K_i \geq 2$ and $j = 1, \ldots, K_i - 1$ since $\boldsymbol{\theta}_{t,0}^{(i)} = \boldsymbol{\theta}_t$. For $j \in [K_i - 1]$, we have from eq. B.4:

$$\mathbb{E}\|\boldsymbol{\theta}_{t,j}^{(i)} - \boldsymbol{\theta}_t\|^2 = \mathbb{E}\|\boldsymbol{\theta}_{t,j-1}^{(i)} - \boldsymbol{\theta}_t - \eta \boldsymbol{g}_{t,j}^{(i)}\|^2$$
$$= \mathbb{E}\|\boldsymbol{\theta}_{t,j-1}^{(i)} - \boldsymbol{\theta}_t - \eta \nabla f_i(\boldsymbol{\theta}_{t,j-1}^{(i)}) + \eta \nabla f_i(\boldsymbol{\theta}_{t,j-1}^{(i)}) - \eta \boldsymbol{g}_{t,j}^{(i)}\|^2$$
$$= \mathbb{E}\|\boldsymbol{\theta}_{t,j-1}^{(i)} - \boldsymbol{\theta}_t - \eta \nabla f_i(\boldsymbol{\theta}_{t,j-1}^{(i)})\|^2 + \mathbb{E}\eta^2\|\nabla f_i(\boldsymbol{\theta}_{t,j-1}^{(i)}) - \boldsymbol{g}_{t,j}^{(i)}\|^2$$
$$= \mathbb{E}\|\boldsymbol{\theta}_{t,j-1}^{(i)} - \boldsymbol{\theta}_t - \eta \nabla f_i(\boldsymbol{\theta}_{t,j-1}^{(i)})\|^2 + \eta^2 \frac{\sigma_i^2}{m}, \quad \text{(B.24)}$$

where in the third line we note that $\boldsymbol{g}_{t,j}^{(i)}$ is an unbiased estimator of $\nabla f_i(\boldsymbol{\theta}_{t,j-1}^{(i)})$ and in the last line we used eq. B.17. The first term in the last line above can be bounded as:

$$\mathbb{E}\|\boldsymbol{\theta}_{t,j-1}^{(i)} - \boldsymbol{\theta}_t - \eta\nabla f_i(\boldsymbol{\theta}_{t,j-1}^{(i)})\|^2 \leq \left(1 + \frac{1}{2K_i - 1}\right)\mathbb{E}\|\boldsymbol{\theta}_{t,j-1}^{(i)} - \boldsymbol{\theta}_t\|^2 + 2K_i\eta^2\|\nabla f_i(\boldsymbol{\theta}_{t,j-1}^{(i)})\|^2, \qquad (B.25)$$

where we used $\|a + b\|^2 \leq (1 + \frac{1}{\alpha})\|a\|^2 + (1 + \alpha)\|b\|^2$ for any vectors $a, b$ with the same dimension and $\alpha > 0$. Since

$$\nabla f_i(\boldsymbol{\theta}_{t,j-1}^{(i)}) = (\nabla f_i(\boldsymbol{\theta}_{t,j-1}^{(i)}) - \nabla f_i(\boldsymbol{\theta}_t)) + (\nabla f_i(\boldsymbol{\theta}_t) - \nabla F(\boldsymbol{\theta}_t)) + \nabla F(\boldsymbol{\theta}_t), \qquad (B.26)$$

taking the squared norm on both sides we have (note that $(a + b + c)^2 \leq 3(a^2 + b^2 + c^2)$):

$$\|\nabla f_i(\boldsymbol{\theta}_{t,j-1}^{(i)})\|^2 \leq 3\|\nabla f_i(\boldsymbol{\theta}_{t,j-1}^{(i)}) - \nabla f_i(\boldsymbol{\theta}_t)\|^2 + 3\|\nabla f_i(\boldsymbol{\theta}_t) - \nabla F(\boldsymbol{\theta}_t)\|^2 + 3\|\nabla F(\boldsymbol{\theta}_t)\|^2$$
$$\leq 3L^2\|\boldsymbol{\theta}_{t,j-1}^{(i)} - \boldsymbol{\theta}_t\|^2 + 3\sigma^2 + 3\|\nabla F(\boldsymbol{\theta}_t)\|^2, \qquad (B.27)$$

where in the second line we used eq. B.21 and Assumption 4.1. Plugging eq. B.27 into eq. B.25 we find:

$$\mathbb{E}\|\boldsymbol{\theta}_{t,j-1}^{(i)} - \boldsymbol{\theta}_t - \eta\nabla f_i(\boldsymbol{\theta}_{t,j-1}^{(i)})\|^2 \leq \left(1 + \frac{1}{2K_i - 1} + 6K_i\eta^2 L^2\right)\mathbb{E}\|\boldsymbol{\theta}_{t,j-1}^{(i)} - \boldsymbol{\theta}_t\|^2 + 6K_i\eta^2\sigma^2 + 6K_i\eta^2\mathbb{E}\|\nabla F(\boldsymbol{\theta}_t)\|^2.$$
$$(B.28)$$

Combined with eq. B.24, we obtain:

$$\mathbb{E}\|\boldsymbol{\theta}_{t,j}^{(i)} - \boldsymbol{\theta}_t\|^2 \leq \left(1 + \frac{1}{2K_i - 1} + 6K_i\eta^2 L^2\right)\mathbb{E}\|\boldsymbol{\theta}_{t,j-1}^{(i)} - \boldsymbol{\theta}_t\|^2 + \eta^2\left(\frac{\sigma_i^2}{m} + 6K_i\sigma^2\right) + 6K_i\eta^2\mathbb{E}\|\nabla F(\boldsymbol{\theta}_t)\|^2.$$
$$(B.29)$$

Recall that we assumed $\eta \leq \min_i\{\frac{1}{6K_i L}\}$. For $K_i \geq 2$ (note the assumption at the beginning of Part III), we have:

$$1 + \frac{1}{2K_i - 1} + 6K_i\eta^2 L^2 \leq 1 + \frac{1}{2K_i - 1} + \frac{1}{6K_i} \leq 1 + \frac{1}{K_i}. \qquad (B.30)$$

Therefore, eq. B.29 becomes:

$$\mathbb{E}\|\boldsymbol{\theta}_{t,j}^{(i)} - \boldsymbol{\theta}_t\|^2 \leq \left(1 + \frac{1}{K_i}\right)\mathbb{E}\|\boldsymbol{\theta}_{t,j-1}^{(i)} - \boldsymbol{\theta}_t\|^2 + \eta^2\left(\frac{\sigma_i^2}{m} + 6K_i\sigma^2\right) + 6K_i\eta^2\mathbb{E}\|\nabla F(\boldsymbol{\theta}_t)\|^2. \qquad (B.31)$$

We can treat $\{a_j = \mathbb{E}\|\boldsymbol{\theta}_{t,j}^{(i)} - \boldsymbol{\theta}_t\|^2\|\}_{j=1}^{K_i - 1}$ as a sequence. Unrolling this sequence and with $a_0 = 0$, we have:

$$\mathbb{E}\|\boldsymbol{\theta}_{t,j}^{(i)} - \boldsymbol{\theta}_t\|^2 \leq \frac{\left(1 + \frac{1}{K_i}\right)^j - 1}{1 + \frac{1}{K_i} - 1}\left(\eta^2\left(\frac{\sigma_i^2}{m} + 6K_i\sigma^2\right) + 6K_i\eta^2\mathbb{E}\|\nabla F(\boldsymbol{\theta}_t)\|^2\right)$$
$$= K_i\left(\left(1 + \frac{1}{K_i}\right)^j - 1\right)\left(\eta^2\left(\frac{\sigma_i^2}{m} + 6K_i\sigma^2\right) + 6K_i\eta^2\mathbb{E}\|\nabla F(\boldsymbol{\theta}_t)\|^2\right). \qquad (B.32)$$

Summing over $j = 0, 1, \ldots, K_i - 1$ gives:

$$\sum_{j=0}^{K_i - 1}\mathbb{E}\|\boldsymbol{\theta}_{t,j}^{(i)} - \boldsymbol{\theta}_t\|^2 \leq K_i^2\left(\left(1 + \frac{1}{K_i}\right)^{K_i} - 2\right)\left(\eta^2\left(\frac{\sigma_i^2}{m} + 6K_i\sigma^2\right) + 6K_i\eta^2\mathbb{E}\|\nabla F(\boldsymbol{\theta}_t)\|^2\right)$$
$$\leq (e - 2)K_i^2\left(\eta^2\left(\frac{\sigma_i^2}{m} + 6K_i\sigma^2\right) + 6K_i\eta^2\mathbb{E}\|\nabla F(\boldsymbol{\theta}_t)\|^2\right)$$
$$= (e - 2)K_i^2\eta^2\left(\frac{\sigma_i^2}{m} + 6K_i\sigma^2 + 6K_i\mathbb{E}\|\nabla F(\boldsymbol{\theta}_t)\|^2\right) \qquad (B.33)$$

where in the first line we used the geometric series formula $1 + q + \cdots + q^{n-1} = \frac{q^n - 1}{q - 1}$; in the second line we used the fact that $\left(1 + \frac{1}{K_i}\right)^{K_i} \leq e$ for $K_i \geq 1$, with $e$ the natural logarithm.

**Part IV** We finally put things together and finish our proof. From eq. B.13 we have:

$$\mathbb{E}F(\boldsymbol{\theta}_{t+1}) \leq \mathbb{E}F(\boldsymbol{\theta}_t) - \eta\frac{11\mu-6}{12}\mathbb{E}\|\nabla F(\boldsymbol{\theta}_t)\|^2 + \frac{L\eta^2}{2}\mathbb{E}\left\|\sum_{i=1}^{n}p_i\sum_{j=1}^{K_i}(\boldsymbol{g}_{t,j}^{(i)} - \nabla F(\boldsymbol{\theta}_t))\right\|^2 +$$

$$+ \frac{1}{2}\eta(1-L\eta\mu)\mathbb{E}\left\|\sum_{i=1}^{n}p_i\sum_{j=1}^{K_i}(\nabla f_i(\boldsymbol{\theta}_{t,j-1}^{(i)}) - \nabla F(\boldsymbol{\theta}_t))\right\|^2$$

$$= \mathbb{E}F(\boldsymbol{\theta}_t) - \eta\frac{11\mu-6}{12}\mathbb{E}\|\nabla F(\boldsymbol{\theta}_t)\|^2 + \frac{L\eta^2}{2}\mathbb{E}\left\|\sum_{i=1}^{n}p_i\sum_{j=1}^{K_i}(\boldsymbol{g}_{t,j}^{(i)} - \nabla f_i(\boldsymbol{\theta}_{t,j-1}^{(i)}))\right\|^2 +$$

$$+ \left(\frac{L\eta^2}{2} + \frac{1}{2}\eta(1-L\eta\mu)\right)\mathbb{E}\left\|\sum_{i=1}^{n}p_i\sum_{j=1}^{K_i}(\nabla f_i(\boldsymbol{\theta}_{t,j-1}^{(i)}) - \nabla F(\boldsymbol{\theta}_t))\right\|^2$$

$$\leq \mathbb{E}F(\boldsymbol{\theta}_t) - \eta\frac{11\mu-6}{12}\mathbb{E}\|\nabla F(\boldsymbol{\theta}_t)\|^2 + \frac{L\eta^2}{2}\|\boldsymbol{p}\|^2\sum_{i=1}^{n}K_i^2\frac{\sigma_i^2}{m} + \frac{\eta}{2}\mathbb{E}\left\|\sum_{i=1}^{n}p_i\sum_{j=1}^{K_i}(\nabla f_i(\boldsymbol{\theta}_{t,j-1}^{(i)}) - \nabla F(\boldsymbol{\theta}_t))\right\|^2$$

$$\leq \mathbb{E}F(\boldsymbol{\theta}_t) - \eta\frac{11\mu-6}{12}\mathbb{E}\|\nabla F(\boldsymbol{\theta}_t)\|^2 + \frac{L\eta^2}{2}\|\boldsymbol{p}\|^2\sum_{i=1}^{n}K_i^2\frac{\sigma_i^2}{m} +$$

$$+ \frac{\eta}{2}\left(2\|\boldsymbol{p}\|^2\sum_{i=1}^{n}K_i^2\sigma^2 + 2L^2\|\boldsymbol{p}\|^2\sum_{i=1}^{n}K_i\sum_{j=1}^{K_i}\mathbb{E}\|\boldsymbol{\theta}_{t,j-1}^{(i)} - \boldsymbol{\theta}_t\|^2\right)$$

$$\leq \mathbb{E}F(\boldsymbol{\theta}_t) - \eta\frac{11\mu-6}{12}\mathbb{E}\|\nabla F(\boldsymbol{\theta}_t)\|^2 + 6(e-2)\eta^3L^2\|\boldsymbol{p}\|^2\sum_{i=1}^{n}K_i^4\mathbb{E}\|\nabla F(\boldsymbol{\theta}_t)\|^2 + \Psi_\sigma, \tag{B.34}$$

where in the third line we used eq. B.15; in the fifth line we used eq. B.16 and note that

$$\frac{L\eta^2}{2} + \frac{1}{2}\eta(1-L\eta\mu) = \frac{\eta}{2} + \frac{L\eta^2}{2}(1-\mu) \leq \frac{\eta}{2}; \tag{B.35}$$

in the seventh line we used eq. B.22; and in the final line we used eq. B.33 and denoted

$$\Psi_\sigma = \eta\|\boldsymbol{p}\|^2\left[\sum_{i=1}^{n}K_i^2\left(\frac{L\eta\sigma_i^2}{2m} + \sigma^2\right) + (e-2)\eta^2L^2\sum_{i=1}^{n}K_i^3\left(\frac{\sigma_i^2}{m} + 6K_i\sigma^2\right)\right]. \tag{B.36}$$

Since we assumed:

$$\eta \leq \frac{1}{L}\sqrt{\frac{1}{24(e-2)\|\boldsymbol{p}\|^2(\sum_{i=1}^{n}K_i^4)}}, \tag{B.37}$$

we have:

$$\frac{11\mu-6}{12} - 6(e-2)L^2\|\boldsymbol{p}\|^2\sum_{i=1}^{n}K_i^4\eta^2 \geq \frac{11\mu-6}{12} - \frac{1}{4} \geq \frac{11\mu-9}{12}. \tag{B.38}$$

Therefore, eq. B.34 becomes:

$$\mathbb{E}F(\boldsymbol{\theta}_{t+1}) \leq \mathbb{E}F(\boldsymbol{\theta}_t) - \eta\frac{11\mu-9}{12}\mathbb{E}\|\nabla F(\boldsymbol{\theta}_t)\|^2 + \Psi_\sigma. \tag{B.39}$$

With some algebra we obtain:

$$\eta\frac{11\mu-9}{12}\mathbb{E}\|\nabla F(\boldsymbol{\theta}_t)\|^2 \leq \mathbb{E}[F(\boldsymbol{\theta}_t) - F(\boldsymbol{\theta}_{t+1})] + \Psi_\sigma. \tag{B.40}$$

Summing both sides over $t = 0, \ldots, T - 1$ and dividing by $T$, we have:

$$\eta \frac{11\mu - 9}{12} \frac{1}{T} \sum_{t=0}^{T-1} \mathbb{E}\|\nabla F(\boldsymbol{\theta}_t)\|^2 \leq \frac{F(\boldsymbol{\theta}_0) - F^*}{T} + \Psi_\sigma, \tag{B.41}$$

which gives:

$$\min_{0 \leq t \leq T-1} \mathbb{E}\|\nabla F(\boldsymbol{\theta}_t)\|^2 \leq \frac{12}{(11\mu - 9)\eta} \left( \frac{F(\boldsymbol{\theta}_0) - F^*}{T} + \Psi_\sigma \right), \tag{B.42}$$

with $F^* = \min_{\boldsymbol{\theta}} F(\boldsymbol{\theta})$ the optimal value.

Finally, for the partial participation, it suffices to replace the client set $\{1, \ldots, n\}$ with its subset. Note that after this substitution, the new variance term satisfies $\Psi'_\sigma \leq \Psi_\sigma$ since this term increases with more participants, and eq. B.38 still holds since we subtract a smaller term with partial participation. We also need to modify eq. B.38 so we further lower bound eq. B.38 with $\mu \geq \min_i K_i$. $\qquad \square$

**Theorem 4.4 (PropFair).** *Denote $\widetilde{L} = \frac{4}{M^2}(\frac{3}{2}ML + L_0^2)$ and $p_i = \frac{n_i}{N}$. Given Assumptions 4.1 and 4.3, assume that the local learning rate satisfies:*

$$\eta \leq \min \left\{ \min_{i \in [n]} \frac{1}{6\widetilde{L}K_i}, \frac{1}{8\widetilde{L}} \sqrt{\frac{1}{(e-2)(\sum_i p_i^2)(\sum_i K_i^4)}} \right\}. \tag{4.3}$$

*By running Algorithm 1 for $T$ global epochs we have:*

$$\min_{0 \leq t \leq T-1} \mathbb{E}\|\nabla \pi(\boldsymbol{\theta}_t)\|^2 \leq \frac{12}{(11\mu - 9)\eta} \left( \frac{\pi_0 - \pi^*}{T} + \widetilde{\Psi}_\sigma \right),$$

*with $\mu = \sum_i p_i K_i$ for full participation and $\mu = \min_i K_i$ for partial participation, $\pi_0 = \pi(\boldsymbol{\theta}_0)$, $\pi^* = \min_{\boldsymbol{\theta}} \pi(\boldsymbol{\theta})$ the optimal value, and*

$$\widetilde{\Psi}_\sigma = \eta \|\boldsymbol{p}\|^2 \left[ \sum_{i=1}^n K_i^2 \left( \frac{\widetilde{\sigma}_i^2}{m} + 2\widetilde{\sigma}^2 \right) + 16(e-2)\eta^2 \widetilde{L}^2 \sum_{i=1}^n K_i^4 \left( \frac{\widetilde{\sigma}_i^2}{m} + \widetilde{\sigma}^2 \right) \right]$$

*where $\widetilde{\sigma}_i^2 = \frac{8}{M^4}(9M^2\sigma_i^2 + 4L_0^2\sigma_{0,i}^2)$ and $\widetilde{\sigma} = \frac{4}{M}\left(\frac{3}{2}\sigma + \frac{L_0}{M}\sigma_0\right)$.*

*Proof.* The proof follows similarly the proof of FedAvg (Theorem 4.2). Denote $\varphi(t) = -\log(M - t)$. The changes of PropFair compared to FedAvg as follows:

- The aggregate loss for each client $i$ is not $f_i$, but $\varphi \circ f_i$;

- The objective function is not $F = \sum_i p_i f_i$, but $\pi = \sum_i p_i \varphi \circ f_i$;

- For each batch $S_i \sim \mathcal{D}_i^m$ from client $i$, the batch loss is not $\ell_{S_i}$, but $\varphi \circ \ell_{S_i}$.

Note that in Assumption 4.1 we implicitly required eq. 2.1:

$$f_i(\boldsymbol{\theta}) = \mathbb{E}_{(\boldsymbol{x}, y) \sim \mathcal{D}_i}[\ell(\boldsymbol{\theta}, (\boldsymbol{x}, y))],$$

or in other words, $\ell(\boldsymbol{\theta}, (\boldsymbol{x}, y))$ is an *unbiased* estimator of $f_i$. This is no longer true if we replace $\ell$ with $\varphi \circ \ell$ and $f_i$ with $\varphi \circ f_i$. Similarly, $\varphi \circ \ell_{S_i}$ is no longer an unbiased estimator of $\varphi \circ f_i$. We will take care of this pitfall in our proof.

First, from the Lipschitzness assumption in Assumption 4.3 we can obtain an upper bound for the gradient: $\|\nabla f_i(\boldsymbol{\theta})\| \leq L_0$ for any $\boldsymbol{\theta} \in \mathbb{R}^d$. We will use this result, as well as the rest of Assumption 4.3, to derive similar bounds as in Assumption 4.1:

- the Lipschitz constant of $\nabla \varphi \circ f_i$;

- the Lipschitz constant of $\varphi \circ f_i - \varphi \circ f_j$;

- the variance of each batch $\nabla \varphi \circ \ell_{S_i}$.

For the Lipschitz smooth constant of $\varphi \circ f_i$, we write:

$$
\begin{aligned}
\|\nabla(\varphi \circ f_i)(\boldsymbol{\theta}) - \nabla(\varphi \circ f_i)(\boldsymbol{\theta}')\| &= \left\| \frac{\nabla f_i(\boldsymbol{\theta})}{M - f_i(\boldsymbol{\theta})} - \frac{\nabla f_i(\boldsymbol{\theta}')}{M - f_i(\boldsymbol{\theta}')} \right\| \\
&= \left\| \frac{M(\nabla f_i(\boldsymbol{\theta}) - \nabla f_i(\boldsymbol{\theta}')) - \nabla f_i(\boldsymbol{\theta}) f_i(\boldsymbol{\theta}') + \nabla f_i(\boldsymbol{\theta}') f_i(\boldsymbol{\theta})}{(M - f_i(\boldsymbol{\theta}))(M - f_i(\boldsymbol{\theta}'))} \right\| \\
&\leq \frac{4}{M^2} \left( M \|\nabla f_i(\boldsymbol{\theta}) - \nabla f_i(\boldsymbol{\theta}')\| + \|\nabla f_i(\boldsymbol{\theta}) f_i(\boldsymbol{\theta}') - \nabla f_i(\boldsymbol{\theta}') f_i(\boldsymbol{\theta})\| \right) \\
&\leq \frac{4}{M^2} (ML\|\boldsymbol{\theta} - \boldsymbol{\theta}'\| + \|\nabla f_i(\boldsymbol{\theta}) f_i(\boldsymbol{\theta}') - \nabla f_i(\boldsymbol{\theta}') f_i(\boldsymbol{\theta})\|).
\end{aligned}
\tag{B.43}
$$

The second term in the parenthesis above can be computed as:

$$
\begin{aligned}
\|\nabla f_i(\boldsymbol{\theta}) f_i(\boldsymbol{\theta}') - \nabla f_i(\boldsymbol{\theta}') f_i(\boldsymbol{\theta}))\| &= \|\nabla f_i(\boldsymbol{\theta}) f_i(\boldsymbol{\theta}') - \nabla f_i(\boldsymbol{\theta}) f_i(\boldsymbol{\theta}) + \nabla f_i(\boldsymbol{\theta}) f_i(\boldsymbol{\theta}) - \nabla f_i(\boldsymbol{\theta}') f_i(\boldsymbol{\theta}))\| \\
&\leq \|\nabla f_i(\boldsymbol{\theta}) f_i(\boldsymbol{\theta}') - \nabla f_i(\boldsymbol{\theta}) f_i(\boldsymbol{\theta})\| + \|\nabla f_i(\boldsymbol{\theta}) f_i(\boldsymbol{\theta}) - \nabla f_i(\boldsymbol{\theta}') f_i(\boldsymbol{\theta})\| \\
&= \|\nabla f_i(\boldsymbol{\theta})\| \cdot \|f_i(\boldsymbol{\theta}') - f_i(\boldsymbol{\theta})\| + \|\nabla f_i(\boldsymbol{\theta}) - \nabla f_i(\boldsymbol{\theta}')\| \cdot \|f_i(\boldsymbol{\theta})\| \\
&\leq L_0^2 \|\boldsymbol{\theta}' - \boldsymbol{\theta}\| + L \frac{M}{2} \|\boldsymbol{\theta}' - \boldsymbol{\theta}\|,
\end{aligned}
\tag{B.44}
$$

where in the second line we used triangle inequality; in the fourth line we used Assumptions 4.1 and 4.3. Plugging in back to eq. B.43 we have:

$$
\|\nabla(\varphi \circ f_i)(\boldsymbol{\theta}) - \nabla(\varphi \circ f_i)(\boldsymbol{\theta}')\| \leq \frac{4}{M^2} \left( \frac{3}{2} ML + L_0^2 \right) \|\boldsymbol{\theta} - \boldsymbol{\theta}'\|.
\tag{B.45}
$$

Let us now figure out the variance terms. For the global variance term, we similarly write:

$$
\begin{aligned}
\|\nabla(\varphi \circ f_i)(\boldsymbol{\theta}) - \nabla(\varphi \circ f_j)(\boldsymbol{\theta})\| &= \left\| \frac{\nabla f_i(\boldsymbol{\theta})}{M - f_i(\boldsymbol{\theta})} - \frac{\nabla f_j(\boldsymbol{\theta})}{M - f_j(\boldsymbol{\theta})} \right\| \\
&= \left\| \frac{M(\nabla f_i(\boldsymbol{\theta}) - \nabla f_j(\boldsymbol{\theta})) - \nabla f_i(\boldsymbol{\theta}) f_j(\boldsymbol{\theta}) + \nabla f_j(\boldsymbol{\theta}) f_i(\boldsymbol{\theta})}{(M - f_i(\boldsymbol{\theta}))(M - f_j(\boldsymbol{\theta}))} \right\| \\
&\leq \frac{4}{M^2} \left( M \|\nabla f_i(\boldsymbol{\theta}) - \nabla f_j(\boldsymbol{\theta})\| + \|\nabla f_i(\boldsymbol{\theta}) f_j(\boldsymbol{\theta}) - \nabla f_j(\boldsymbol{\theta}) f_i(\boldsymbol{\theta})\| \right) \\
&\leq \frac{4}{M^2} (M\sigma + \|\nabla f_i(\boldsymbol{\theta}) f_j(\boldsymbol{\theta}) - \nabla f_j(\boldsymbol{\theta}) f_i(\boldsymbol{\theta})\|).
\end{aligned}
\tag{B.46}
$$

The second term in the parenthesis above can be computed as:

$$
\begin{aligned}
\|\nabla f_i(\boldsymbol{\theta}) f_j(\boldsymbol{\theta}) - \nabla f_j(\boldsymbol{\theta}) f_i(\boldsymbol{\theta}))\| &= \|\nabla f_i(\boldsymbol{\theta}) f_j(\boldsymbol{\theta}) - \nabla f_i(\boldsymbol{\theta}) f_i(\boldsymbol{\theta}) + \nabla f_i(\boldsymbol{\theta}) f_i(\boldsymbol{\theta}) - \nabla f_j(\boldsymbol{\theta}) f_i(\boldsymbol{\theta})\| \\
&\leq \|\nabla f_i(\boldsymbol{\theta}) f_j(\boldsymbol{\theta}) - \nabla f_i(\boldsymbol{\theta}) f_i(\boldsymbol{\theta})\| + \|\nabla f_i(\boldsymbol{\theta}) f_i(\boldsymbol{\theta}) - \nabla f_j(\boldsymbol{\theta}) f_i(\boldsymbol{\theta})\| \\
&= \|\nabla f_i(\boldsymbol{\theta})\| \cdot \|f_j(\boldsymbol{\theta}) - f_i(\boldsymbol{\theta})\| + \|\nabla f_i(\boldsymbol{\theta}) - \nabla f_j(\boldsymbol{\theta})\| \cdot \|f_i(\boldsymbol{\theta})\| \\
&\leq L_0 \sigma_0 + \frac{M}{2} \sigma,
\end{aligned}
\tag{B.47}
$$

where in the second line we used triangle inequality; in the last line we used Assumptions 4.1 and 4.3. Plugging eq. B.47 into eq. B.46 we find:

$$
\|\nabla(\varphi \circ f_i)(\boldsymbol{\theta}) - \nabla(\varphi \circ f_j)(\boldsymbol{\theta})\| \leq \frac{4}{M} \left( \frac{3}{2} \sigma + \frac{L_0}{M} \sigma_0 \right).
\tag{B.48}
$$

Let us finally compute the new local variance term for each batch. Recall that we denoted $\ell_{S_i}(\theta) := \frac{1}{|S_i|} \sum_{(\boldsymbol{x},y) \in S_i} \ell(\boldsymbol{\theta}, (\boldsymbol{x}, y))$, with $S_i \sim \mathcal{D}_i^m$. We can write

$$
\|\nabla(\varphi \circ f_i)(\boldsymbol{\theta}) - \nabla(\varphi \circ \ell_{S_i})(\boldsymbol{\theta})\| = \left\| \frac{\nabla f_i(\boldsymbol{\theta})}{M - f_i(\boldsymbol{\theta})} - \frac{\nabla \ell_{S_i}(\boldsymbol{\theta})}{M - \ell_{S_i}(\boldsymbol{\theta})} \right\|
$$
$$
\leq \frac{4}{M^2} \left( \frac{3M}{2} \|\nabla f_i(\boldsymbol{\theta}) - \nabla \ell_{S_i}(\boldsymbol{\theta})\| + L_0 \|f_i(\boldsymbol{\theta}) - \ell_{S_i}(\boldsymbol{\theta})\| \right), \quad \text{(B.49)}
$$

and the derivation follows similarly as eq. B.46. Taking the square on both sides and taking the expectation over $S_i \sim \mathcal{D}_i^m$, we obtain:

$$
\mathbb{E}_{S_i \sim \mathcal{D}_i^m} \|\nabla(\varphi \circ f_i)(\boldsymbol{\theta}) - \nabla(\varphi \circ \ell_{S_i})(\boldsymbol{\theta})\|^2 \leq \frac{16}{M^4} \left( 2 \frac{9M^2}{4} \mathbb{E}_{S_i \sim \mathcal{D}_i^m} \|\nabla f_i(\boldsymbol{\theta}) - \nabla \ell_{S_i}(\boldsymbol{\theta})\|^2 + \right.
$$
$$
\left. + 2L_0^2 \mathbb{E}_{S_i \sim \mathcal{D}_i^m} \|f_i(\boldsymbol{\theta}) - \ell_{S_i}(\boldsymbol{\theta})\|^2 \right)
$$
$$
\leq \frac{8}{M^4} \left( 9M^2 \frac{\sigma_i^2}{m} + 4L_0^2 \frac{\sigma_{0,i}^2}{m} \right), \quad \text{(B.50)}
$$

where in the first line we used $(a + b)^2 \leq 2(a^2 + b^2)$ and in the second line we used Assumptions 4.1 and 4.3.

For convenience we will use the following notations:

$$
\widetilde{L} = \frac{4}{M^2} \left( \frac{3}{2} ML + L_0^2 \right), \ \widetilde{\sigma}_i^2 = \frac{8}{M^4} \left( 9M^2 \sigma_i^2 + 4L_0^2 \sigma_{0,i}^2 \right), \ \tilde{\sigma} = \frac{4}{M} \left( \frac{3}{2} \sigma + \frac{L_0}{M} \sigma_0 \right), \quad \text{(B.51)}
$$

which are the new Lipschitz constant of $\nabla \varphi \circ f_i$, the new local variance term of $\nabla \varphi \circ \ell_{S_i}$, and the new Lipschitz constant of $\varphi \circ f_i - \varphi \circ f_j$. Note that if we average after the composition, then the local variance would be:

$$
\mathbb{E}_{S_i \sim \mathcal{D}_i^m} \left\| \frac{1}{|S_i|} \sum_{(\boldsymbol{x},y) \in S_i} \nabla \varphi \circ \ell(\boldsymbol{\theta}, (\boldsymbol{x}, y)) - \nabla \varphi \circ f_i(\boldsymbol{\theta}) \right\|^2 \leq \frac{1}{|S_i|} \sum_{(\boldsymbol{x},y) \in S_i} \mathbb{E}_{(\boldsymbol{x},y) \sim \mathcal{D}_i} \|\nabla \varphi \circ \ell(\boldsymbol{\theta}, (\boldsymbol{x}, y)) - \nabla \varphi \circ f_i(\boldsymbol{\theta})\|^2
$$
$$
\leq \widetilde{\sigma}_i^2, \quad \text{(B.52)}
$$

where we can only use Cauchy–Schwarz inequality since $\varphi \circ \ell$ is biased. Therefore, if we do it in this way, the variance (upper bound) will be $m$ times larger than the current way, which will slow down the convergence.

Let us now follow the proof of FedAvg (Theorem 4.2) to prove the convergence of PropFair. Our proof follows the one of Theorem 4.2. Note that the global update now is:

$$
\boldsymbol{\theta}_{t+1} = \boldsymbol{\theta}_t - \eta \sum_{i=1}^{n} p_i \sum_{j=1}^{K_i} \widetilde{\boldsymbol{g}}_{t,j}^{(i)}, \quad \text{(B.53)}
$$

with $\widetilde{\boldsymbol{g}}_{t,j}^{(i)} = \nabla \varphi \circ \ell_{S_i^j}(\boldsymbol{\theta}_{t,j-1}^{(i)})$ and $S_i^j$ the $j^{\text{th}}$ batch from client $i$. Similar to eq. B.10 we obtain:

$$
\pi(\boldsymbol{\theta}_{t+1}) \leq \pi(\boldsymbol{\theta}_t) - \eta \mu \left( 1 - \frac{L\eta}{2} \mu \right) \|\nabla \pi(\boldsymbol{\theta}_t)\|^2
$$
$$
- \eta(1 - L\eta\mu) \left\langle \nabla \pi(\boldsymbol{\theta}_t), \sum_{i=1}^{n} p_i \sum_{j=1}^{K_i} (\widetilde{\boldsymbol{g}}_{t,j}^{(i)} - \nabla \pi(\boldsymbol{\theta}_t)) \right\rangle + \frac{L\eta^2}{2} \left\| \sum_{i=1}^{n} p_i \sum_{j=1}^{K_i} (\widetilde{\boldsymbol{g}}_{t,j}^{(i)} - \nabla \pi(\boldsymbol{\theta}_t)) \right\|^2. \quad \text{(B.54)}
$$

However, since $\widetilde{\boldsymbol{g}}_{t,j}^{(i)}$ is no longer unbiased, we need to rewrite eq. B.11 as:

$$\mathbb{E}\pi(\boldsymbol{\theta}_{t+1}) \leq \mathbb{E}\pi(\boldsymbol{\theta}_t) - \eta\mu\left(1 - \frac{L\eta}{2}\mu\right)\mathbb{E}\|\nabla\pi(\boldsymbol{\theta}_t)\|^2 + \eta(1 - L\eta\mu)\mathbb{E}\left[\|\nabla\pi(\boldsymbol{\theta}_t)\| \cdot \left\|\sum_{i=1}^{n} p_i \sum_{j=1}^{K_i}(\widetilde{\boldsymbol{g}}_{t,j}^{(i)} - \nabla\pi(\boldsymbol{\theta}_t))\right\|\right] +$$

$$+ \frac{L\eta^2}{2}\mathbb{E}\left\|\sum_{i=1}^{n} p_i \sum_{j=1}^{K_i}(\widetilde{\boldsymbol{g}}_{t,j}^{(i)} - \nabla\pi(\boldsymbol{\theta}_t))\right\|^2$$

$$\leq \mathbb{E}\pi(\boldsymbol{\theta}_t) + \left(-\eta\mu\left(1 - \frac{L\eta}{2}\mu\right) + \frac{1}{2}\eta(1 - L\eta\mu)\right)\mathbb{E}\|\nabla\pi(\boldsymbol{\theta}_t)\|^2 +$$

$$+ \left(\frac{L\eta^2}{2} + \frac{1}{2}\eta(1 - L\eta\mu)\right)\mathbb{E}\left\|\sum_{i=1}^{n} p_i \sum_{j=1}^{K_i}(\widetilde{\boldsymbol{g}}_{t,j}^{(i)} - \nabla\pi(\boldsymbol{\theta}_t))\right\|^2$$

$$\leq \mathbb{E}\pi(\boldsymbol{\theta}_t) - \eta\frac{11\mu - 6}{12}\mathbb{E}\|\nabla\pi(\boldsymbol{\theta}_t)\|^2 + \frac{\eta}{2}\mathbb{E}\left\|\sum_{i=1}^{n} p_i \sum_{j=1}^{K_i}(\widetilde{\boldsymbol{g}}_{t,j}^{(i)} - \nabla\pi(\boldsymbol{\theta}_t))\right\|^2, \tag{B.55}$$

where we recycled eq. B.12 and eq. B.35. Similar to eq. B.14 we write:

$$\widetilde{\boldsymbol{g}}_{t,j}^{(i)} - \nabla\pi(\boldsymbol{\theta}_t) = \widetilde{\boldsymbol{g}}_{t,j}^{(i)} - \nabla\varphi \circ f_i(\boldsymbol{\theta}_{t,j-1}^{(i)}) + \nabla\varphi \circ f_i(\boldsymbol{\theta}_{t,j-1}^{(i)}) - \nabla\pi(\boldsymbol{\theta}_t), \tag{B.56}$$

and using $\|\boldsymbol{a} + \boldsymbol{b}\|^2 \leq 2(\|\boldsymbol{a}\|^2 + \|\boldsymbol{b}\|^2)$ eq. B.55 becomes:

$$\mathbb{E}\pi(\boldsymbol{\theta}_{t+1}) \leq \mathbb{E}\pi(\boldsymbol{\theta}_t) - \eta\frac{11\mu - 6}{12}\mathbb{E}\|\nabla\pi(\boldsymbol{\theta}_t)\|^2 + \eta\mathbb{E}\left\|\sum_{i=1}^{n} p_i \sum_{j=1}^{K_i}(\widetilde{\boldsymbol{g}}_{t,j}^{(i)} - \nabla\widetilde{f}_i(\boldsymbol{\theta}_{t,j-1}^{(i)}))\right\|^2 +$$

$$+ \eta\mathbb{E}\left\|\sum_{i=1}^{n} p_i \sum_{j=1}^{K_i}(\nabla\widetilde{f}_i(\boldsymbol{\theta}_{t,j-1}^{(i)}) - \nabla\pi(\boldsymbol{\theta}_t))\right\|^2, \tag{B.57}$$

with $\tilde{f}_i$ a shorthand for $\varphi \circ f_i$. With eq. B.50 and similar to eq. B.16, we have:

$$\mathbb{E}\left\|\sum_{i=1}^{n} p_i \sum_{j=1}^{K_i}(\widetilde{\boldsymbol{g}}_{t,j}^{(i)} - \nabla\widetilde{f}_i(\boldsymbol{\theta}_{t,j-1}^{(i)}))\right\|^2 \leq \|\boldsymbol{p}\|^2 \sum_{i=1}^{n} K_i^2 \frac{\widetilde{\sigma}_i^2}{m}, \tag{B.58}$$

and similar to eq. B.22, we obtain:

$$\mathbb{E}\left\|\sum_{i=1}^{n} p_i \sum_{j=1}^{K_i}(\nabla\widetilde{f}_i(\boldsymbol{\theta}_{t,j-1}^{(i)}) - \nabla\pi(\boldsymbol{\theta}_t))\right\|^2 \leq 2\|\boldsymbol{p}\|^2 \sum_{i=1}^{n} K_i^2 \tilde{\sigma}^2 + 2\tilde{L}^2\|\boldsymbol{p}\|^2 \sum_{i=1}^{n} K_i \sum_{j=1}^{K_i} \mathbb{E}\|\boldsymbol{\theta}_{t,j-1}^{(i)} - \boldsymbol{\theta}_t\|^2. \tag{B.59}$$

For $j \in [K_i - 1]$, we can write similarly to eq. B.25:

$$\mathbb{E}\|\boldsymbol{\theta}_{t,j}^{(i)} - \boldsymbol{\theta}_t\|^2 = \mathbb{E}\|\boldsymbol{\theta}_{t,j-1}^{(i)} - \boldsymbol{\theta}_t - \eta\widetilde{\boldsymbol{g}}_{t,j}^{(i)}\|^2 \leq \left(1 + \frac{1}{2K_i - 1}\right)\mathbb{E}\|\boldsymbol{\theta}_{t,j-1}^{(i)} - \boldsymbol{\theta}_t\|^2 + 2K_i\eta^2\mathbb{E}\|\widetilde{\boldsymbol{g}}_{t,j}^{(i)}\|^2. \tag{B.60}$$

With the following equality:

$$\widetilde{\boldsymbol{g}}_{t,j}^{(i)} = (\widetilde{\boldsymbol{g}}_{t,j}^{(i)} - \nabla\tilde{f}_i(\boldsymbol{\theta}_{t,j-1}^{(i)})) + (\nabla\tilde{f}_i(\boldsymbol{\theta}_{t,j-1}^{(i)}) - \nabla\tilde{f}_i(\boldsymbol{\theta}_t)) + (\nabla\tilde{f}_i(\boldsymbol{\theta}_t) - \nabla\pi(\boldsymbol{\theta}_t)) + \nabla\pi(\boldsymbol{\theta}_t), \tag{B.61}$$

we use $\|\boldsymbol{a} + \boldsymbol{b} + \boldsymbol{c} + \boldsymbol{d}\|^2 \leq 4(\|\boldsymbol{a}\|^2 + \|\boldsymbol{b}\|^2 + \|\boldsymbol{c}\|^2 + \|\boldsymbol{d}\|^2)$ to obtain:

$$\mathbb{E}\|\widetilde{\boldsymbol{g}}_{t,j}^{(i)}\|^2 \leq 4\mathbb{E}\|\widetilde{\boldsymbol{g}}_{t,j}^{(i)} - \nabla\tilde{f}_i(\boldsymbol{\theta}_{t,j-1}^{(i)})\|^2 + 4\mathbb{E}\|\nabla\tilde{f}_i(\boldsymbol{\theta}_{t,j-1}^{(i)}) - \nabla\tilde{f}_i(\boldsymbol{\theta}_t)\|^2 + 4\mathbb{E}\|\nabla\tilde{f}_i(\boldsymbol{\theta}_t) - \nabla\pi(\boldsymbol{\theta}_t)\|^2 + 4\mathbb{E}\|\nabla\pi(\boldsymbol{\theta}_t)\|^2$$

$$\leq 4\frac{\widetilde{\sigma}_i^2}{m} + 4\tilde{L}^2\mathbb{E}\|\boldsymbol{\theta}_{t,j-1}^{(i)} - \boldsymbol{\theta}_t\|^2 + 4\tilde{\sigma}^2 + 4\mathbb{E}\|\nabla\pi(\boldsymbol{\theta}_t)\|^2. \tag{B.62}$$

Plugging it back into eq. B.60 we have:

$$
\begin{aligned}
\mathbb{E}\|\boldsymbol{\theta}_{t,j}^{(i)} - \boldsymbol{\theta}_t\|^2 &\leq \left(1 + \frac{1}{2K_i - 1} + 8K_i\eta^2\tilde{L}^2\right)\mathbb{E}\|\boldsymbol{\theta}_{t,j-1}^{(i)} - \boldsymbol{\theta}_t\|^2 + 8K_i\eta^2\left(\frac{\widetilde{\sigma}_i^2}{m} + \tilde{\sigma}^2\right) + 8K_i\eta^2\mathbb{E}\|\nabla\pi(\boldsymbol{\theta}_t)\|^2 \\
&\leq \left(1 + \frac{1}{K_i}\right)\mathbb{E}\|\boldsymbol{\theta}_{t,j-1}^{(i)} - \boldsymbol{\theta}_t\|^2 + 8K_i\eta^2\left(\frac{\widetilde{\sigma}_i^2}{m} + \tilde{\sigma}^2\right) + 8K_i\eta^2\mathbb{E}\|\nabla\pi(\boldsymbol{\theta}_t)\|^2 \\
&\leq K_i\left(\left(1 + \frac{1}{K_i}\right)^j - 1\right)\left(8K_i\eta^2\left(\frac{\widetilde{\sigma}_i^2}{m} + \tilde{\sigma}^2\right) + 8K_i\eta^2\mathbb{E}\|\nabla\pi(\boldsymbol{\theta}_t)\|^2\right),
\end{aligned}
\tag{B.63}
$$

where in the second line we used $\eta \leq \min_i\{\frac{1}{6K_i\tilde{L}}\}$, and the last line is telescoping. Similar to eq. B.33, summing over $j = 0, 1, \ldots, K_i - 1$ gives:

$$
\sum_{j=0}^{K_i-1}\mathbb{E}\|\boldsymbol{\theta}_{t,j}^{(i)} - \boldsymbol{\theta}_t\|^2 \leq 8(e-2)K_i^3\eta^2\left(\frac{\widetilde{\sigma}_i^2}{m} + \tilde{\sigma}^2 + \mathbb{E}\|\nabla\pi(\boldsymbol{\theta}_t)\|^2\right).
\tag{B.64}
$$

From eq. B.57 we have:

$$
\begin{aligned}
\mathbb{E}\pi(\boldsymbol{\theta}_{t+1}) &\leq \mathbb{E}\pi(\boldsymbol{\theta}_t) - \eta\frac{11\mu - 6}{12}\mathbb{E}\|\nabla\pi(\boldsymbol{\theta}_t)\|^2 + \eta\mathbb{E}\left\|\sum_{i=1}^n p_i\sum_{j=1}^{K_i}(\widetilde{\boldsymbol{g}}_{t,j}^{(i)} - \nabla\widetilde{f}_i(\boldsymbol{\theta}_{t,j-1}^{(i)}))\right\|^2 + \\
&\quad + \eta\mathbb{E}\left\|\sum_{i=1}^n p_i\sum_{j=1}^{K_i}(\nabla\widetilde{f}_i(\boldsymbol{\theta}_{t,j-1}^{(i)}) - \nabla\pi(\boldsymbol{\theta}_t))\right\|^2 \\
&\leq \mathbb{E}\pi(\boldsymbol{\theta}_t) - \eta\frac{11\mu - 6}{12}\mathbb{E}\|\nabla\pi(\boldsymbol{\theta}_t)\|^2 + \eta\|\boldsymbol{p}\|^2\sum_{i=1}^n K_i^2\frac{\widetilde{\sigma}_i^2}{m} + 2\eta\|\boldsymbol{p}\|^2\sum_{i=1}^n K_i^2\tilde{\sigma}^2 + \\
&\quad + 2\eta\tilde{L}^2\|\boldsymbol{p}\|^2\sum_{i=1}^n K_i\sum_{j=1}^{K_i}\mathbb{E}\|\boldsymbol{\theta}_{t,j-1}^{(i)} - \boldsymbol{\theta}_t\|^2 \\
&\leq \mathbb{E}\pi(\boldsymbol{\theta}_t) - \eta\frac{11\mu - 6}{12}\mathbb{E}\|\nabla\pi(\boldsymbol{\theta}_t)\|^2 + \eta\|\boldsymbol{p}\|^2\sum_{i=1}^n K_i^2\left(\frac{\widetilde{\sigma}_i^2}{m} + 2\tilde{\sigma}^2\right) + \\
&\quad + 2\eta\tilde{L}^2\|\boldsymbol{p}\|^2\sum_{i=1}^n 8(e-2)K_i^4\eta^2\left(\frac{\widetilde{\sigma}_i^2}{m} + \tilde{\sigma}^2 + \mathbb{E}\|\nabla\pi(\boldsymbol{\theta}_t)\|^2\right) \\
&= \mathbb{E}\pi(\boldsymbol{\theta}_t) - \eta\left(\frac{11\mu - 6}{12} - 16(e-2)\eta^2\tilde{L}^2\|\boldsymbol{p}\|^2\sum_{i=1}^n K_i^4\right)\mathbb{E}\|\nabla\pi(\boldsymbol{\theta}_t)\|^2 + \widetilde{\Psi}_\sigma \\
&\leq \mathbb{E}\pi(\boldsymbol{\theta}_t) - \eta\frac{11\mu - 9}{12}\mathbb{E}\|\nabla\pi(\boldsymbol{\theta}_t)\|^2 + \widetilde{\Psi}_\sigma,
\end{aligned}
\tag{B.65}
$$

where in the second inequality we used eq. B.58 and eq. B.59; in the third inequality we used eq. B.64, in the second last inequality, we denoted:

$$
\widetilde{\Psi}_\sigma = \eta\|\boldsymbol{p}\|^2\left[\sum_{i=1}^n K_i^2\left(\frac{\widetilde{\sigma}_i^2}{m} + 2\widetilde{\sigma}^2\right) + 16(e-2)\eta^2\tilde{L}^2\sum_{i=1}^n K_i^4\left(\frac{\widetilde{\sigma}_i^2}{m} + \widetilde{\sigma}^2\right)\right];
\tag{B.66}
$$

and in the last line, we note that:

$$
\frac{11\mu - 6}{12} - 16(e-2)\eta^2\tilde{L}^2\|\boldsymbol{p}\|^2\sum_{i=1}^n K_i^4 \leq \frac{11\mu - 9}{12},
\tag{B.67}
$$

since we assumed:

$$
\eta \leq \frac{1}{8\tilde{L}}\sqrt{\frac{1}{(e-2)\|\boldsymbol{p}\|^2(\sum_{i=1}^n K_i^4)}}.
\tag{B.68}
$$

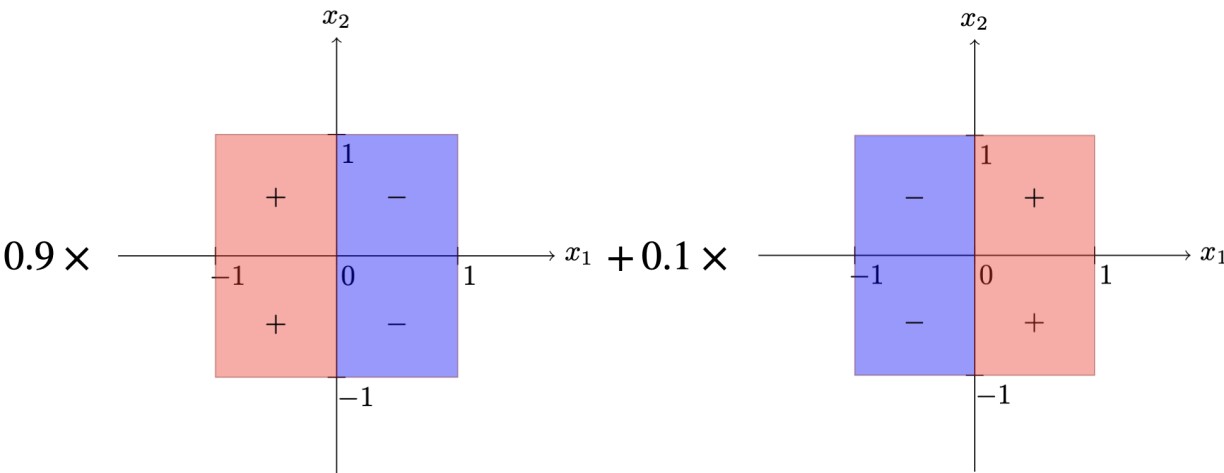

Figure 6: The visualization of the distribution shown in eq. C.1. The plus sign means the positive label $y = 1$ and the negative sign means the negative label $y = -1$.

Similar to eq. B.42 we obtain:

$$\min_{0 \le t \le T-1} \mathbb{E}\|\nabla \pi(\boldsymbol{\theta}_t)\|^2 \le \frac{12}{(11\mu - 9)\eta}\left(\frac{\pi(\boldsymbol{\theta}_0) - \pi^*}{T} + \widetilde{\Psi}_\sigma\right). \tag{B.69}$$

$\square$

## C  A Failure Case of Agnostic Federated Learning

In this section we show that AFL might suffer from the generalization issue, in the case when some of the clients have very few samples that are outliers. Suppose the input space is $\mathbb{R}^2$ and the classification task is binary with a linear classifier. We assume the simple case where every client has the same underlying distribution:

$$p(x|y) = \begin{cases} 0.9U([-1,0] \times [-1,1]) + 0.1U([0,1] \times [-1,1]) & \text{if } y = 1, \\ 0.9U([0,1] \times [-1,1]) + 0.1U([-1,0] \times [-1,1]) & \text{if } y = -1. \end{cases} \tag{C.1}$$

Note that $U(I)$ represents the density of the uniform distribution on interval $I$. A visualization of eq. C.1 can be found in Figure 6.

In practice, we draw samples from each of the client. However, if one of the clients do not have enough samples, AFL might have an issue. For instance, two clients could have to opposite sample sets:

$$S_1 = \{((0.5, \pm 0.5), 1), ((-0.5, \pm 0.5), -1)\}, \ S_2 = \{((0.5, \pm 0.5), -1), ((-0.5, \pm 0.5), 1)\}.$$

In this case, AFL could give an unfavorable generalization error, since the optimal training error is 50%. For example, this optimal AFL solution can be reached if one chooses the linear classifier to be perpendicular to the $x_1$-axis, resulting in the test error to be 50%. However, there exists an optimal classifier $\boldsymbol{w} = (-1, 0)$ such that the test error is 10%.

We can also verify this claim from the proof of Theorem 1 in Appendix C.2, Mohri et al. (2019). If one of the clients has too few samples (i.e., some $m_k$ is small), then the generalization bound on the right can be very large or even vacuous.

Note that our PropFair algorithm does not suffer from this generalization problem, since if some client $i$ has too few samples, then the corresponding weight $p_i = n_i/N$ will be small, and thus according to equation 3.7 the overall performance will not be heavily affected.

# D Additional Experiments

In this section, we provide more details about our experimental results. Results for all experiments are provided based on an average over three runs with different seeds.

## D.1 Datasets and models

We describe the benchmark datasets in this subsection. For all datasets we fix the batch size to be 64.

**CIFAR-{10, 100}** (Krizhevsky et al., 2009) are standard image classification datasets. There are 50000 samples with 10/100 balanced classes for CIFAR-{10, 100}. By doing Dirichlet allocation (Wang et al., 2019a) we achieve the heterogeneity of label distributions. For all samples in each class $k$, denoted as the set $\mathcal{S}_k$, we split $\mathcal{S}_k = \mathcal{S}_{k,1} \cup \mathcal{S}_{k,2} \ldots \mathcal{S}_{k,n}$ into $n$ clients according a symmetric Dirichlet distribution $\mathrm{Dir}(\beta)$. Then we gather the samples for client $j$ as $\mathcal{S}_{1,j} \cup \mathcal{S}_{2,j} \ldots \mathcal{S}_{C,j}$ if we have $C$ classes in total. We note that some of the clients might have too few samples (a few hundred). In this case the FL algorithm might overfit for such clients and we regenerate the data split. We choose the number of clients to be 10 for both CIFAR-{10, 100}. For each of the client dataset, we split it further into 80% training data and 20% test data.

**TinyImageNet** is from the course project of Stanford CS231N.[2] It contains 200 classes and each class has 500 images. Our FL split setting is the same as CIFAR-{10, 100}, except that we choose 20 clients and the Dirichlet parameter $\beta = 0.05$.

**Shakespeare** (Shakespeare, 1614; McMahan et al., 2017) is a text dataset of Shakespeare dialogues, and we use it for the task of next character prediction. We treat each speaking role as a client resulting in a natural heterogeneous partition. We first filter out the clients with less than 10,000 samples and sample 20 clients from the remaining. Also, each client's dataset is split into 50% for training and 50% for test.

In Table 3, we summarize these datasets, our partition methods, as well as the models we implement.

Table 3: Details of the experiments and the used datasets. ResNet-18 is the residual neural network defined in He et al. (2016). GN: Group Normalization (Wu & He, 2018); FC: fully connected layer; CNN: Convolutional Neural Network; Conv: convolution layer; RNN: Recurrent Neural Network; LSTM: Long Short-Term Memory layer. The plus sign means composition.

| Datasets | Training set size | Test set size | Partition method | # of clients | Model |
|---|---|---|---|---|---|
| CIFAR-10 | 39963 | 10037 | Dirichlet partition ($\beta = 0.5$) | 10 | ResNet-18 + GN |
| CIFAR-100 | 39764 | 10236 | Dirichlet partition ($\beta = 0.1$) | 10 | ResNet-18 + GN |
| TinyImageNet | 78044 | 20135 | Dirichlet partition ($\beta = 0.05$) | 20 | ResNet-18 + GN |
| Shakespeare | 178796 | 177231 | realistic partition | 20 | RNN (1 LSTM + 1 FC) |

## D.2 Algorithms to compare and tuning hyperparameters

We compare our PropFair algorithm with common FL baselines, including FedAvg (McMahan et al., 2017), $q$-FFL (Li et al., 2020c) and AFL (Mohri et al., 2019). For each dataset and each algorithm (algorithms with different hyperparameters are counted as different), we find the best learning rate from a grid. Here are the grids we used for each dataset:

- CIFAR-10: {5e-3, 1e-2, 2e-2, 5e-2};

- CIFAR-100: {5e-3, 1e-2, 2e-2, 5e-2};

- TinyImageNet: {5e-3, 1e-2, 2e-2, 5e-2};

---

[2]http://cs231n.stanford.edu/

- Shakespeare: {1e-1, 5e-1, 1, 2};

Table 4: The best values of hyperparameters used for different datasets, chosen based on grid search.

| Algorithm | Hyperparameter | CIFAR-10 | CIFAR-100 | TinyImageNet | Shakespeare |
|---|---|---|---|---|---|
| $q$-FFL | $q$ | 0.1 | 0.1 | 0.1 | 0.1 |
| TERM | $\alpha$ | 0.5 | 0.5 | 0.5 | 0.5 |
| GIFAIR-FL | $\lambda/\lambda_{\max}$ | 0.9 | 0.1 | 0.1 | 0.5 |
| FedMGDA+ | $\epsilon$ | 0.5 | 0.05 | 0.05 | 0.5 |
| PropFair | $M$ | 5.0 | 2.0 | 2.0 | 2.0 |

Table 5: The best learning rates used for different datasets and algorithms, based on grid search.

| Datasets | FedAvg | $q$-FFL | AFL | PropFair | TERM | GIFAIR-FL | FedMGDA+ |
|---|---|---|---|---|---|---|---|
| CIFAR-10 | 5e-3 | 5e-2 | 1e-2 | 5e-2 | 1e-2 | 1e-2 | 1e-2 |
| CIFAR-100 | 5e-3 | 2e-2 | 1e-2 | 1e-2 | 5e-3 | 1e-2 | 1e-2 |
| TinyImageNet | 2e-2 | 2e-2 | 2e-2 | 5e-2 | 2e-2 | 2e-2 | 1e-2 |
| Shakespeare | 2 | 2 | 2 | 2 | 2 | 2 | 2 |

We adapt hierarchical TERM from Li et al. (2020a), with client-level fairness ($\alpha > 0$) and no sample-level fairness ($\tau = 0$). For each dataset, we tune $\alpha$ (user-level parameter) from $\{0.01, 0.1, 0.5\}$. Table 4 shows the optimal value of $\alpha$ used for different datasets is 0.5. For AFL we tune the learning rate $\gamma_w$ from the corresponding grid and choose the default hyperparameter $\gamma_\lambda = 0.1$. For $q$-FFL, we run the $q$-FedAvg algorithm from Li et al. (2020c) with the default Lipschitz constant $L = 1/\eta$ from where $\eta$ is the learning rate.[3] For each dataset we tune $q$ from $\{0.1, 1.0, 5.0\}$. For all datasets we find $q = 0.1$ has the best performance. We also find that $q = 5$ often leads to divergence during training.

For PropFair we fix $\epsilon = 0.2$ and tune $M$ (Algorithm 1) from $M = 2, 3, 4, 5$. Table 4 shows the optimal values of $M$ used for different datasets. A rule of thumb is to first take a large $M$ (say $M = 10$) and then gradually reduce this value so as to obtain better performance. Given a learning rate $\eta$, we use the learning rate $\eta \frac{\epsilon}{M}$ when the loss is greater than $M - \epsilon$, and $\eta$ otherwise.

In addition to the fair FL algorithms in the main text, we compare with two additional baselines in our appendices: GIFAIR-FL (Yue et al., 2022) and FedMGDA+ (Hu et al., 2022). For GIFAIR-FL we first compute $\lambda_{\max}$ and choose $\lambda$ from $\{0.1\lambda_{\max}, 0.5\lambda_{\max}, 0.9\lambda_{\max}\}$. For FedMGDA+, we choose $\epsilon$ from $\{0.05, 0.1, 0.5\}$ as implemented in Hu et al. (2022). One minor difference is that we fix the global learning rate to be 1.0.

After finding the best hyperparameters for each algorithm, we record the best learning rates in Table 5. For CIFAR-10/CIFAR-100/TinyImageNet/Shakespeare, we take 100/400/400/100 communication rounds respectively, in which cases we find most fair FL algorithms converge.

## D.3 Detailed results

In Table 6, we report different statistics across clients, for all the algorithms and datasets we study in this work. These statistical quantities include:

- The mean of test accuracies of all clients;

- The standard deviation of client accuracies;

- The worst test accuracy among the clients;

- The mean of test accuracies across the worst 10% clients;

---

[3]https://github.com/litian96/fair_flearn/tree/master/flearn/trainers

- The best test accuracy among the clients.

- The mean of test accuracies across the best 10% clients.

For each algorithm we take three different runs and report the mean and standard deviation of different statistical indices. In all the experiments we have used 64 as the default batch size. Table 6 shows that PropFair is comparable with state-of-the-art algorithms across various datasets.

Table 6: Comparison among federated learning algorithms on CIFAR-10, CIFAR-100, TinyImageNet and Shakespeare datasets with test accuracies (%) from clients. All algorithms are fine-tuned. **Mean**: the average of performances across all clients; **Std**: standard deviation of client test accuracies; **Worst/Best**: the worst/best test accuracy from clients; **Worst (10%)/Best(10%)**: the average of performance across the worst/best 10% clients. Note that for CIFAR-{10, 100} the worst (best) case accuracy is the same as the worst (best) 10% accuracy since we have 10 clients.

| Dataset | Algorithm | Mean | Std | Worst | Worst (10%) | Best | Best (10%) |
|---|---|---|---|---|---|---|---|
| CIFAR-10 | FedAvg | $63.63_{\pm 0.48}$ | $5.38_{\pm 0.43}$ | $53.49_{\pm 1.67}$ | $53.49_{\pm 1.67}$ | $72.37_{\pm 0.53}$ | $72.37_{\pm 0.53}$ |
| | $q$-FFL | $57.27_{\pm 0.47}$ | $5.68_{\pm 0.16}$ | $47.28_{\pm 0.26}$ | $47.28_{\pm 0.26}$ | $66.71_{\pm 1.24}$ | $66.71_{\pm 1.24}$ |
| | AFL | $64.29_{\pm 0.40}$ | $4.48_{\pm 0.70}$ | $56.16_{\pm 1.56}$ | $56.16_{\pm 1.56}$ | $71.55_{\pm 0.84}$ | $71.55_{\pm 0.84}$ |
| | TERM | $63.81_{\pm 0.62}$ | $4.96_{\pm 0.42}$ | $56.22_{\pm 1.24}$ | $56.22_{\pm 1.24}$ | $71.51_{\pm 0.42}$ | $71.51_{\pm 0.42}$ |
| | GIFAIR-FL | $63.81_{\pm 0.23}$ | $5.05_{\pm 0.04}$ | $54.24_{\pm 1.14}$ | $54.24_{\pm 1.14}$ | $72.41_{\pm 0.88}$ | $72.41_{\pm 0.88}$ |
| | FedMGDA+ | $61.92_{\pm 0.93}$ | $4.93_{\pm 0.44}$ | $52.84_{\pm 1.12}$ | $52.84_{\pm 1.12}$ | $70.42_{\pm 1.72}$ | $70.42_{\pm 1.72}$ |
| | PropFair | $\mathbf{64.75}_{\pm 0.10}$ | $\mathbf{4.46}_{\pm 0.63}$ | $\mathbf{58.14}_{\pm 0.89}$ | $\mathbf{58.14}_{\pm 0.89}$ | $\mathbf{72.72}_{\pm 2.35}$ | $\mathbf{72.72}_{\pm 2.35}$ |
| CIFAR-100 | FedAvg | $29.94_{\pm 0.81}$ | $4.06_{\pm 0.37}$ | $25.26_{\pm 1.50}$ | $25.26_{\pm 1.50}$ | $40.29_{\pm 0.85}$ | $40.29_{\pm 0.85}$ |
| | $q$-FFL | $28.53_{\pm 0.58}$ | $4.53_{\pm 0.11}$ | $23.33_{\pm 0.72}$ | $23.33_{\pm 0.72}$ | $39.82_{\pm 1.02}$ | $39.82_{\pm 1.02}$ |
| | AFL | $30.33_{\pm 0.27}$ | $3.68_{\pm 0.40}$ | $25.49_{\pm 1.12}$ | $25.49_{\pm 1.12}$ | $39.21_{\pm 0.98}$ | $39.21_{\pm 0.98}$ |
| | TERM | $30.35_{\pm 0.28}$ | $3.50_{\pm 0.37}$ | $26.46_{\pm 0.36}$ | $26.46_{\pm 0.36}$ | $39.39_{\pm 0.90}$ | $39.39_{\pm 0.90}$ |
| | GIFAIR-FL | $30.63_{\pm 0.37}$ | $3.58_{\pm 0.17}$ | $26.99_{\pm 0.38}$ | $26.99_{\pm 0.38}$ | $40.03_{\pm 0.62}$ | $40.03_{\pm 0.62}$ |
| | FedMGDA+ | $23.69_{\pm 0.98}$ | $3.52_{\pm 0.33}$ | $19.01_{\pm 0.87}$ | $19.01_{\pm 0.87}$ | $32.51_{\pm 1.86}$ | $32.51_{\pm 1.86}$ |
| | PropFair | $\mathbf{31.84}_{\pm 0.67}$ | $\mathbf{3.10}_{\pm 0.47}$ | $\mathbf{28.85}_{\pm 0.94}$ | $\mathbf{28.85}_{\pm 0.94}$ | $\mathbf{40.12}_{\pm 1.80}$ | $\mathbf{40.12}_{\pm 1.80}$ |
| TinyImageNet | FedAvg | $16.14_{\pm 0.59}$ | $\mathbf{2.33}_{\pm 0.07}$ | $11.07_{\pm 0.78}$ | $11.81_{\pm 0.67}$ | $20.23_{\pm 1.11}$ | $19.91_{\pm 0.90}$ |
| | $q$-FFL | $\mathbf{18.84}_{\pm 0.02}$ | $3.23_{\pm 0.25}$ | $12.12_{\pm 0.58}$ | $13.06_{\pm 0.66}$ | $\mathbf{24.19}_{\pm 0.25}$ | $\mathbf{23.69}_{\pm 0.19}$ |
| | AFL | $16.43_{\pm 0.58}$ | $2.34_{\pm 0.04}$ | $11.34_{\pm 1.24}$ | $12.32_{\pm 0.66}$ | $20.70_{\pm 0.64}$ | $20.21_{\pm 0.49}$ |
| | TERM | $16.41_{\pm 0.29}$ | $2.75_{\pm 0.27}$ | $10.67_{\pm 0.47}$ | $11.55_{\pm 0.40}$ | $21.75_{\pm 1.19}$ | $20.97_{\pm 0.71}$ |
| | GIFAIR-FL | $16.54_{\pm 0.41}$ | $2.70_{\pm 0.17}$ | $11.34_{\pm 0.47}$ | $11.92_{\pm 0.15}$ | $22.28_{\pm 0.46}$ | $21.47_{\pm 0.50}$ |
| | FedMGDA+ | $13.94_{\pm 0.20}$ | $2.70_{\pm 0.30}$ | $9.45_{\pm 0.03}$ | $9.73_{\pm 0.12}$ | $19.15_{\pm 0.06}$ | $18.62_{\pm 0.53}$ |
| | PropFair | $18.04_{\pm 0.74}$ | $2.69_{\pm 0.08}$ | $\mathbf{12.63}_{\pm 1.57}$ | $\mathbf{13.51}_{\pm 1.19}$ | $23.68_{\pm 0.49}$ | $23.02_{\pm 0.30}$ |
| Shakespeare | FedAvg | $50.54_{\pm 0.12}$ | $1.22_{\pm 0.07}$ | $48.18_{\pm 0.17}$ | $48.26_{\pm 0.17}$ | $52.33_{\pm 0.29}$ | $52.15_{\pm 0.12}$ |
| | $q$-FFL | $50.69_{\pm 0.14}$ | $1.05_{\pm 0.02}$ | $48.74_{\pm 0.21}$ | $48.83_{\pm 0.22}$ | $52.35_{\pm 0.08}$ | $52.25_{\pm 0.13}$ |
| | AFL | $\mathbf{52.54}_{\pm 0.08}$ | $1.25_{\pm 0.05}$ | $\mathbf{49.86}_{\pm 0.29}$ | $\mathbf{50.13}_{\pm 0.12}$ | $\mathbf{54.47}_{\pm 0.29}$ | $\mathbf{54.22}_{\pm 0.18}$ |
| | TERM | $50.90_{\pm 0.11}$ | $1.27_{\pm 0.03}$ | $48.10_{\pm 0.15}$ | $48.45_{\pm 0.20}$ | $52.65_{\pm 0.39}$ | $52.47_{\pm 0.26}$ |
| | GIFAIR-FL | $50.67_{\pm 0.28}$ | $1.25_{\pm 0.04}$ | $48.22_{\pm 0.26}$ | $48.32_{\pm 0.30}$ | $52.50_{\pm 0.24}$ | $52.45_{\pm 0.21}$ |
| | FedMGDA+ | $44.17_{\pm 0.18}$ | $\mathbf{0.99}_{\pm 0.02}$ | $42.42_{\pm 0.10}$ | $42.67_{\pm 0.05}$ | $46.30_{\pm 0.20}$ | $46.10_{\pm 0.20}$ |
| | PropFair | $52.28_{\pm 0.08}$ | $1.20_{\pm 0.04}$ | $49.50_{\pm 0.41}$ | $49.76_{\pm 0.20}$ | $54.10_{\pm 0.11}$ | $53.88_{\pm 0.12}$ |

## D.4 Additional evaluation metrics

In this subsection, we perform comparison with baseline algorithms on CIFAR-100 using additional evaluating metrics, including worst 20% and 30% test accuracies. One can see that our algorithm remains the state-of-the-art among a large variety of algorithms.

Table 7: Comparison using worst 20% and 30% test accuracies on the CIFAR-100 dataset. The hyperparameters and learning rates are the same as in Table 4 and Table 5.

| Metric | PropFair | AFL | FedAvg | TERM | $q$-FFL | GIFAIR-FL | FedMGDA+ |
|---|---|---|---|---|---|---|---|
| **worst 20%** | $\mathbf{29.08}_{\pm 0.77}$ | $26.15_{\pm 0.69}$ | $25.68_{\pm 1.66}$ | $26.90_{\pm 0.33}$ | $23.94_{\pm 0.51}$ | $27.33_{\pm 0.35}$ | $19.77_{\pm 0.87}$ |
| **worst 30%** | $\mathbf{29.29}_{\pm 0.63}$ | $26.63_{\pm 0.24}$ | $26.26_{\pm 1.48}$ | $27.25_{\pm 0.25}$ | $24.53_{\pm 0.53}$ | $27.66_{\pm 0.23}$ | $20.33_{\pm 1.10}$ |

# E   Dual View of Fair FL Algorithms

In this section we derive the convex conjugates of the generalized means for each algorithm. We sometimes extend the domain of $\boldsymbol{f}$ to obtain a clear form of $\mathsf{A}_\varphi^*$, while ensuring the equality of eq. 2.10.

## E.1   Dual View of FedAvg

For FedAvg, we have $\varphi(t) = t$ and the generalized mean can be written as:

$$\mathsf{A}_\varphi(\boldsymbol{f}) = \sum_i p_i f_i, \tag{E.1}$$

where we extend the domain of $\boldsymbol{f}$ to be $\mathbb{R}^n$. The convex conjugate can be written as:

$$\mathsf{A}_\varphi^*(\boldsymbol{\lambda}) = \sup_{\boldsymbol{f} \in \mathbb{R}^n} (\boldsymbol{\lambda} - \boldsymbol{p})^\top \boldsymbol{f} \tag{E.2}$$

Solving it yields:

$$\mathsf{A}_\varphi^*(\boldsymbol{\lambda}) = \begin{cases} 0 & \text{if } \boldsymbol{\lambda} = \boldsymbol{p}, \\ \infty & \text{otherwise.} \end{cases} \tag{E.3}$$

Bringing the equation above to eq. 2.10 we obtain the original form of FedAvg.

## E.2   Dual View of $q$-FFL and AFL

Let us now derive the conjugate function for $q$-FFL. With $\varphi(t) = t^{q+1}$ ($q > 0$) we have:

$$\mathsf{A}_\varphi(\boldsymbol{f}) = \left( \sum_i p_i f_i^{q+1} \right)^{\frac{1}{q+1}}, \tag{E.4}$$

where we assume dom $\boldsymbol{f} = \mathbb{R}_+^n$. The convex conjugate can thus be written as:

$$\mathsf{A}_\varphi^*(\boldsymbol{\lambda}) = \sup_{\boldsymbol{f} \geq \boldsymbol{0}} \boldsymbol{\lambda}^\top \boldsymbol{f} - \left( \sum_i p_i f_i^{q+1} \right)^{\frac{1}{q+1}}. \tag{E.5}$$

If $\sum_i p_i^{-1/q} \lambda_i^{(q+1)/q} > 1$, we can take $f_i = \lambda_i^{1/q} p_i^{-1/q} t$ and the maximand of eq. E.5 becomes:

$$\boldsymbol{\lambda}^\top \boldsymbol{f} - \left( \sum_i p_i f_i^{q+1} \right)^{\frac{1}{q+1}} = \sum_i \lambda_i^{(q+1)/q} p_i^{-1/q} t - \left( \sum_i \lambda_i^{(q+1)/q} p_i^{-1/q} \right)^{\frac{1}{q+1}} t$$

$$= \left( \sum_i \lambda_i^{(q+1)/q} p_i^{-1/q} - \left( \sum_i \lambda_i^{(q+1)/q} p_i^{-1/q} \right)^{\frac{1}{q+1}} \right) t. \tag{E.6}$$

By taking $t \to \infty$ we have $\mathsf{A}_\varphi^*(\boldsymbol{\lambda}) = \infty$. Therefore we must constrain $\sum_i p_i^{-1/q} \lambda_i^{(q+1)/q} \leq 1$. In this case, we can utilize Hölder's inequality to obtain $\mathsf{A}_\varphi^*(\boldsymbol{\lambda}) = 0$. In summary, the convex conjugate for $\boldsymbol{\lambda} \geq \mathbf{0}$ is:

$$\mathsf{A}_\varphi^*(\boldsymbol{\lambda}) = \begin{cases} 0, & \text{if } \sum_i p_i^{-1/q} \lambda_i^{(q+1)/q} \leq 1, \\ \infty & \text{otherwise.} \end{cases} \tag{E.7}$$

Taking $q \to \infty$ the function above becomes the one for AFL:

$$\mathsf{A}_\varphi^*(\boldsymbol{\lambda}) = \begin{cases} 0, & \text{if } \|\boldsymbol{\lambda}\|_1 \leq 1, \\ \infty & \text{otherwise.} \end{cases} \tag{E.8}$$

### E.3 Dual View of TERM

We continue to derive the convex conjugate of the generalized mean of TERM. Recall that $\varphi(t) = e^{\alpha t}$ with $\alpha > 0$. The generalized mean can be written as:

$$\mathsf{A}_\varphi(\boldsymbol{f}) = \frac{1}{\alpha} \log \left( \sum_i p_i e^{\alpha f_i} \right), \tag{E.9}$$

where we extend the domain of $\boldsymbol{f}$ to be $\mathbb{R}^n$. The convex conjugate is:

$$\mathsf{A}_\varphi^*(\boldsymbol{\lambda}) = \sup_{\boldsymbol{f} \in \mathbb{R}^n} \boldsymbol{\lambda}^\top \boldsymbol{f} - \frac{1}{\alpha} \log \left( \sum_i p_i e^{\alpha f_i} \right). \tag{E.10}$$

If any $\lambda_i < 0$, we can take the corresponding $f_i \to -\infty$ and thus $\mathsf{A}_\varphi^*(\boldsymbol{\lambda}) = \infty$. If $\boldsymbol{\lambda}^\top \mathbf{1} \neq 1$, we can impose $\boldsymbol{f} = t\mathbf{1}$ and obtain:

$$\boldsymbol{\lambda}^\top \boldsymbol{f} - \frac{1}{\alpha} \log \left( \sum_i p_i e^{\alpha f_i} \right) = (\boldsymbol{\lambda}^\top \mathbf{1} - 1)t. \tag{E.11}$$

By taking $t \to \infty$ or $t \to -\infty$ we get $\mathsf{A}_\varphi^*(\boldsymbol{\lambda}) = \infty$. Now let us assume $\boldsymbol{\lambda} \geq \mathbf{0}$ and $\boldsymbol{\lambda}^\top \mathbf{1} = 1$. By requiring stationarity in eq. E.10 we find the necessary and sufficient optimality condition:

$$\lambda_i = \frac{p_i e^{\alpha f_i}}{\sum_i p_i e^{\alpha f_i}}, \tag{E.12}$$

which can always to satisfied with our assumption. Denote $c = \sum_i p_i e^{\alpha f_i}$ we can solve eq. E.12 to obtain $f_i = \frac{1}{\alpha} \log \left( \frac{c \lambda_i}{p_i} \right)$. Bringing it back to eq. E.10 the convex conjugate becomes:

$$\begin{aligned} \mathsf{A}_\varphi^*(\boldsymbol{\lambda}) &= \sum_i \frac{\lambda_i}{\alpha} \log \left( \frac{c \lambda_i}{p_i} \right) - \frac{1}{\alpha} \log c \\ &= \sum_i \frac{\lambda_i}{\alpha} \log \frac{\lambda_i}{p_i}, \end{aligned} \tag{E.13}$$

where we used the condition $\boldsymbol{\lambda}^\top \mathbf{1} = 1$. Since we have the constraint that $\boldsymbol{\lambda} \geq \mathbf{0}$, eq. 2.10 still holds. Therefore, we get:

$$\mathsf{A}_\varphi^*(\boldsymbol{\lambda}) = \begin{cases} \sum_i \frac{\lambda_i}{\alpha} \log \frac{\lambda_i}{p_i} & \text{if } \boldsymbol{\lambda} \geq \mathbf{0}, \boldsymbol{\lambda}^\top \mathbf{1} = 1, \\ \infty & \text{otherwise.} \end{cases} \tag{E.14}$$

### E.4 Dual View of PropFair

Let us derive the dual of the generalized mean for PropFair in the same framework as in Section 2.4. Note that

$$\varphi(t) = -\log(M - t), \tag{E.15}$$

and therefore the generalized mean is:

$$\mathsf{A}_\varphi(\boldsymbol{f}) = \varphi^{-1}\left(\sum_i p_i \varphi(f_i)\right) = M - \prod_{i=1}^n (M - f_i)^{p_i}, \tag{E.16}$$

where we require $\boldsymbol{f} \leq M\mathbf{1}$. We observe that $\mathsf{A}_\varphi$ is a convex function, since it is composition of the generalized geometric mean (which is concave) and affine transformation. Now we compute the dual function

$$
\begin{aligned}
\mathsf{A}_\varphi^*(\boldsymbol{\lambda}) &= \sup_{\boldsymbol{f} \leq M\mathbf{1}} \boldsymbol{\lambda}^\top \boldsymbol{f} - \mathsf{A}_\varphi(\boldsymbol{f}) \\
&= \sup_{\boldsymbol{f} \leq M\mathbf{1}} \boldsymbol{\lambda}^\top \boldsymbol{f} + \prod_{i=1}^n (M - f_i)^{p_i} - M
\end{aligned}
\tag{E.17}
$$

If any entry $\lambda_i$ is non-positive, clearly we can let $f_i \to -\infty$ so that $\mathsf{A}_\varphi^*(\boldsymbol{\lambda}) \to \infty$. For positive $\boldsymbol{\lambda}$, and $\prod_{i=1}^n \left(\frac{\lambda_i}{p_i}\right)^{p_i} < 1$, we can take $f_i = M - c\frac{p_i}{\lambda_i}$ and get:

$$
\begin{aligned}
\boldsymbol{\lambda}^\top \boldsymbol{f} + \prod_{i=1}^n (M - f_i)^{p_i} - M &= \sum_{i=1}^n (M\lambda_i - cp_i) + \prod_{i=1}^n \left(\frac{cp_i}{\lambda_i}\right)^{p_i} - M \\
&= M(\boldsymbol{\lambda}^\top \mathbf{1} - 1) + \left(\prod_i \left(\frac{p_i}{\lambda_i}\right)^{p_i} - 1\right) c
\end{aligned}
\tag{E.18}
$$

Since $c \geq 0$ is arbitrary, we can take $c \to \infty$ and thus $\mathsf{A}_\varphi^*(\boldsymbol{\lambda}) = \infty$. Otherwise, if $\prod_{i=1}^n \left(\frac{\lambda_i}{p_i}\right)^{p_i} \geq 1$, then we have:

$$
\begin{aligned}
\boldsymbol{\lambda}^\top \boldsymbol{f} + \prod_{i=1}^n (M - f_i)^{p_i} - M &= \prod_{i=1}^n (M - f_i)^{p_i} - \boldsymbol{\lambda}^\top (M\mathbf{1} - \boldsymbol{f}) + M(\boldsymbol{\lambda}^\top \mathbf{1} - 1) \\
&\leq \prod_{i=1}^n (M - f_i)^{p_i} - \prod_{i=1}^n \left(\frac{\lambda_i}{p_i}\right)^{p_i} \prod_{i=1}^n (M - f_i)^{p_i} + M(\boldsymbol{\lambda}^\top \mathbf{1} - 1) \\
&\leq M(\boldsymbol{\lambda}^\top \mathbf{1} - 1),
\end{aligned}
\tag{E.19}
$$

where in the second line we used the AM-GM inequality and in the last line we used $\prod_{i=1}^n \left(\frac{\lambda_i}{p_i}\right)^{p_i} \geq 1$. This equality can always be achieved by taking $\boldsymbol{f} = M\mathbf{1}$. In summary, we have:

$$
\mathsf{A}_\varphi^*(\boldsymbol{\lambda}) = \begin{cases} M(\boldsymbol{\lambda}^\top \mathbf{1} - 1), & \text{if } \boldsymbol{\lambda} \geq \mathbf{0} \text{ and } \prod_{i=1}^n \left(\frac{\lambda_i}{p_i}\right)^{p_i} \geq 1, \\ \infty, & \text{otherwise.} \end{cases}
\tag{E.20}
$$

We remark that $\mathsf{A}_\varphi^*$ is closed (since its domain is closed). If we want to enforce $\boldsymbol{f} \geq \mathbf{0}$ when computing the dual function, we simply apply the convolution formula:

$$\overline{\mathsf{A}}_\varphi^*(\boldsymbol{\lambda}) = \inf_{\boldsymbol{\lambda} \leq \boldsymbol{\gamma}} \mathsf{A}_s^*(\boldsymbol{\gamma}). \tag{E.21}$$

However, the formula for $\mathsf{A}_\varphi^*$ suffices for our purpose so we need not compute the above explicitly.

Applying the above conjugation result we can rewrite PropFair's generalized mean as:

$$\min_{\boldsymbol{\theta}} \mathsf{A}_{\varphi}(\boldsymbol{f}(\boldsymbol{\theta})) = \min_{\boldsymbol{\theta}} \max_{\boldsymbol{\lambda} \geq \boldsymbol{0}} \boldsymbol{\lambda}^{\top} \boldsymbol{f}(\boldsymbol{\theta}) - \mathsf{A}_{\varphi}^{*}(\boldsymbol{\lambda}).$$ (E.22)

We focus on the inner maximization so that we know the weights we put on each client:

$$\max_{\boldsymbol{\lambda} \geq \boldsymbol{0}} \boldsymbol{\lambda}^{\top} \boldsymbol{f}(\boldsymbol{\theta}) - \mathsf{A}_{\varphi}^{*}(\boldsymbol{\lambda}) = \max_{\boldsymbol{\lambda} \geq \boldsymbol{0}, \prod_{i=1}^{n} (\lambda_i/p_i)^{p_i} \geq 1} \boldsymbol{\lambda}^{\top} \boldsymbol{f} - (\boldsymbol{\lambda}^{\top} \boldsymbol{1} - 1) M$$

$$= \max_{\boldsymbol{\lambda} \geq \boldsymbol{0}, \prod_{i=1}^{n} (\lambda_i/p_i)^{p_i} \geq 1} M - \boldsymbol{\lambda}^{\top} (M\boldsymbol{1} - \boldsymbol{f}).$$ (E.23)

Using the AM-GM inequality we have:

$$\boldsymbol{\lambda}^{\top}(M\boldsymbol{1} - \boldsymbol{f}) \geq \prod_{i=1}^{n} \left(\frac{\lambda_i}{p_i}\right)^{p_i} \prod_{i=1}^{n} (M - f_i)^{p_i} \geq \prod_{i=1}^{n} (M - f_i)^{p_i},$$ (E.24)

where the equality is attained iff $\prod_{i=1}^{n} \left(\frac{\lambda_i}{p_i}\right)^{p_i} = 1$ and

$$\lambda_i \propto \frac{p_i}{M - f_i}.$$ (E.25)

Thus, we verify again that the optimal value of eq. E.23 is:

$$M - \prod_{i=1}^{n} (M - f_i)^{p_i} = \mathsf{A}_{\varphi}(\boldsymbol{f}),$$ (E.26)

and we retrieve our original objective. eq. E.25 tells us that we are essentially solving a linearly weighted combination of $f_1, \ldots, f_n$, but with more weights on the worse-off clients, since $\frac{p_i}{M - f_i}$ is larger for larger $f_i$.

# F   More Related Work

In this appendix we introduce more related work, including multi-objective optimization, fairness in FL, as well as various definitions of fairness from multiple fields.

## F.1   Multi-objective optimization

Multi-Objective Optimization (MOO) has been intensively studied in the field of operation research (Geoffrion, 1968; Yu & Zeleny, 1975; Jahn et al., 2009). The goal of MOO is to minimize a series of objectives $f_1, f_2, \ldots, f_n$ based on their best trade-offs. This is directly related to federated learning (Hu et al., 2022) because one can treat the loss function of each client as an objective.

In MOO, Pareto optimality is often desired. To find a Pareto optimum, one way is to use an *aggregating objective* (a.k.a. scalarizing function, Lootsma et al. 1995). We list some common choices of this aggregating objective:

- *Linear weighting method (Geoffrion, 1968):* this method converts MOO into the problem of minimizing the convex combination of client objectives:

$$\min_{\boldsymbol{x} \in \mathcal{X}} \sum_{i=1}^{n} \lambda_i f_i(\boldsymbol{x}),$$ (F.1)

  with $\boldsymbol{\lambda} \in \Delta_{n-1}$ in the $(n-1)$-simplex, and $\mathcal{X}$ the domain of $\boldsymbol{x}$. Such solution is always Pareto optimal and the method has been used in FedAvg (McMahan et al., 2017). A well-known difficulty is that it cannot generate point in the nonconvex part of the Pareto front (Audet et al., 2008).

- *Reference point* (Audet et al., 2008): This method requires proximity to the *ideal point*: $\boldsymbol{r} = (\min_{\boldsymbol{x} \in \mathcal{X}} f_1(\boldsymbol{x}), \ldots, \min_{\boldsymbol{x} \in \mathcal{X}} f_n(\boldsymbol{x}))$, measured by $\ell_q$-norm:

$$\min_{\boldsymbol{x} \in \mathcal{X}} \|\boldsymbol{f}(\boldsymbol{x}) - \boldsymbol{r}\|_q^q := \sum_{i=1}^{n} (f_i(\boldsymbol{x}) - r_i)^q, \tag{F.2}$$

with $\boldsymbol{f}(\boldsymbol{x}) := (f_1(\boldsymbol{x}), \ldots, f_n(\boldsymbol{x}))$ and $\|\cdot\|_q$ the $\ell_q$-norm ($q \geq 1$). This method has been applied to federated learning as $q$-FFL (Li et al., 2020c) (by assuming $\boldsymbol{r} = \boldsymbol{0}$).

- *Weighted geometric mean* (Lootsma et al., 1995): this method converts MOO to a single-objective formulation by maximizing the weighted geometric mean between elements of the *nadir point* and the client objectives:

$$\max_{x \in \mathcal{X}} \prod_{i=1}^{n} (q_i - f_i(\boldsymbol{x}))^{\lambda_i}, \text{ such that } f_i(\boldsymbol{x}) \leq q_i \text{ for any } i \text{ and } \boldsymbol{x} \in \mathcal{X}, \tag{F.3}$$

where $\boldsymbol{q}$ is called a *nadir point*, defined as (Lootsma et al., 1995):

$$q_i = \max_{j=1,2,\ldots,n} f_i(\boldsymbol{x}_j^*), \tag{F.4}$$

with $\boldsymbol{x}_j^* = \arg\min_{\boldsymbol{x} \in \mathcal{X}} f_j(\boldsymbol{x})$ the optimizer of function $f_j$. The $\lambda_i$'s are the weights for each client and they are positive. If we take $\boldsymbol{\lambda} = (\lambda_1, \ldots, \lambda_n) = \boldsymbol{1}$, then it resembles our objective in eq. 3.7.

## F.2 Fairness in Federated Learning

As FL has been deployed to more and more real-world applications, it has become a major challenge to guarantee that FL models has no discrimination against certain clients and/or sensitive attributes. Since different participants may contribute differently to the final model's quality, it is necessary to provide a fair mechanism to encourage user participation.

Besides the related work we mentioned in the main paper (McMahan et al., 2017; Mohri et al., 2019; Li et al., 2020b), another direction of research tries to directly encourage the involvement of user participation, by providing some rewards to fairly recognize the *contributions* of clients. For example, Lyu et al. (2020) designed a local credibility mutual evaluation mechanism to enforce good contributors get more credits. Concretely, each client computes the contribution of every other client by investigating the label similarities of the synthetic samples generated by the clients' differential private GANs (Goodfellow et al., 2014). Kang et al. (2020) proposed a pairwise measurement of contribution. Reputation scores are kept at each client for all other clients, and are updated by a multi-weight subjective logic model. Yu et al. (2020) proposed a Federated Learning Incentivizer (FLI) payoff-sharing scheme, which dynamically divides a given budget among clients by optimizing their joint utility while minimizing their discrepancy. The objective function takes into account the amount of payoff and the waiting time to receive the payoff. Wang et al. (2020) analyzed the contribution from the data side, and proposed the federated Shapley Value (SV) for data valuation. While preserving the desirable properties of the canonical SV, this federated SV can be calculated with no extra communication overhead, making it suitable for the FL scenarios.

The above methods already applied some objective functions that reflect the concept of proportional fairness, e.g., payoff proportional to the contribution. However, they mostly apply fixed contribution-reward assignment rules, without explicit definitions of proportional fairness or theoretical guarantee.

## F.3 Definitions of fairness

Fairness has been a perennial topic in social choice (Sen, 1986), communication (Jain et al., 1984), law (Rawls, 1999) and machine learning (Barocas et al., 2017). Whenever we have multiple agents and limited resources, we need fairness to allocate the resources. There have been many definitions of fairness, such as individual fairness (Dwork et al., 2012), demographic fairness, counterfactual fairness and proportional fairness.

In this section, we introduce definitions of fairness from various perspectives including social choice, communication and machine learning, and study the implications in the setting of FL.

### F.3.1 Social Choice and Law

We review some principles for fairness and justice in social choice (Sen, 1986) and law (Rawls, 1999), which resembles FL: we can treat the shared global model as a public policy and clients as social agents.

- *Utilitarian rule (Maskin, 1978):* suppose we have $n$ clients and their loss functions are $f_i$, the utilitarian rule aims to minimize the sum of the loss functions, e.g.,

$$\min_{\boldsymbol{\theta}} \sum_i f_i(\boldsymbol{\theta}), \tag{F.5}$$

with $\boldsymbol{\theta}$ the global model parameters. This utilitarian rule represents the utilitarian philosophy: as long as the overall performance of the whole society is optimal, we call the society to be fair. A utilitarian policy is Pareto-optimal but not vice versa. With model homogeneity, equation eq. F.5 is nothing but the objective for FedAvg (McMahan et al., 2017), although the FedAvg algorithm may not always converge to the global optimum even in linear regression (Pathak & Wainwright, 2020).

- *Egalitarian rule (Rawls, 1974; 1999):* The egalitarian rule, also known as the maximin criterion represents egalitarianism in political philosophy. Instead of maximizing the overall performance as in eq. F.5, an egalitarian wants to maximizing the performance of the worst-case client, i.e., we solve the following optimization problem:

$$\min_{\boldsymbol{\theta}} \max_i f_i(\boldsymbol{\theta}). \tag{F.6}$$

This accords with Agnostic FL (Mohri et al., 2019). The egalitarian problem eq. F.6 may not always be Pareto optimal, e.g., $(f_1, f_2, f_3) = (1, 1, 1)$ and $(f_1, f_2, f_3) = (1, 0.9, 0.8)$ can both be the optimal solution of eq. F.6, but the former is not Pareto optimal.

### F.3.2 Fairness in wireless communications

Since resource allocation is common in communication, different notions of fairness have also been proposed and studied. We review some common fairness definitions in communication:

- *Max-min fairness / Pareto optimal* (Bertsekas & Gallager, 1987): this definition says at the fair solution, one cannot simultaneously improve the performance of all clients, which is equivalent to the definition of Pareto optimal. The corresponding algorithm in FL for finding a Pareto optimum is FedMGDA+ (Hu et al., 2022).

- *Proportional-fair rule (Kelly, 1997; Bertsimas et al., 2011):* proportional fairness aims to find a solution $\boldsymbol{\theta}^*$ such that for all $\boldsymbol{\theta}$ in the domain:

$$\sum_i \frac{u_i(\boldsymbol{\theta}) - u_i(\boldsymbol{\theta}^*)}{u_i(\boldsymbol{\theta}^*)} \leq 0, \tag{F.7}$$

with $u_i$ the utility function of client $i$, e.g., the test accuracy. This problem aims to find a policy such that the total relative utility cannot be improved. Proportional fairness has been studied in communication (e.g. Seo & Lee, 2006) for scheduling but the application in FL has not been seen.

- *Harmonic mean* (Dashti et al., 2013): the method maximizes the harmonic mean of the utility functions of each client, that is, we solve the following optimization problem:

$$\max_{\boldsymbol{\theta}} \frac{n}{\sum_i u_i(\boldsymbol{\theta})^{-1}} \tag{F.8}$$

In a similar vein we can find its optimality condition, assuming the utility set $\mathcal{U}$ is convex:

$$\sum_{i=1}^n \frac{u_i - u_i^*}{(u_i^*)^2} \leq 0, \ \text{ for all } \boldsymbol{u} \in \mathcal{U}. \tag{F.9}$$

Compared to proportional fairness, it simply amounts to squaring the denominator.

### F.3.3 Fairness in machine learning

Fairness has been studied in machine learning for almost a decade (Barocas et al., 2017). A large body of work focuses on proposing machine learning algorithms for achieving different definitions of fairness. These definitions are often incompatible with each other, i.e., one cannot achieve two definitions of fairness simultaneously. Let us review some common definitions, using classification as an illustrating example:

- *Group fairness / statistical parity / demographic parity* (DP, Dwork et al., 2012; Zemel et al., 2013): this definition requires that the prediction is independent of the subgroup (e.g., race, gender). Denote $Y$ as the prediction and $S$ as the sensitive attribute, this definition requires $Y \perp S$, where the symbol $\perp$ denotes statistical independence. This is the simplest definition of fairness, and probably what people think of at a first thought. However, this definition can be problematic. For instance, suppose a subgroup of clients have poor performance (e.g. due to communication, memory), and then to achieve better group fairness one can deliberately lower the performance of high-performing clients, and thus the overall performance is lower. Moreover, DP would forbid us to achieve the optimal performance if the true labels are not independent of the sensitive attribute (Hardt et al., 2016; Zhao & Gordon, 2019).

- *Equalized odds (EO)* (Hardt et al., 2016): this defintion requires demographic parity given each true label class. Define $T$ as the random variable for the true label. Equalized odds requires that $Y \perp S \,|\, T$ for *any* $T$ and equal opportunity requires that $Y \perp S \,|\, T$ for *some* $T$. Different from DP, this conditioning allows the prediction to align with the true label. In the binary setting, EO and DP cannot be simultaneously achieved (Barocas et al., 2017).

- *Calibration / Predictive Rate Parity* (Gebel, 2009): this definition requires that among the samples having a prediction score $Y$, the expectation of the true label $T$ should match the prediction score, i.e., $\mathbb{E}[T|Y] = Y$. In the context of fairness, calibration says that $T \perp S \,|\, Y$. Under mild assumptions, calibration and EO cannot be simultaneously achieved (Pleiss et al., 2017). Similarly, calibration and DP cannot be simultaneously achieved.

- *Individual fairness* (Dwork et al., 2012): this concept requires that similar samples, as measured by some metric, should have similar predictions.

- *Counterfactual fairness* (Kusner et al., 2017): this definition requires that from any sample, the prediction should be the same had the sensitive attribute taken different values. It follows the notion of counterfactual from casual inference (Pearl, 2000).

- *Accuracy parity* (Zafar et al., 2017): the accuracy for each group remains the same.

Since many concepts conflict with each other (Barocas et al., 2017), there is no unified definition of fairness. In light of this, a dynamical definition of fairness has been proposed (Awasthi et al., 2020). Algorithms for achieving different definitions of fairness include mutual information (Zemel et al., 2013), representation learning (Zemel et al., 2013; Zhao & Gordon, 2019) and Rényi correlation (Baharlouei et al., 2019).

