# OpenReview forum: "Proportional Fairness in Federated Learning"
_TMLR — Accepted by TMLR_

### Review · Reviewer_zYYU · 2022-11-09

**Summary Of Contributions:**

In general, I find the idea of thinking about different notions of fairness in federated learning is interesting. This paper proposes proportional fairness, formulates another fair FL objective, discusses its solvers, provides some analysis, and conducts experiments. I like the unified view of summarizing some previous objectives using generalized mean, and I also enjoyed reading the dual formulations.

**Audience:**

Yes

**Broader Impact Concerns:**

None.

**Claims And Evidence:**

Yes

**Requested Changes:**

1. clearly discuss the rationale of mentioning NBS (along with axiomatic characterization) and multi-objective optimization, and why the dual view is important/how that helps

2. address my concerns about the correctness of convergence proof

3. explain more clearly why proportional fairness is important, and how it addresses limitations of previous fairness notions [because the paper in its current form is talking about what it is/how to interpret it, instead of why it is needed]

**Strengths And Weaknesses:**

In general, I find the idea of thinking about different notions of fairness in federated learning is interesting. This paper proposes proportional fairness, formulates another fair FL objective, discusses its solvers, provides some analysis, and conducts experiments. I like the unified view of summarizing some previous objectives using generalized mean, and I also enjoyed reading the dual formulations. My major concerns are as follows.

* It is not clear what proportional fairness means in practice, and what additional benefits it has relative to prior works. It is an interesting new objective, but the implications of minimizing such a formulation are not well illustrated. There are some nice mathematical interpretations of the proposed objective, but intuitively why proportional fairness is important is unclear. What are the limitations of previous fairness definitions? It provides a failure case of AFL on a simple problem, but does not discuss how/why the proposed approach can prevent such deficiencies. Throughout the paper, the submission measures fairness similarly to previous works (e.g., how uniform the performance is, and how good the tail performance is), and I am wondering why PropFair is better than q-FFL/TERM in terms of optimizing for tail performance.


* I am not sure if I understand why the convergence of PorpFair can follow the convergence of FedAvg. One major challenge is that the local mini-batch gradient at each local step is not an unbiased estimator of the full local gradient, unless f_i in Eq. (4.1) is the actual average loss of all samples on client i (which may be very expensive if there are many samples or computing f_i is costly).

    Please correct me if I am wrong. I briefly checked the proof, and it seems that In Algorithm 1, if K_i batches mean K_i mini-batches, then the current proof cannot go through, due to the issue I mentioned above.

Others:

* This submission mentions the equivalence between PropFair and Nash Bargaining solutions. It would be better if the paper can further explain why readers should care about such equivalence, as well as those axiomatic characterizations. Similarly, it spends much text on multi-objective optimization, but not clear how this is connected to the main message of the paper.

* section 2.4: ‘when s’/s’’ is convex’—what is s?


* after theorem 4.2: What does ‘uniform aggregation for Adam-type methods’ mean? It is a bit confusing because the discussion jumps from FedAvg to FedAdam, but the convergence here doesn’t consider adaptive optimizers. I am also wondering if it is easy to extend Reddi et al 2018 to support different number of local steps and aggregation by p_k (as those don’t seem to require lots of changes of the proof Reddi et al, but i am just guessing). It would be better to make this point more clear.

---

> ### Author Response · Authors · 2022-11-21
> **Response to Reviewer zYYU (part 1)**
>
> We would like to thank the reviewer for the constructive and timely feedback. We address the major concerns below:
>
> > **Q1:** It is not clear what proportional fairness means in practice. Explain more clearly why proportional fairness is important.
>
> **A1:** We have clearly demonstrated the practical meaning of PF in the introduction (the end of page 1), and we would like to reiterate here. Essentially, proportional fairness says that the total relative utility cannot be improved. In other words, at the PF solution, if some client improves its performance relatively by $p$ percent, then other clients will have to lose relatively, and the total relative decrease is at least $p$ percent.
>
> This concept is important when people care more about the relative performance gain/loss, e.g., in stock market, and in telecommunication networks. In FL, this concept is also needed if we seek the relative change of the client test accuracies, which we show in Figures 1-3.
>
> >**Q2:** What are the limitations of previous fairness definitions? What additional benefits does it have relative to prior works?
>
> **A2:** We have explained the limitations of previous definitions in the introduction and we would like to reiterate here. It is true that $q$-FFL, TERM and our PropFair all aim to achieve some balance between the tail performance and the average performance. However, $q$-FFL and TERM do not have clear intuition about the balance. Instead, our PropFair achieves the special tradeoff where the total relative utility cannot be improved. This perspective is novel and can be verified in experiments (Figures 1-3), by plotting the relative changes of each client performance.
>
> >**Q3:** It provides a failure case of AFL on a simple problem, but does not discuss how/why the proposed approach can prevent such deficiencies.
>
> **A3:** We have added the following discussion in the revised draft (Appendix C): ''Note that our PropFair algorithm does not suffer from this generalization problem, since if some client $i$ have too few samples, then the corresponding weight $p_i = n_i / N$ will be small, and thus according to eq. (3.7) the overall performance will not be heavily affected.''
>
> >**Q4:** Throughout the paper, the submission measures fairness similarly to previous works (e.g., how uniform the performance is, and how good the tail performance is), and I am wondering why PropFair is better than q-FFL/TERM in terms of optimizing for tail performance.
>
> **A4:** The main fairness metric of our work is proportional fairness, as shown in Section 5.2, rather than the uniformity or the tail performance. Since PropFair/$q$-FFL/TERM all aim to achieve some balance between the tail performance and the average performance, it is natural to compare them w.r.t. the tail and the average. The exact reason why PropFair is better than $q$-FFL or TERM in terms of the tail performance is unknown, but we can provide some evidence below:
>
> * For $q$-FFL, the main issue is the Lipschitz constant for each client loss, which is needed in the convergence proof. Take $$\varphi \circ f_i(\theta) = {f_i(\theta)^{q+1}}/(q+1),$$ the gradient can be computed as $$\nabla \varphi \circ f_i(\theta) = f_i(\theta)^q \nabla f_i(\theta).$$ If $f_i(\theta) > 1$ and $q$ is large, then the Lipschitz constant of $\varphi \circ f_i$ would be large, and thus the convergence of $q$-FFL would be affected. In fact, as we observe in practice, when $q = 5$, the $q$-FFL algorithm often diverges (see Appendix D.2). Therefore, although in principle, $q$-FFL can approximate AFL by taking $q \to \infty$, this approach suffers from convergence issue when $q$ is large, and thus the tail performance cannot always be improved.
> * For TERM, we would like to mention that TERM is very similar to AFL. The only difference is that during the aggregation, the weights of clients are taken to be some ''soft'' version of maximum, i.e., for each client $i$, the weight is:
> $$
> w_i = \frac{\exp(\alpha f_i)}{\sum_i \exp(\alpha f_i)}.
> $$
> Therefore, if $\alpha \to \infty$, the TERM algorithm would be very similar to AFL and thus it suffers from the same generalization problem as AFL (Appendix C). In fact, as we see in Table 6, the tail performances of AFL and TERM are quite similar.

---

> ### Author Response · Authors · 2022-11-21
> **Response to Reviewer zYYU (part 2)**
>
>
> >**Q5:** I am not sure if I understand why the convergence of PropFair can follow the convergence of FedAvg. One major challenge is that the local mini-batch gradient at each local step is not an unbiased estimator of the full local gradient. I briefly checked the proof, and it seems that In Algorithm 1, if $K_i$ batches mean $K_i$ mini-batches, then the current proof cannot go through, due to the issue I mentioned above.
>
> **A5:** Thank you for checking our proof and pointing out this issue. We agree that the convergence of PropFair is not a direct consequence of FedAvg, and its proof needs more careful analysis. We have modified the convergence proof of PropFair and now it does not rely on the unbiased estimator assumption of each mini-batch gradient. The convergence result is slightly weaker but the qualitative conclusions are the same (see Theorem 4.4 and its proof).
>
> >**Q6:** This submission mentions the equivalence between PropFair and Nash Bargaining solutions. It would be better if the paper can further explain why readers should care about such equivalence, as well as those axiomatic characterizations.
>
> **A6:** The equivalence is needed because finding Nash bargaining solutions (NBSs) is easier than directly using the definition, eq.(3.3). In other words, this connection allows us to provide a practical algorithm for finding PF solutions in FL, i.e., PropFair. The axiomatic characterizations are not the main focus of our work, but we could explain them more here:
>
> * Pareto optimality says that one cannot improve all client performances simultaneously, i.e., if ${\bf u}^*$ is the NBS, there does not exist another ${\bf u} \in \mathcal{U}$, ${\bf u} \neq {\bf u}^*$, and for each client, $u_i \geq u_i^*$. In contrast, the solution found by AFL is not Pareto optimal (see below eq.F.6, Section F.3.1).
> * The symmetry property says that if the utility set $\mathcal{U}$ is symmetric (i.e., if ${\bf u} \in \mathcal{U}$, then switching any two components, the resulting vector is still in $\mathcal{U}$), and $p_i = 1/n$ for all $i\in [n]$, then the Nash bargaining solution must satisfy ${\bf u}^* = u_0 \bf 1$, where $\bf 1$ is a vector with all elements one. This property means that if the utility set is symmetric for all clients, then the NBS will give a uniform solution, which agrees with the usual concept of fairness.
> * The scale equivariance property says that if we change the utility evaluation of each client $i$, $u_i$, by a positive factor $a_i > 0$, to $a_i u_i$, then the corresponding NBS changes by corresponding ratios, i.e., ${\bf u}^*$ becomes ${\bf a} \odot {\bf u}^* = (a_1 u_1^*, \dots, a_n u_n^*)$. This property says that NBS is independent of the units of measurement (Maschler-Solan-Zamir 2020). Note that $q$-FFL (or the corresponding $\alpha$-fairness with $\alpha \neq 1$) does not satisfy this property.
> * Finally, the monotonicity property means that if the NBS ${\bf u}^*$ on $\mathcal{U}$ is in a subset $\mathcal{U}' \subseteq \mathcal{U}$, then ${\bf u}^*$ is also an NBS on $\mathcal{U}'$. This property is also called the independence of irrelevant alternatives, saying that if we remove the irrelevant alternatives, the NBS remains the same.
>
> As we have seen above, each axiom helps us understand NBS better and distinguish it from other fairness concepts. Moreover, these axioms uniquely characterizes the Nash bargaining solution given convexity and compactness.
>
> >**Q7:** Similarly, it spends much text on multi-objective optimization, but not clear how this is connected to the main message of the paper.
>
> **A7:** We did not spend much text on multi-objective optimization except at the beginning of Section 2. The main message of this section is to show that existing fair FL approaches (e.g.FedAvg, AFL) are nothing but different ways to scalarize the vector loss ${\bf f} = (f_1, \dots, f_n)$. This allows us to look at the fair FL algorithms in a unified perspective.
>
> >**Q8:** section 2.4: ‘when $s'/s''$ is convex — what is $s$?
>
> **A8:** $s$ is the scalar function; it should be $\varphi$ and we have modified the text.

---

> ### Author Response · Authors · 2022-11-21
> **Response to Reviewer zYYU (part 3)**
>
> >**Q9:** after theorem 4.2: What does ‘uniform aggregation for Adam-type methods’ mean? It is a bit confusing because the discussion jumps from FedAvg to FedAdam, but the convergence here doesn’t consider adaptive optimizers.
>
> **A9:** the 'uniform aggregation' means that the weights for aggregation are uniform. For example, in line 7 of Algorithm 2, we take all $p_i$'s to be the same. We agree it is a bit confusing and thus we have modified the draft. Now this discussion only focuses on FedAvg.
>
> >**Q10:** I am also wondering if it is easy to extend Reddi et al 2018 to support different number of local steps and aggregation by $p_k$ (as those don’t seem to require lots of changes of the proof Reddi et al, but i am just guessing). It would be better to make this point more clear.
>
> **A10:**  It is possible to extend the proof of Reddi et al to different number of local steps and aggregation by $p_k$, but that requires rewriting the whole proof and changing the assumptions. As mentioned before, since Reddi et al 2018 studies Adam-type methods and our focus is on FedAvg and PropFair, we are not sure how to adapt their proof easily.
>
> >**Q11:** clearly discuss the rationale of mentioning NBS (along with axiomatic characterization) and multi-objective optimization, and why the dual view is important/how that helps
>
> **A11:** For the first part, see A6. The dual formulation is a way to ''linearize'' the fair FL objectives and it provides a unified framework to for the fair FL algorithms. In other words, all the fair FL algorithms we study are minimizing some weighted average of the client losses (see eq.(2.10)), and the linear weights can be computed explicitly so that we can compare the fair FL algorithms on the same ground (Table 1).
>
> >**Q12:** address my concerns about the correctness of convergence proof
>
> **A12:** See A5.
>
> >**Q13:** explain more clearly why proportional fairness is important, and how it addresses limitations of previous fairness notions [because the paper in its current form is talking about what it is/how to interpret it, instead of why it is needed]
>
> **A13:** See A1 and A2.

---

### Review · Reviewer_CEYg · 2022-11-09

**Summary Of Contributions:**

This paper proposes a new fairness notion for federated learning, named proportional fairness, which is rooted in cooperative game theory. The paper first presents a unified framework of existing fairness notions. Then, the paper shows the connection between a special case of $\alpha$-fairness and the Nash bargaining solution, which leads to the notion of proportional fairness. The idea of proportional fairness is that the average relative utility cannot be improved anymore. The paper shows that proportional fairness achieves a balance between maximizing the average and the worst-case utilities. Then, the paper derives the objective function for achieving proportional fairness, develops the PropFair algorithm for federated learning, and analyzes its convergence. Experiments verify the properties of the PropFair algorithm.

**Audience:**

Yes

**Broader Impact Concerns:**

I have no concerns.

**Claims And Evidence:**

Yes

**Requested Changes:**

Add a paragraph in the introduction to explain the difference between fair federated learning and general fair machine learning.

Add a notation table, since there are many notations in the paper.

Both “eq” and “equation” are used in the paper. The notations should be consistent throughout the paper.

Explain why in Section 3.1.1 the authors consider both minimizing and maximizing the same objective function.


**Strengths And Weaknesses:**

Overall it is a work of high quality. The paper proposes a new fairness notion for federated learning and conducts a comprehensive and solid analysis of the proposed notion. The connection between the proposed notion and the Nash bargaining solution makes the whole story quite convincing. The experiments using both image and language data verify that the developed objective function and algorithm indeed achieve proportional fairness.

The only issue I have is that the fairness notions in federated learning are quite different from those in the traditional machine learning context, like demographic parity, equal opportunity, etc. The authors can include a paragraph in the introduction for explaining their differences to the audience from general fair machine learning fields.

---

> ### Author Response · Authors · 2022-11-21
> **Response to Reviewer CEYg**
>
> Thank you for your positive feedback and we really appreciate it! We have modified the draft according to your suggestions.
>
> >**Q1:** Add a paragraph in the introduction to explain the difference between fair federated learning and general fair machine learning.
>
> **A1:** Thanks for the great suggestion and we have added the paragraph in the introduction. We had some explanation in Section 2 and we have appended it to the new paragraph.
>
> >**Q2:** Add a notation table, since there are many notations in the paper.
>
> **A2:** We added a notation paragraph at the end of the introduction and a notation table in Appendix A.
>
> >**Q3:** Both “eq” and “equation” are used in the paper. The notations should be consistent throughout the paper.
>
> **A3:** We have modified all the notations to eq. for consistency.
>
> >**Q4:** Explain why in Section 3.1.1 the authors consider both minimizing and maximizing the same objective function.
>
> **A4:** The reason why we consider minimizing the objective is that we should minimize each client loss; the reason why we consider maximizing the objective is that we
> should maximize each client utility. This dilemma comes from treating the utility as the loss.

---

### Review · Reviewer_zcsx · 2022-11-11

**Summary Of Contributions:**

- This paper proposes a novel fair federated learning objective based on the notion of proportional fairness. The proposed framework extends q-FFL and capture a case where q-FFL fails to capture.
- The authors provide convergence guarantee for the proposed objective.
- Empirical results from 4 different datasets show that the proposed method outperforms previous baselines.

**Audience:**

Yes

**Claims And Evidence:**

Yes

**Requested Changes:**

- Providing formal fairness guarantee would be great. With the current definition, it is not clear to me that 1. on the train distribution, how is proportional fairness different from perfect prediction (Note that different from other fairness metrics like representation parity or demographic parity where perfect prediction is just a special case for achieving fairness, proportional fairness seems to be suggesting a unique solution globally, which is essentially perfect prediction)? 2. on the test distribution, does the proposed algorithm provides any guarantee?
- Could the authors explain how proportional fairness relate to other fairness notions? (Like representation disparity, the one used in q-FFL)
- Could the authors provide ablation study on how the choice of $q$ affects the comparison between q-FFL and the proposed algorithm. As the authors stated, the proposed method could be viewed as 1-fairness. Therefore, I'm wondering whether we could view it as a special case of q-FFL by setting q to be values close to -1.

Nit:
- Section 2.1: 'approach to for' -> 'approach for'
- Section 2.2: 'constract' -> 'contrast'

**Strengths And Weaknesses:**

Strength:

- This work provides a uniformed framework for fair FL (Equation 2.9) that generalizes prior works (e.g. q-FFL, TERM, AFL).
- This work provides a new perspective of fairness in federated learning (namely proportional fairness).
- Experimental results seem to show that the proposed method achieves stronger average / worst 10% performance.

Weakness:

- The definition of proportional fairness seems more closely related to some kind of equilibrium rather than fairness. Specifically, what is the domain $\mathcal{U}$ here. By definition, it could be the case where, for example taking $u$ as the train accuracy, we have $u_1^*=1$ while $u_2^*=0$, without specifying what $\mathcal{U}$ is, which does not seem to be consistent / similar to other fairness definitions like representation disparity.
- It is not clear to me why we average the loss over the batch before applying $\-log_{[\varepsilon]}$ in line 9 of Algorithm 1. Could the authors explain why not directly apply the proposed loss function and then average the loss over a batch?
- The authors didn't provide formal fairness guarantee (for example, how does the model output by the algorithm satisfy the proportional fairness definition).

---

> ### Author Response · Authors · 2022-11-21
> **Response to Reviewer zcsx (Part 1)**
>
> Thank you for taking the time to read our manuscript and give multiple comments/questions. We are glad to clarify them.
>
> >**Q1:** The definition of proportional fairness seems more closely related to some kind of equilibrium rather than fairness. Specifically, what is the domain $\mathcal{U}$ here. By definition, it could be the case where, for example taking $u$ as the train accuracy, we have $u_1^* = 1$ while $u_2^* = 0$, without specifying what $\mathcal{U}$ is, which does not seem to be consistent / similar to other fairness definitions like representation disparity.
>
> **A1:** We argue that proportional fairness is indeed related to fairness, and we show it here. Since proportional fairness is equivalent to Nash bargaining solutions (NBS) through Prop. 3.1, it suffices to show it for NBS. As we have explained in our response A6 to Review zYYU, NBS satisfies the symmetry property (see e.g. Maschler et al. 2020). If the utility set is symmetric (i.e., if ${\bf u} \in \mathcal{U}$, then switching any two components, the resulting vector is still in $\mathcal{U}$), and $p_i = 1/n$ for all $i\in [n]$, then the Nash bargaining solution must satisfy ${\bf u}^* = u_0 \bf 1$, where $\bf 1$ is a vector with all elements one. This property means that if the utility set is symmetric for all clients, then the NBS will give a uniform solution, which agrees with the usual concept of fairness, i.e., representation disparity.
>
> As a simple example, we have added Figure 1 in the draft for illustration.
>
> We also explain the meaning of $\mathcal{U}$. In FL, if we treat $u$ as the training accuracy, then the utility set $\mathcal{U}$ becomes the set of possible training accuracies $(u_1(\theta), \dots, u_n(\theta))$, given some prediction model $\theta$. Although it is possible to have a model $\theta$ such that $u_1(\theta) = 1$ and $u_2(\theta) = 0$, this case will be rare.
>
> >**Q2:** It is not clear to me why we average the loss over the batch before applying $-\log_{[\epsilon]}$ in line 9 of Algorithm 1. Could the authors explain why not directly apply the proposed loss function and then average the loss over a batch?
>
> **A2:** The reason why we average the loss before the composition is to minimize the variance. As pointed out by Reviewer zYYU, after the composition, the gradient estimator is no longer unbiased. If we do it in this way, the variance (upper bound) will be $m$ times larger than the current way, which will slow down the convergence. The reviewer can verify our claim below eq.(3.8) and from eq. (B.50) to eq. (B.52).
>
> >**Q3:** The authors didn't provide formal fairness guarantee (for example, how does the model output by the algorithm satisfy the proportional fairness definition). Providing formal fairness guarantee would be great.
>
> **A3:** Our Theorem 4.4 gives formal guarantee for the convergence of PropFair to stationary points, and in the convex case, the stationary points are optimal. This means that our algorithm can approximately find the Nash bargaining solutions. By the equivalence in Prop. 3.1, the Nash bargaining solutions are surrogates of proportionally fair solutions, and for convex utility sets, they are equivalent. Therefore, our paper indeed provides some formal fairness guarantee, which we have also verified through experiments (Sec 5.2).
>
> >**Q4:** On the train distribution, how is proportional fairness different from perfect prediction (Note that different from other fairness metrics like representation parity or demographic parity where perfect prediction is just a special case for achieving fairness, proportional fairness seems to be suggesting a unique solution globally, which is essentially perfect prediction)?
>
> **A4:** We agree with the reviewer that proportional fairness is trying to achieve some equilibrium, as eq.(3.3) suggests. However, it differs from perfect prediction. If the model can perfectly predict, then all clients would have perfect utilities, and according to eq.(3.3), the solution is proportionally fair. However, in practice, perfect prediction is not always achievable, and it often happens in practice that by improving the prediction performance of one client, the performance of another client degrades. Even for neural networks, due to the model capacity, perfect prediction cannot be achieved and thus it is not in the feasible utility set, therefore, given the utility set described in A1, proportional fairness differs from perfect prediction which is not achievable.
>
>
> >**Q5:** on the test distribution, does the proposed algorithm provides any guarantee?
>
> **A5:** Generalization to the test distribution is an interesting problem to study, but that is not the main focus of our work. However, if our PropFair objective is small, then each client loss $f_i$ is also small. Therefore, we can use the conventional theory of generalization to provide guarantee on each client test distribution.

---

> ### Author Response · Authors · 2022-11-21
> **Response to Reviewer zcsx (Part 2)**
>
> >**Q6:** Could the authors explain how proportional fairness relate to other fairness notions? (Like representation disparity, the one used in q-FFL)
>
> **A6:** See A1. We have also discussed the relation to $\alpha$-fairness in Sec 2.3, and the relation to utilitarianism and egalitarianism below eq. (3.5).
>
>
> >**Q7:** Could the authors provide ablation study on how the choice of $q$ affects the comparison between q-FFL and the proposed algorithm. As the authors stated, the proposed method could be viewed as 1-fairness. Therefore, I'm wondering whether we could view it as a special case of $q$-FFL by setting $q$ to be values close to -1.
>
> **A7:** As we explain in Section D.2, we tune $q$ from $\{0.1, 1.0, 5.0\}$ (with additional tuning of learning rates) and find that the best $q$ is $0.1$. We use this best hyperparameter choice in our experiments. For the ablation study, we provide the following table to show the ablation study of $q$ in $q$-FFL:
> | Algorithm   | mean (%)  | std (%)  | worst (%) |
> | -------- | -------- | -------- | -------- |
> | $0.1$-FFL   | 57.19 | 5.52 | 47.55 |
> | $1.0$-FFL   | 44.48 | 6.31 | 34.42 |
> | $5.0$-FFL   | 32.52 | 5.38 | 22.00 |
> | PropFair    | 64.38 | 3.74 | 59.38 |
>
> Note that $q$-FFL (Li et al. 2020c) assumed that $q \geq 0$ and thus we cannot take $q < 0$, let alone $q = -1$. As we have explained in Sec 2.3 and Sec 3.1.1, taking $q = -1$ would result in the minimizing of $\sum_i p_i \log f_i$, which will encourage the clients to be more disparate.
>
> To further verify our claim, we tried to run $q$-FFL with $q = -0.5, -0.9, -0.99$ in our experiments. We found that in all cases, the training loss increases and thus the algorithm diverges. This is because the Lipschitz constant will be large if $q$ is negative: taking the gradient of $f_i^{q+1}/(q+1)$, we have $f_i^q \nabla f_i$, which goes to infinity if the loss is minimized, i.e., $f_i \to 0$.
>
> >**Q8:** Nit
>
> **A8:** Thanks for pointing out the typos. We have fixed them.

---

### Author Response · Authors · 2022-11-21
**Modifications of our manuscript**

We would like to thank all reviewers and the action editor for your time and effort on this manuscript and your constructive feedbacks. We have updated the draft accordingly and the changes are colored in blue for easy navigation. We summarize the main changes as follows:

* As suggested by Reviewer CEYg, we have added a paragraph in the introduction to explain the difference between fair federated learning and general fair machine learning;
* As suggested by Reviewer CEYg, we have added a paragraph in the introduction for the notations and a notation table in Appendix A;
* To answer the question of Reviewer zYYU, we have updated the convergence proofs of PropFair and FedAvg;
* To answer the question of Reviewer zYYU, we have added a paragraph in Appendix C to explain why the proposed PropFair can prevent such deficiencies.
* Other minor modifications for typos as also pointed out by Reviewer zcsx.
* Added Figure 1 to demonstrate the concept of Nash bargaining solutions, as asked by Reviewer zcsx and Reviewer zYYU.

---

### Decision · Action_Editors · 2022-12-21

**Recommendation:** Accept as is

**Comment:**

All three reviewers had positive feedback on the paper, but suggested that the authors address certain issues in order to improve the clarity and completeness of the work. These issues include providing formal fairness guarantees, clarifying the relationship between the proposed fairness notion and other fairness notions, providing more details on the implementation and experimental setup, and fixing the convergence proof. The authors successfully addressed these issues during the revision. I recommend accepting the paper as it is.

**Audience:**

Yes. This paper presents a valuable contribution by examining a significant problem at the intersection of machine learning fairness and federated learning. Those interested in these topics within the TMLR audience will likely find this paper interesting.

**Claims And Evidence:**

This paper introduces a new fairness notion called proportional fairness that considers the relative change in performance for each client and proposes an algorithm, PropFair, for finding proportionally fair solutions. This proposed definition is closely related to the existing fairness notion proposed in the digital communication literature, and the authors also clearly stated the connection. The algorithm is shown to converge and to achieve good fairness on various tasks.